# Uncovering the Computational Roles of Nonlinearity in Sequence Modeling Using Almost-Linear RNNs

**Manuel Brenner**[1,2,3]                                        *manuel.brenner@esi-frankfurt.de*

**Georgia Koppe**[1,4,5]                                       *georgia.koppe@iwr.uni-heidelberg.de*

[1]*Interdisciplinary Center for Scientific Computing, Heidelberg University* [2]*Ernst Strüngmann Institute of the Max Planck Society, Frankfurt* [3]*Dept. of Psychiatry, Goethe University Frankfurt* [4]*Hector Institute for AI in Psychiatry and Dept. of Psychiatry and Psychotherapy, Central Institute for Mental Health Mannheim* [5]*Hertie Institute for AI in Brain Health, Tübingen*

**Reviewed on OpenReview:** *https://openreview.net/forum?id=qI2Vt9P9rl*

## Abstract

Sequence modeling tasks across domains such as natural language processing, time-series forecasting, speech recognition, and control require learning complex mappings from input to output sequences. In recurrent networks, nonlinear recurrence is theoretically required to universally approximate such sequence-to-sequence functions; yet in practice, linear recurrent models have often proven surprisingly effective. This raises the question of when nonlinearity is truly required. In this study, we present a framework to systematically dissect the functional role of nonlinearity in recurrent networks— allowing to identify both when it is computationally necessary, and what mechanisms it enables. We address the question using Almost Linear Recurrent Neural Networks (AL-RNNs), which allow the recurrence nonlinearity to be gradually attenuated and decompose network dynamics into analyzable linear regimes, making the underlying computational mechanisms explicit. We illustrate the framework across a diverse set of synthetic and real-world tasks, including classic sequence modeling benchmarks, an empirical neuroscientific stimulus-selection task, and a multi-task suite. We demonstrate how the AL-RNN's piecewise linear structure enables direct identification of computational primitives such as gating, rule-based integration, and memory-dependent transients, revealing that these operations emerge within predominantly linear dynamical backbones. Across tasks, sparse nonlinearity plays several functional roles: it improves interpretability by reducing and localizing nonlinear computations, promotes shared (rather than highly distributed) representations in multi-task settings, and reduces computational cost by limiting nonlinear operations. Moreover, sparse nonlinearity acts as a useful inductive bias: in low-data regimes, or when tasks require discrete switching between linear regimes, sparsely nonlinear models often match or exceed the performance of fully nonlinear architectures. Our findings provide a principled approach for identifying where nonlinearity is functionally necessary in sequence models, guiding the design of recurrent architectures that balance performance, efficiency, and mechanistic interpretability.

## 1 Introduction

Modeling sequences with temporal dependencies remains a core challenge in machine learning, with applications spanning natural language processing, speech recognition, video analysis, control, and forecasting (Hochreiter and Schmidhuber, 1997; Vaswani et al., 2017; Lillicrap et al., 2019; Lim et al., 2021). Sequence modeling tasks often depend on long-range memory (Bengio et al., 1994), requiring models to store, transform, and retrieve

information across time. Classical results have long pointed to the necessity of nonlinear recurrence for such operations, showing that recurrent neural networks (RNNs) with nonlinear dynamics are both universal approximators and Turing complete (Siegelmann and Sontag, 1995).

Yet, recent work has revived interest in linear recurrent models, particularly due to their computational advantages (Orvieto et al., 2023). Linear systems support parallel computation and scale efficiently (Martin and Cundy, 2018), and - when equipped with expressive input and output mappings - perform surprisingly well on long-horizon tasks (Gu et al., 2022; Orvieto et al., 2023). These findings revive a deeper question: How can we systematically assess to what extent nonlinearity is necessary for temporal computation, and what role it plays?

This question is especially relevant when considering interpretability. Linear systems, despite their dynamical limitations (Strogatz, 2018), are analytically tractable and structurally transparent, making them appealing in settings where understanding the internal dynamics of a model is as important as its raw performance (as, for instance, in neuroscience; Sussillo and Barak (2013); Durstewitz et al. (2023)). Nonlinear systems, by contrast, afford richer dynamics, but typically introduce a higher degree of complexity. Reflecting this trade-off, the landscape of sequence models has grown into a diverse collection of architectures combining linearity and nonlinearity in different ways (Patro and Agneeswaran, 2024), for instance through mechanisms such as input-dependent parameter nonlinearities (Gu and Dao, 2023) or gating mechanisms (Mehta et al., 2023). These innovations, while empirically successful, often blur the functional roles of linear and nonlinear components, obscuring their individual contributions to sequence modeling.

Motivated by this, we aim to provide a framework that disentangles how, when, and if nonlinearity is functionally necessary, and to demonstrate its utility for uncovering the computational mechanisms underlying sequence modeling. We tackle this question using the Almost-Linear RNN (AL-RNN), a model initially proposed for dynamical systems reconstruction (Brenner et al., 2024a). Here, we repurpose the AL-RNN as a natural tool to investigate a wide class of sequence modeling tasks. It is uniquely suited for this purpose because it makes nonlinearity both controllable and observable. Built from both linear and piecewise-linear units, the AL-RNN allows to *tune* the strength of nonlinear recurrence by varying the number of nonlinear units. At the same time, its piecewise-linear structure partitions the hidden state space into distinct linear subregions. Tracking which subregion is active over time yields a symbolic bitcode trace of the computation, which we use to analyze the switching structure that implements temporal processing. Finally, the linear dynamics within each subregion enable complementary dynamical analyses (e.g., fixed points and stability) (Eisenmann et al., 2024), connecting the bitcode-level view to interpretable dynamical mechanisms.

We apply the AL-RNN across a diverse set of classic memory benchmarks, a multi-task setting (Driscoll et al., 2024), as well as an empirical dataset of neural recordings from a rodent performing a stimulus selection task, using each setting to demonstrate how nonlinear mechanisms can be systematically identified, while exploring scenarios where linearity suffices, minimal nonlinearity is beneficial, or strong nonlinearity is required. In doing so, we reveal task-specific nonlinear mechanisms such as gating, rule-based integration, and memory-dependent transients, often *embedded within a mostly linear dynamical backbone*.

In most single task settings, reducing nonlinearity matches or even exceeds the performance of fully nonlinear models while retaining interpretability and making computational motifs and mechanisms directly traceable. In the multi-task setting, it forces reuse of a small set of nonlinear computational motifs, promoting *shared representations* across tasks, improving sample efficiency and exposing a mechanistic map of cross-task structure. Collectively, our results suggest that sparse nonlinearity provides a useful inductive bias that enables the emergence of nonlinear motifs where needed, while preserving predominantly linear dynamics. Overall, our study sheds light on the minimum necessary conditions for memory in AL-RNNs and related architectures, offering a principled framework guiding the design of sequence models that are both computationally efficient and analytically tractable.

## 2 Related Work

The ability to represent, retain, and manipulate memory over time has been a central challenge in sequence modeling. Early RNNs, such as those proposed by Elman (1990) and Jordan (1997), introduced the paradigm

of learning temporal dependencies through hidden state recurrence. However, these simple RNNs struggled to capture long-range dependencies due to vanishing and exploding gradients (Bengio et al., 1994). This limitation was addressed by architectures like Long Short-Term Memory (LSTM) networks (Hochreiter and Schmidhuber, 1997) and Gated Recurrent Units (Chung et al., 2014). Such gated RNNs, and more recent extensions such as Long Expressive Memory (Rusch et al., 2022), established gated recurrence as a cornerstone of sequence modeling, supporting stable memory retention across extended time spans. These models successfully mitigated gradient issues, but the introduction of nonlinearity through gating also increased the complexity of their internal state dynamics. To address this, an alternative line of work has focused on combining the expressivity of nonlinear dynamics with the analytical tractability of linear models. This gave rise to piecewise linear and switching models—such as switching linear dynamical systems (Fox et al., 2008; Linderman et al., 2016; 2017) and piecewise linear recurrent neural networks (PLRNNs) (Durstewitz, 2017; Brenner et al., 2022; 2024a). These architectures segment the state space into locally linear regions, enabling rich dynamical behaviors through structured transitions between subspaces while preserving mathematical tractability.

While the computational complexity of RNNs scales with $\mathcal{O}(T)$, $T$ being the sequence length, the inherently sequential nature of RNNs makes their training inefficient to parallelize on modern GPU hardware. The Transformer architecture (Vaswani et al., 2017) has therefore largely replaced recurrence with global self-attention across many sequence-modeling tasks. Self-attention enables parallel sequence processing without relying on a persistent hidden state. This parallelism, however, comes at the cost of computational efficiency, scaling with $\mathcal{O}(T^2)$. Further, it models memory only within a finite context window. As a result, Transformers lack causal, evolving memory, making them harder to interpret and less inherently efficient for very long-range dependencies (Patro and Agneeswaran, 2024).

This quadratic scaling has motivated the exploration of more efficient alternatives for long-sequence modeling. Recent interest has focused on structured linear state-space models (SSMs). These models implement linear recurrence, which can be unrolled using spectral methods or convolutions. Advanced parallel scanning techniques, such as Blelloch's scan (Blelloch, 1990), enable linear RNNs to achieve highly efficient parallelization, reducing complexity to $\mathcal{O}(\log T)$ (Martin and Cundy, 2018). Theoretical advancements have even demonstrated that deep linear RNNs with nonlinear mixing layers are universal approximators of regular sequence maps, broadening their applicability to complex temporal tasks (Orvieto et al., 2023; 2024). Modern SSMs like S4 (Gu et al., 2022), grounded in HiPPO theory (Gu et al., 2020), leverage these principles to match, or even surpass, Transformers on long-range sequence benchmarks (Tay et al., 2021). The Legendre Memory Unit (LMU) (Voelker et al., 2019) similarly enforces a fixed linear memory subspace derived from continuous-time dynamics, combined with a nonlinear update. Recent extensions have introduced more efficient initialization schemes and streamlined designs (Gupta et al., 2022; Smith et al., 2023; Hasani et al., 2023), enhancing both stability and performance. Some models, like Mamba (Gu and Dao, 2023) and Gated SSMs (Mehta et al., 2023), reintroduce nonlinearity through structured gating mechanisms while preserving the parallelizable structure of linear SSMs. Sequence models also increasingly incorporate hybrid architectures that blend recurrent, convolutional, and attention mechanisms (Peng, 2023; Qin et al., 2023; Poli et al., 2023; Lieber et al., 2024; Sieber et al., 2024). However, this proliferation of architectures, driven primarily by performance and efficiency, makes mechanistic interpretation increasingly challenging, as the structural assumptions each model encodes often remain implicit.

## 3  Method

**Almost-Linear Recurrent Neural Networks**   To dissect the roles of linear and nonlinear recurrence, we adopt an SSM-inspired architecture: the recently proposed AL-RNN (Brenner et al., 2024a). In this model, latent dynamics evolve under a combination of linear and PWL transition functions, modulated by external inputs. Nonlinearities are applied only to a subset $P \leq M$ of the hidden state, making AL-RNNs a simplified variant of piecewise linear RNNs (Durstewitz, 2017; Brenner et al., 2022). An AL-RNN is defined by:

$$z_t = A z_{t-1} + W \Phi^*(z_{t-1}) + C s_t + h, \tag{1}$$

where $\boldsymbol{z}_t \in \mathbb{R}^M$ is the system's latent state. The function $\Phi^*(\cdot)$ applies a nonlinearity selectively to only the last $P$ units:

$$\Phi^*(\boldsymbol{z}_t) = [z_{1,t}, \cdots, z_{M-P,t}, \phi(z_{M-P+1,t}), \cdots, \phi(z_{M,t})]^T, \tag{2}$$

where $\phi(\cdot)$ is a scalar nonlinearity. The original AL-RNN formulation (Brenner et al., 2024a) used the ReLU nonlinearity $\phi(z) = \max(0, z)$. While we also tested other nonlinearities including GeLU, tanh, and hardtanh, only PWL functions like ReLU preserve the property that the system operates as a composition of linear dynamical systems. To avoid redundant parameterization, we define $\boldsymbol{A} \in \mathbb{R}^{M \times M}$ as a diagonal matrix where entries corresponding to the linear (first $M - P$) units are set to zero, $\boldsymbol{A} = \mathrm{diag}(0, \dots, 0, a_{M-P+1}, \dots, a_M)$, so that linear self-connections are only assigned to the nonlinear units. The matrix $\boldsymbol{W} \in \mathbb{R}^{M \times M}$ captures interactions based on the partially nonlinear transformed state $\Phi^*(\boldsymbol{z})$, $\boldsymbol{C} \in \mathbb{R}^{M \times K}$ weighs the K-dimensional external inputs $\boldsymbol{s}_t$, and $\boldsymbol{h} \in \mathbb{R}^M$ is a bias term.

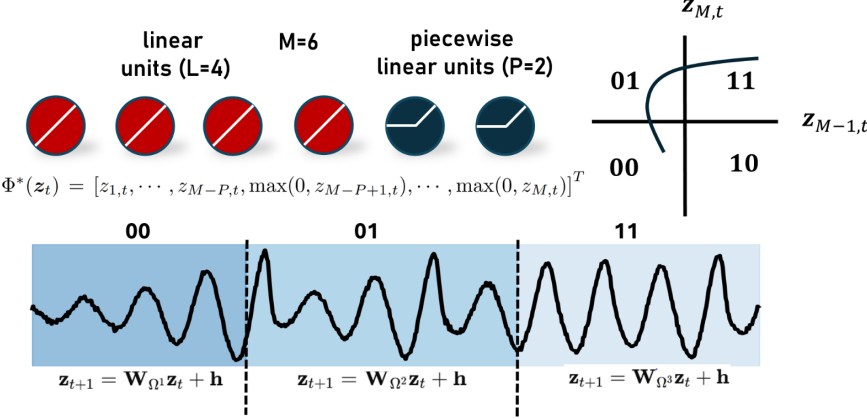

Figure 1: Illustration of the AL-RNN and bitcode assignment. The example displays a 6-dimensional AL-RNN with four linear and two PWL units (left). The PWL units partition the state space into four linear subregions (right). Bitcodes encode positive (1) and negative (0) activation values for these units. Each subregion corresponds to a unique bitcode (00, 01, 10, 11) and is governed by a distinct linear dynamical system (DS) with its own recurrence matrix $\mathbf{W}_{\Omega^i}$. An example trajectory (bottom) traverses a sequence of these subregions over time, experiencing discrete switches in dynamics when crossing subregion boundaries (marked by dashed vertical lines). Within each subregion, the dynamics remain linear.

**Computational Mechanisms in RNNs** Sequential tasks require recurrent networks to store information over time and to manipulate it in ways that depend on both past states and incoming inputs. These functions can be understood through the dynamics that govern hidden states and the computations they enable, implementing objects such as slow modes, attractors, and transient trajectories. From a computational perspective, these structures implement operations such as accumulation, recall, gating, and context-dependent routing.

**Linear RNNs** exhibit a restricted but analytically tractable set of dynamics, governed by the spectral properties of the recurrence matrix (Strogatz, 2018). Autonomous dynamics are limited to fixed points and modal decompositions: slow modes (eigenvalues near the unit circle), which support information storage through leaky integration or graded retention, complex eigenpairs, which generate oscillatory modes for rhythmic or phase-sensitive coding, and unstable modes, which can produce exponential growth or decay depending on gain and initialization (Seung, 1996; Ratcliff and McKoon, 2008; Gold and Shadlen, 2007; Henaff et al., 2016). These autonomous motifs can support stable (or deliberately unstable) memory traces but, because the update is globally linear, do not by themselves implement state-dependent reconfiguration. Input-driven dynamics in linear RNNs therefore reduce to uniform accumulation or filtering, lacking mechanisms for gating, selective recall, or context-dependent switching. A key strength lies in their analytic transparency and stability when dominated by slow modes, but their computational repertoire is fundamentally limited compared to nonlinear models.

**Nonlinear RNNs** differ both in terms of the autonomous dynamics they can generate, and how they can integrate external inputs. Autonomous nonlinear recurrence enables multistability, where multiple attractors allow for distinct, stable internal states that support e.g. associative memory (Hopfield, 1982; Amari, 1977). More complex motifs such as k-cycles, chaotic attractors, itinerancy, and metastability generate structured trajectories that do not converge to a single fixed point, supporting sequential processing and structured recall (Rabinovich et al., 2008; Tsuda, 2015). In terms of input integration, thresholded activations can act as switches, enabling state-dependent transitions and context-dependent routing (Balaguer-Ballester et al., 2011; Wang, 2008). This allows identical inputs to produce different outcomes depending on the internal state. Nonlinear gating, in turn, selectively integrates or suppresses information (Ackley et al., 1985). Nonlinear RNNs therefore excel when tasks require flexible memory or complex temporal structure, but their richer dynamics also make them harder to infer, analyze, and track compared to linear models.

**Piecewise Linear RNNs** aim to reconcile these trade-offs by combining the analytical tractability of linear dynamics with the computational expressivity of nonlinear models. From a dynamical systems perspective, piecewise linear systems offer a principled and general framework: any sufficiently smooth nonlinear DS can be approximated arbitrarily well by partitioning the state space into linear subregions (Storace and De Feo, 2004). The AL-RNN, which we employ here, implements the simplest version of this idea that can be learned automatically from data in an end-to-end fashion. It makes no assumptions about the switching mechanism itself, and requires no prior specification of how many subregions are needed or where transitions should occur. The AL-RNN partitions the state space into $2^P$ linear subregions separated by switching boundaries, within which the dynamics are linear and analytically tractable (see Fig. 1). Each linear subregion of the AL-RNN can be uniquely identified by a binary bitcode of length $P$, corresponding to the on/off (positive/negative) activation state of the $P$ ReLU units (Fig. 1,right). Since the ReLU nonlinearity partitions each dimension at zero, this yields $2^P$ possible configurations of active units. By tracking which bitcodes occur during inference, we can quantify and visualize how many of these subregions are actually used by the network, offering a compact representation of its functional complexity. This PWL structure thus inherits the advantages from both paradigms: the analytic transparency of linear systems within each subregion, combined with the representational flexibility of nonlinear models. Further, transitions between subregions are explicitly detectable, and within each regime, the system's dynamics can be analytically characterized in terms of fixed points and stability properties (Durstewitz, 2017; Eisenmann et al., 2024) (see Appx. A.2).

**Training Details** To promote the reconstruction of both fast and slow time scales - and associated memory - in the latent space, we incorporate a manifold regularization term based on Schmidt et al. (2021) (see Appx. A.1.2 for details and performance comparisons). The AL-RNN processes latent inputs $\{s_t\}_{t=1:T}$ derived from raw inputs $\{x_t\}_{t=1:T}$ (for details, see Appx. A.1.3), and task outputs are decoded from the latent states $\{z_t\}_{t=1:T}$ or the final state $z_T$ via a linear readout layer. This consistent use of linear decoding is central to our interpretability framework. By constraining the decoder to be linear, we ensure that the recurrent dynamics themselves must carry the full computational burden of the task solution. Since the AL-RNN implements a first-order Markov process, this task solution naturally takes the form of a DS, mirroring the classical computational neuroscience perspective on how stateful recurrence encodes computation through evolving trajectories (Durstewitz et al., 2023).

## 4 Experimental Results

**Task Selection and Experimental Structure** To probe how recurrent nonlinearity shapes memory, we selected a set of tasks spanning those that can be solved by linear dynamics and those that require nonlinear computation, allowing us to highlight distinct linear and nonlinear memory mechanisms (Sect. 3) engaged by different computational demands. In addition, to assess the robustness of the framework under more challenging conditions, we apply it to a multi-task setting and to an empirical neuroscientific dataset.
Our primary goal is interpretability rather than performance optimization, making direct architectural comparisons less straightforward. However, to contextualize AL-RNN performance, we included standard gated RNN baselines (LSTM, GRU) on the performance-oriented benchmarks (sMNIST, Speech Commands, Copy, Addition), and investigated the influence of sparse nonlinearity within S4 and a Linear Transformer on the Copy and Addition task (see Fig. 27 and Table 2). We also compared AL-RNNs with different activation

functions (ReLU, GeLU, tanh, hardtanh), finding qualitatively similar results (see Table 3), though some systematic differences emerge, discussed further in Sect. 4.3.

## 4.1 Modality-Specific Encoding Enables Simple Linear Integration

We begin with a class of tasks that require gradual accumulation of information over time to support a final decision. At each time step the AL-RNN's latent state $z_t$ integrates incoming representations $s_t$ over time, and the class label $\hat{y}$ is predicted from the final state $z_T$. We test this setup across images, audio, and text, employing domain-appropriate encoders to map raw inputs to latent representations: pretrained GloVe embeddings (Pennington et al., 2014) for text, 1D convolutional layers for sequential pixels, and spectral features for audio (see Appx. A.1.3). While these encoders necessarily differ in their treatment of modality-specific structure, they serve the common purpose of transforming heterogeneous inputs into a shared representational space amenable to temporal integration. For instance, the GloVe embeddings provide semantically meaningful representations where valenced words (e.g., "good" vs. "bad") are already separable, putting focus on the temporal integration of information rather than learning semantic structure from scratch. By constraining the recurrent dynamics to operate on these normalized representations and requiring classification through a linear readout, we enforce that the main computational mechanism, the temporal integration itself, is expressed explicitly in the AL-RNN's dynamics, revealing that despite the diversity of input modalities, the underlying task solution converges to the same dynamical principle across all three domains.

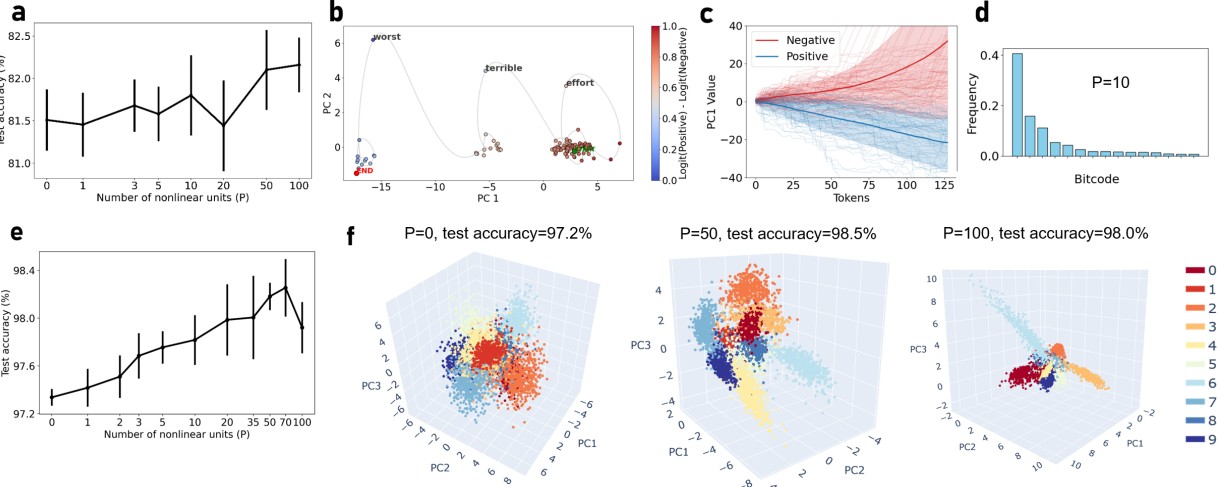

Figure 2: **Top row:** Sentiment classification on IMDb reviews. **a**: Test accuracy (y-axis) as a function of nonlinear units, $P$. **b**: Trajectory through latent space showing sentiment-relevant keywords guiding classification. **c**: The dominant first PC clearly separates positive and negative reviews. **d**: Bitcode frequencies are highly concentrated on a small subset of linear subregions. **Bottom row:** Digit classification on sequential MNIST. **e**: Test accuracy (y-axis) as a function of nonlinear units (x-axis; mean ± std over 10 seeds). **f**: Final latent state projection onto the first 3 PCs show that nonlinearity partitions latent space by class.

For *binary sentiment classification* using the IMDb dataset (Maas et al., 2011), the integration mechanism is remarkably simple, and increasing nonlinearity has minimal effect on performance (Fig. 2**a**; Welch's $t$-test comparing $P = 0$ vs $P = 100$: $t = -1.34$, $p = 0.20$). During a negative review (Fig. 2**b**), large updates to the latent state occur almost exclusively when highly valenced keywords such as *worst* or *terrible* are encountered, suggesting that memory integration is dominated by sparse lexical cues already emphasized by the input embedding. Accordingly, the dynamics are dominated by a single slow mode: an approximate line attractor with one eigenvalue near 1 and all others substantially smaller [1]. The eigenvector associated with the slow mode is almost perfectly aligned (cosine similarity > 0.999), which alone accounts for 98%

---

[1]We use the term "line attractor" in a functional sense: the dynamics exhibit very slow integration along a dominant mode (eigenvalue $\approx 1$) that preserves information over task-relevant timescales. Strictly speaking, true line attractors require a

of the total variance, indicating that the system's activity is effectively constrained to a one-dimensional (sentiment-related) integration axis (Fig. 2c), mirroring prior observations in the literature (Maheswaranathan et al., 2019). The distribution of nonlinear bitcodes is correspondingly degenerate (Fig. 2d): a handful of codes account for almost all examples.

We observed similar slow-mode integration mechanisms in image classification (Sequential MNIST= sMNIST) and audio classification (Speech Commands), detailed in Appx. A.4.2. While the overall dynamical mechanism is very similar, in both cases, the data manifold is higher-dimensional and requires more complex latent representations. Here, sparse nonlinearity provides a modest but significant improvement (Fig. 2e for sMNIST, Fig. 13 for Speech Commands).

For sMNIST, bitcode analysis reveals a clear mechanism by which nonlinearity affects performance: each of the 10 digit classes aligns with its own set of closely neighboring bitcodes, enabling class-specific linear integration dynamics (Fig. 10). While all classes are processed through slow modes with eigenvalues near 1 (Fig. 9), small differences in these eigenvalues, arising from distinct linear subregions, cause trajectories to diverge into more well-separated clusters (Fig. 2f). Performance peaks at intermediate nonlinearity and declines for fully nonlinear models, which overdisperse the data manifold (Fig. 12). The fact that this same slow-mode integration structure emerges across text, images, and audio demonstrates a universal computational motif for evidence accumulation tasks, consistent with similar mechanisms observed in biological systems (Seung, 1996; Ratcliff and McKoon, 2008; Maheswaranathan et al., 2019).

## 4.2 Minimal Nonlinearity Stabilizes Transients For Structured Memory Recall

Next, to evaluate the AL-RNN's ability to sustain stable internal representations over time, we tested it on the classic *copy task* (Fig. 3a). During the encoding phase, the network receives a random sequence of $N_{sym}$ distinct symbols, presented one per time step. This is followed by a delay of length $D$ time steps with no input, and then a "cue" signal indicates that the network should reproduce the original sequence over the next $N_{seq}$ time steps (the "recall phase"). Performance is measured as the number of sequences recalled correctly. This task is particularly interesting from a neuroscience perspective, as it parallels experimental paradigms used to study working memory in prefrontal cortex, where stable task-relevant information must be maintained over delay intervals despite complex and heterogeneous underlying dynamics (Murray et al., 2017; Rajan et al., 2016).

In this configuration ($N_{sym} = 4, N_{seq} = 8$ and $D = 200$), we first find that purely linear models perform above chance, suggesting they can partially solve the task. This aligns with theory: linear systems can support marginally stable cycles when eigenvalues lie on the unit circle, enabling long recurrence times that could, in principle, encode sequence and timing. However, such solutions are highly fragile to noise and perturbations (Strogatz, 2018), confirmed by the observation that despite extensive hyperparameter tuning, linear models fail to solve this task with high accuracy (Fig. 3c). Minimal nonlinearity therefore greatly improves performance and robustness. A single nonlinear unit ($P = 1$) often yields perfect recall, with performance declining for increased nonlinear units and being worst for fully nonlinear models (Fig. 3c).

To understand the underlying computational mechanism underlying successful solutions, we analyzed a representative AL-RNN with $P = 1$ that perfectly solves the task, i.e. decodes all encoded symbols with 100% accuracy on the test set. We find that the PWL unit leverages two distinct linear subregions to separate encoding/decoding vs. storage dynamics. During the encoding and decoding phases, the network remains confined to the primary linear subregion optimized for integrating and recalling symbols (Fig. 3b, top). In contrast, during the delay phase, the network transitions to the second linear subregion, where it implements a complex transient cycle that stores the encoded sequence. Because the encoded sequence uniquely determines the entry point into the cycle's transient, the network preserves this phase information throughout the delay, allowing the sequence to be decoded from the corresponding exit point at recall.

Crucially, we find that sparsely nonlinear models often only use a small subset of available linear subregions (Fig. 3d, Fig. 16 for a full statistical analysis). Fig. 17a illustrates why fully nonlinear models fail to converge

---

continuous manifold of neutrally stable fixed points, while our linear models instead have a single fixed point with one very slow decay mode.

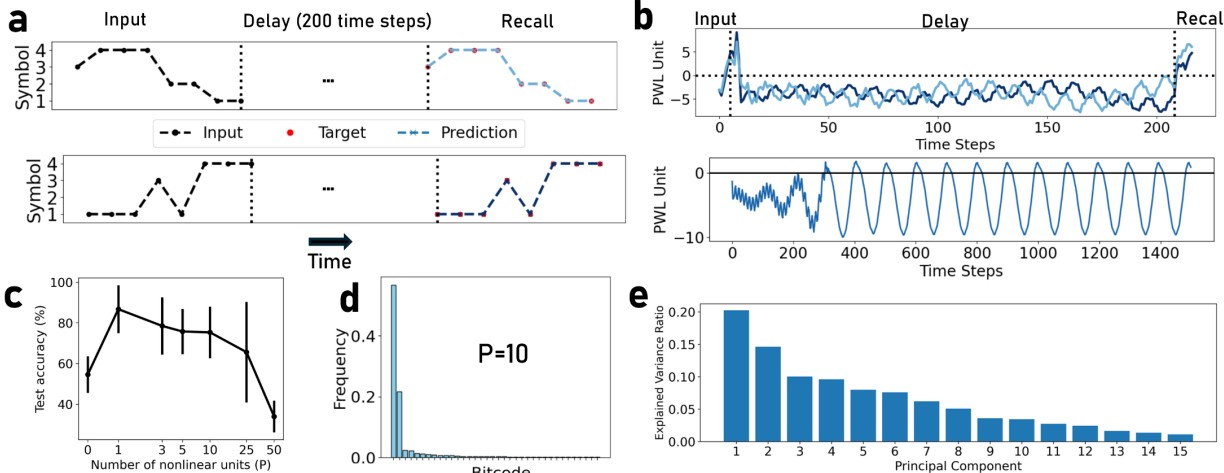

Figure 3: **a**: Structure of the copy task and two example trajectories for $P = 1$. **b**: Top: Activity of the PWL unit ($P = 1$) for the input sequences in **a**. The latent activity follows a complex limit cycle primarily located in one linear subregion (PWL unit negative), which switches to the second subregion during decoding (PWL unit positive). Bottom: Autonomous activity of the AL-RNN in the absence of inputs encodes a 100-cycle, with its transient located only in one subregion (PWL unit negative). **c**: Symbol-wise recall accuracy (mean± std over 10 seeds) vs. number of PWL units $P$. **d**: Histogram of binary "bitcodes" during recall for $P = 10$, concentrated on a small subset out of $2^{10} = 1024$ possible bitcodes. **e**: Explained variance ratio of PCs of latent network activity (for $P = 1$) indicates relatively high-dimensional, complex dynamics.

to good solutions: their loss curves are highly irregular and feature frequent bifurcations, directly linked to high gradient norms throughout training. For trained models, solutions are spread out across hundreds of subregions (bitcodes), failing to learn simple task solutions (Fig. 18**a**). For results on the variable delay task, where the recall cue occurs after a random interval, see Sect. A.4.3.

### 4.3 Nonlinearity Enables Gating in the Addition Task

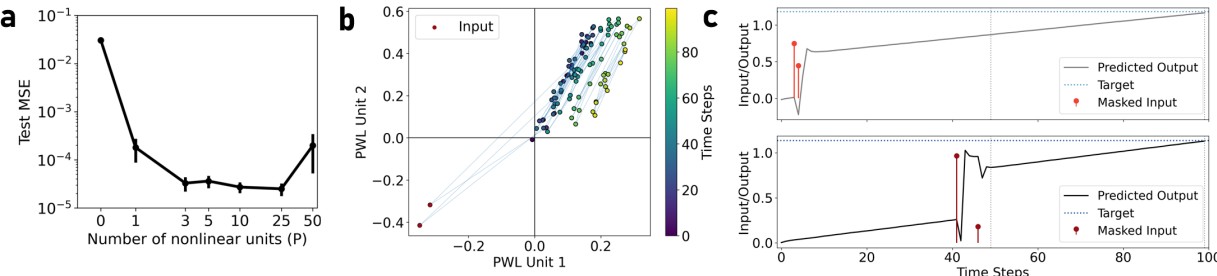

Figure 4: **a**: Performance as a function of units $P$ (mean± std over 10 seeds). **b**: Latent trajectories in PWL space with $P = 2$ nonlinear units. The second linear subregion is selectively activated only at the two masked time points ("input"), indicated by sharp transitions across quadrant boundaries. Outside these events, the network remains in a single linear regime, leading to smooth integration of the cumulative sum. **c**: Time course of masked inputs (red), network outputs (grey/black), and target values (blue) for two example trials, where the input occurs either early (light grey) or late (black) in the sequence. In both cases, the trajectory initially follows a linear drift prior to input, then transitions sharply to an elevated integration path with the same slope—offset to reach the target after 100 steps.

Next, we consider the *addition problem* (Hochreiter and Schmidhuber, 1997), an established test of selective memory. The input consists of a continuous stream of random numbers, along with a sparse binary mask

indicating two marked time points. The goal is to output the sum of the two marked times, ignoring all other inputs. Crucially, the mask is not directly fed into a separate gating mechanism, but the AL-RNN must internally learn to accumulate inputs only at the relevant times. This requires the network to integrate incoming values differently depending on the input. This task is impossible to solve by a linear AL-RNN (see Appx. A.4.1 for a formal proof), as accurately identified by our pipeline (Fig. 4a). In contrast, with just a single nonlinear unit ($P = 1$) the AL-RNN achieves orders-of-magnitude lower error.

Fig. 4b illustrates that nonlinearity enables the emergence of an internal gating mechanism, where the AL-RNN learns to selectively "open" and "close" an internal integration pathway conditional on the input mask. To sum the inputs, the network leverages a slow linear drift that accumulates over the full sequence length (Fig. 4c), while nonlinear units rapidly reposition the state onto the appropriate integration trajectory when gated inputs occur. This mechanism cleanly separates the functional contributions of linear and nonlinear units and aligns with theoretically derived optimal PWL solutions for this task previously described in the literature (Monfared and Durstewitz, 2020; Schmidt et al., 2021). Interestingly, as before, even with increased nonlinearity, the network preserves this strategy (Fig. 14a), distributing linear integration and gating across two distinct small sets of subregions. While fully nonlinear models can outperform sparsely nonlinear models (Fig. 18b), some fully nonlinear models fail to capture this simplicity of the task mechanism effectively, fragmenting the linear accumulation mechanism (Fig. 18b, middle vs. right), an effect that is conserved across training and test set (Fig. 14c). In line with this, we find that fully nonlinear models frequently "linearize" units by shifting their mean activity far from zero, effectively enforcing linear dynamics but at the cost of more difficult optimization (Fig. 14b).

Interestingly, all tested nonlinearities showed a similar performance pattern: marked improvement at $P = 1$, modest gains with sparse nonlinearity, stable plateaus at intermediate $P$, and worse performance at full $P$ (Table 3). What varied was how cleanly interpretable mechanisms emerged. The addition task requires combining two linear processes: slow drift accumulating time, and selective integration of marked inputs. PWL activations like ReLU naturally separate these into distinct linear subregions, whereas tanh must approximate the separation by shifting activity between near-linear and saturated regimes. This softer partition is harder to learn and yields roughly an order of magnitude higher MSE, highlighting the inductive bias of PWL units for switching-linear tasks.

### 4.4   Nonlinearity Facilitates Routing in Contextual Integration

Beyond simple integration, we examined context-dependent tasks where the model must respond differently to identical stimuli based on preceding rule cues. We tested this in two settings: a simple benchmark requiring conditional integration of a stimulus sequence, and a neuroscientific task involving joint modeling of both behavioral decisions and neural population activity from rodent auditory cortex (PFC-1 dataset, Rodgers and DeWeese (2014)). For the latter, we jointly trained the AL-RNN to both solve the task and reconstruct neural population activity from the same latent states.

In both settings, linear models fail completely, plateauing at chance performance, while introducing even a single nonlinear unit enables near-perfect accuracy (Appx. A.4.5 & A.4.4). In both cases, the central mechanism was straightforward to interpret: nonlinearity routes different task rules into distinct linear subregions, implementing context-dependent gating that linear dynamics fundamentally cannot achieve. While the overall task exhibits nonlinear dependencies, requiring opposite responses to identical inputs depending on context, the piecewise structure allows each subregion to implement its own linear solution to its respective subtask. Building on this insight that nonlinearity enables computational routing, we next explore how sparse nonlinearity structures computational reuse across multiple related tasks.

### 4.5   Shared Nonlinear Structure in a Multi-Task Paradigm

A natural extension of these results is to examine multi-task learning, where networks must balance task-specific specialization with cross-task generalization. This setting is particularly relevant from a neuroscientific perspective, as biological neural circuits routinely solve multiple related tasks with shared neural resources. We trained AL-RNNs on a suite of 11 cognitive benchmark tasks adapted from Driscoll et al. (2024), presented in randomly interleaved trials (see Appx. A.3 for details). The suite includes pro- and anti-response

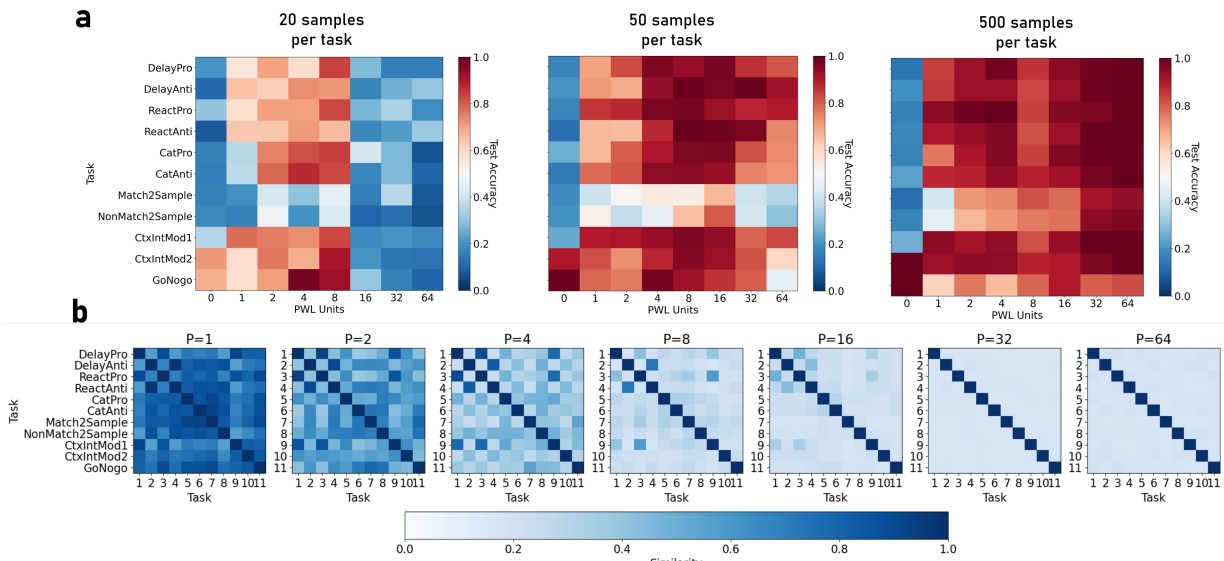

Figure 5: AL-RNN training on a *multi-task* paradigm. **(a)** Test accuracy (mean across 5 models per setting) across 11 cognitive tasks as a function of $P$ (PWL units), for different training set sizes. For limited data (20 samples per task), sparse nonlinearity ($P = 1$ to $P = 8$) outperforms both linear and fully nonlinear models by reusing structures tasks. For medium data (50 samples per task), more nonlinearity provides additional benefit, while sparse nonlinearity ($P = 4$ to $P = 16$) performs best. With abundant data (500 samples per task, right), this advantage diminishes as fully nonlinear models can afford task-specific solutions, though sparsely nonlinear architectures remain competitive. **(b)** Task similarity via Jensen-Shannon divergence between bitcode distributions (mean across 5 independent training runs). Sparse nonlinearity ($P = 2, 4$) reveals interpretable structure where related tasks share subregions—for instance, pro variants cluster together while anti variants form a separate group. As nonlinearity increases, this structure vanishes as the model has sufficient capacity to learn essentially independent representations for each task, reducing overlap and eliminating the interpretable similarity structure visible at lower $P$.

variants of delayed response, reaction time, and category decision tasks (requiring context-dependent mirror-image mappings), match and non-match variants of delayed match-to-sample, two context-integration tasks that demand selective attention to one of two sensory modalities, and a go/nogo task probing simple threshold detection. These tasks share a common temporal structure and input format, but require distinct computations, making them a natural testbed for understanding how nonlinearity supports flexible reuse of computations across tasks.

Fig. 5a shows test accuracy across the 11 tasks as a function of the number of PWL units $P$ and number of task samples used for training. While a significant portion of the multi-task suite requires nonlinearity, with linear models ($P = 0$) solving only two of eleven tasks, a single PWL unit ($P = 1$) achieves significant performance gains across nearly all tasks simultaneously. The pro/anti task pairs make this particularly transparent. Each variant is individually solvable by a linear system (e.g., mapping a stimulus to the same vs. opposite response), but a single linear network cannot implement both mappings at once, because it would need to "flip" the input-output relationship based on context. Introducing just one nonlinear unit provides this context-dependent switch: it allows the network to route pro and anti computations into different linear regions and thus solve both tasks simultaneously. Crucially, the same single PWL boundary is leveraged to achieve this separation for multiple different tasks simultaneously (see Fig. 24 for bitcodes and Fig. 25 for example latent trajectories).

The advantage of sparse nonlinearity depends strongly on the training data. With very limited training data (20 samples per task), sparse nonlinearity ($P = 1$ to $8$) substantially outperforms both linear and fully nonlinear architectures across most tasks. At intermediate data levels (50 samples per task), sparse models

maintain strong performance while fully nonlinear models begin to catch up. With abundant data (500 samples per task), fully nonlinear models achieve the best performance as they can fully converge to task-specific solutions without the constraint of shared subregions. Even in this high-data regime, however, sparse models ($P = 2$ to $8$) remain competitive.

This pattern aligns with the hypothesis that sparse nonlinearity is particularly advantageous in the low-data regime by forcing the network to reuse a small set of nonlinear computations across tasks. To test this "reuse" hypothesis, we examined the network's distribution of bitcodes on the test set, and quantified similarity via Jensen-Shannon divergence (Fig. 5b, see Fig. 24 for examples of bitcodes). The resulting similarity matrices reveal how tasks share nonlinear resources: at $P = 1$, the structure is relatively stark with binary either-or distributions, while $P = 4$ and $P = 8$ feature more nuanced relationships, where pro variants cluster together, anti variants form a separate group, and complex tasks like match-to-sample develop distinct signatures. This shared structure helps explain the sample efficiency advantage by enforcing cross-task generalization when data is scarce. At full nonlinearity ($P = 64$), this structure disappears to a near-diagonal matrix as tasks are spread across independent subregions without recurring shared patterns.

### 4.6 Nonlinear Decoding Enables Compositional Generalization

Finally, we investigate the *SCAN benchmark* (Lake and Baroni, 2018), which requires to systematically recombine learned action primitives according to syntactic structures, tapping into executive functions such as memory manipulation and control. In this setting, a sequence of linguistic commands (e.g., *run opposite left after walk right*) must be mapped to a corresponding sequence of low-level actions (e.g., *turn right, walk, turn left, turn left, run*; Fig. 8). The SCAN grammar is composed of conjunctions (*and*) and temporal dependencies (*after*), requiring the model to either execute commands sequentially or invert their order during decoding (Lake and Baroni, 2018).

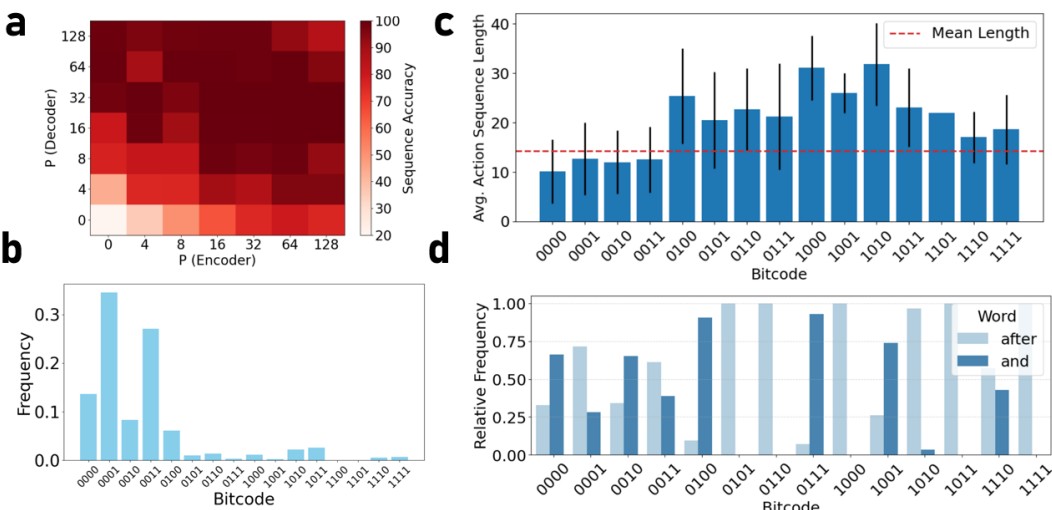

Figure 6: **a**: Sequence-level accuracy (in %) vs. number of nonlinear units $P$ in encoder (x-axis) and decoder (y-axis). **b**: Bitcode distribution of initial states of the decoder indicate a dominant subspace for which the first two units are zero. **c**: Average action sequence length associated with initial states lying in different subregions. **d**: Relative frequency of commands containing *and* and *after* within subregions shows clear syntactical separation.

We adopt a modular encoder–decoder architecture (illustrated in Fig. 8), assigning two separate AL-RNNs to encode the commands and to decode the resulting latent state into action sequences, separately varying the number of nonlinear units $P$ in the encoder and decoder AL-RNNs (out of a total of 128 units each). Purely linear models remain stuck at around $20 - 30\%$ sequence accuracy (Fig. 6a). Sparse nonlinearity ($P = 1, 4$) -

while improving performance - remains insufficient to solve the task reliably. Performance peaks when both encoder and decoder include moderate nonlinearity ($P = 32$), with near-perfect accuracy ($> 99.9\%$), and then slightly deteriorates for fully nonlinear models ($P = 128, \approx 96\%$). In addition, we observe a clear asymmetry: a linear encoder can suffice, achieving up to 99% accuracy when paired with a sufficiently nonlinear decoder ($P = 64$), while a linear decoder remains below 80% regardless of the encoder.

To better understand this behavior, we analyzed the latent structure produced by the encoder. Even when the encoder is purely linear, the final encoded states map onto a large number of distinct subregions in the decoder's PWL space: thus, while the encoder itself remains linear, end-to-end training organizes the latent representations such that the decoder's nonlinearity can be exploited effectively.

Second, the combination of a fully nonlinear encoder ($P = 128$) and a moderately nonlinear decoder ($P = 4$) can also achieve $> 99\%$ test accuracy while enabling a clear syntactic interpretation of the latent units. We found that this behavior emerges from the interaction between the expressive encoder, which maps syntactic constructs onto the PWL latent space of the decoder, effectively leveraging its subregions for decoding (Fig. 6). Specifically, a compact subspace of four subregions, with the first two PWL units inactive, captures most initial states for shorter sequences (Fig. 6**b**). In contrast, more complex and lengthier command sequences are consistently mapped to a separate latent subspace, where nonlinear dynamics become more pronounced (Fig. 6**c**). Within this second subspace, we observe a clear syntactic segregation, with commands containing *and* and *after* being mapped to mutually exclusive linear subregions (Fig. 6**d**). Additionally, we find that during decoding, transitions between composite commands in an action sequence are synchronized with sign flips in the PWL units (Fig. 26). Collectively, these findings suggest that while symbolic encoding in SCAN can remain largely linear, compositional generalization depends on the decoder's PWL structure tiling latent space into modular, syntax-aware subregion. This decomposition reveals a pathway toward interpreting how abstract syntactic operations are organized in more complex sequence tasks.

Table 1: Recurrent motifs identified by AL-RNNs across tasks, linking dynamics to computation.

| Task | Motif(s) | Computation |
|---|---|---|
| IMDb | Slow mode | Sentiment integration |
| sMNIST, SpeechCmd | Class-dependent slow modes | Evidence accumulation & pattern-specific routing |
| Addition | Slow mode + gating | Selective integration of marked inputs |
| Copy (fixed delay) | Limit cycle + subregion transition | Stable storage & timed recall |
| Copy (variable delay) | Limit cycle + subregion transitions | Flexible storage & sequence decoding |
| Context-dependent integration | Rule-specific slow modes | Rule-based integration |
| CRCNS-PFC-1 (rodent data) | Rule-specific slow modes | Rule-based routing |
| SCAN | Syntax-dependent subregion partitioning | Syntax-based modular decoding |
| Multi-task training | Task-specific subregion allocation | Compositional reuse of nonlinear motifs |

## 5 Discussion

This work asks whether—and how—systematically attenuating nonlinearity can make the computational mechanisms underlying sequence learning tasks explicit and tractable. Instead of treating nonlinearity as a binary design choice, we study how progressively tuning its strength reveals which aspects of temporal computation genuinely require nonlinear recurrence, and which can be supported by predominantly linear dynamics.

We pursue this question using the AL-RNN, whose piecewise-linear structure allows nonlinearity to be precisely controlled and directly observed. Varying the number of nonlinear units enables a step-wise interpolation between linear and nonlinear recurrence, while the induced partitioning of state space enables bitcode-based analyses and analytical characterization of the underlying dynamics. This combination allows us to directly identify mechanisms—such as gating, switching, and regime changes—through which nonlinear recurrence contributes to sequence computation.

Across tasks, we observe a consistent organizational principle, summarized in Table 1: memory itself is often implemented via a slow linear mode, while computational operations are layered on top through sparse nonlinear mechanisms, often realized as discrete switches/gates. The PWL structure of the AL-RNN makes

these mechanisms transparent, directly linking switches in model dynamics to computation. Our multi-task experiments further reveal that sparsely nonlinear models enable sample-efficient transfer learning, with computational reuse across tasks directly linked to shared linear subregions. This suggests that multi-task settings provide a promising avenue for understanding how neural systems discover and share computational primitives across related problems.

Beyond reducing interpretability, we find that for some tasks, fully nonlinear models indeed perform worse than a mix of linear and nonlinear units, which we could link to less stable optimization. We interpret this as a useful *inductive bias* of sparsely nonlinear models, resting on converging observations: low-$P$ models tend to yield compact, interpretable solutions, whereas fully nonlinear models fragment computation across many subregions; fully nonlinear models often end up "linearizing" units during training, which is harder to optimize than starting with linear units directly; and these effects are robust across activation functions. While tailored regularization or initialization could in principle mitigate these gaps, our results provide mechanistic insight into why sparse nonlinearity can perform better "out of the box", and suggest that the sparse introduction of nonlinearity in otherwise linear backbones could be leveraged as a design principle for more sophisticated sequence modeling architectures.

A natural concern is that our findings may be overly specific to the AL-RNN and its PWL activations. We use the AL-RNN in a deliberately reductionist way - not to argue that PWL nonlinearities are the "right" way to solve these tasks, but to break the model down into the simplest components needed for understanding. The PWL formulation makes it easy to see when and how the network's dynamics change, which in turn makes the underlying computations more transparent. Our goal is therefore not necessarily invariance at the level of low-level mechanisms, since different nonlinearities can realize the same computation in different ways, but invariance at the level of functional roles. From this perspective, the patterns we observe—largely linear memory dynamics interrupted by occasional nonlinear events that gate, switch, or change regimes—are not specific to the AL-RNN, but reflect common strategies that recur across tasks and activation functions.

More broadly, this structured separation of linear and nonlinear dynamics provides a useful lens for analyzing latent neural dynamics where similar mechanisms underlie storage, gating, and context-dependent computation (Durstewitz et al., 2023; Wang, 2008; Mante et al., 2013; Cho et al., 2014). Our results demonstrate how structured models can expose such mechanisms even in noisy, real-world settings, and suggest that modular, sparsely nonlinear latent representations - reminiscent of cortical organization (Fusi et al., 2016; Goyal et al., 2021) - offer a promising pathway toward mechanistic interpretability in both artificial and biological systems. More broadly, the structured separation of linear and nonlinear dynamics provides a useful lens for analyzing latent neural dynamics where similar mechanisms underlie storage, gating, and context-dependent computation (Durstewitz et al., 2023; Wang, 2008; Mante et al., 2013; Cho et al., 2014). Our results demonstrate how structured models can expose such mechanisms even in noisy, real-world settings, and suggest that modular, sparsely nonlinear latent representations - reminiscent of cortical organization (Fusi et al., 2016; Goyal et al., 2021) - offer a promising pathway toward mechanistic interpretability in both artificial and biological systems, suggesting potential applications to understanding how language models perform complex reasoning.

**Limitations and Future Directions**  Our study covers a structured but limited set of sequence tasks. While the identified dynamics capture important computational strategies, many real-world tasks, particularly in language, may blend these mechanisms more intricately. While we conducted preliminary analyses of sparse nonlinearity in more complex architectures, including S4 (Gu et al., 2022) and linear Transformers (Katharopoulos et al., 2020), understanding the role of nonlinearity in these models presents distinct challenges, since here, linear recurrence and linear attention mechanisms are integral to optimization and cannot be straightforwardly replaced with AL-RNN-style sparse PWL reccurence. Moreover, when nonlinearity is introduced through layer-wise transformations rather than within the recurrence itself, the connection to interpretable dynamical objects (fixed points, eigenspectra, flow fields) becomes less clear. Although we observed benefits of strategically placed sparse nonlinearity even in these architectures for the tasks tested (Fig. 27), a comprehensive mechanistic understanding of these effects requires future work. Extending our interpretability framework to modern architectures represents a promising path toward bridging mechanistic understanding with state-of-the-art sequence models.

## Acknowledgements

This research was funded by the German Research Foundation (DFG) within the Collaborative Research Centre 265 subprojects A06 and B08, the Federal Ministry of Research, Technology and Space (BMFTR), the Baden-Württemberg Ministry of Science as part of the Excellence Strategy of the German Federal and State Governments, and the DYNAMIC Centre, funded by the LOEWE program of the Hessian Ministry of Science and Arts (Grant Number: LOEWE 1/16/519/03/09.001(0009)/98).

## Reproducibility

Code for experiments and datasets is publicly available under https://github.com/ManuelBrenner/Linear_Nonlinear_Memory.

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

## A   Appendix

This appendix provides further methodological details and extended results. Sect. A.1 describes training procedures, regularization, encoders, and spike data modeling. Sect. A.2 covers analytical methods for bitcode analysis, stability, Lyapunov exponents, and flow fields. Sect. A.3 details all datasets. Sect. A.4 presents extended results across all tasks, including analysis of a stimulus selection task on rodent spike data (A.4.5).

### A.1   Methodological Details

#### A.1.1   Training Details

**Optimization and Hyperparameters**   The primary focus of this study was to investigate how nonlinearity affects performance. To ensure that our observations were not confounded by latent dimensionality, we used relatively high-dimensional AL-RNN models, scanning over several values of the latent size $M$ in the range of 10 to 100 (up to 128 for SCAN given the higher complexity of the task). This allowed us to operate in a regime where performance was not limited by the latent state capacity, and where a range of values for $P$ could be tested. We also conducted targeted sweeps over the manifold-attractor regularization strength $\tau \in \{0, 0.01, 0.1, 0.5, 1.0, 5.10\}$ (see Sect. A.1.2). In practice, $\tau = 0.1$ consistently yielded good results, and we therefore fixed this value across tasks to reflect that our results are not dependent on overly specific hyperparameter tuning. Importantly, the qualitative performance patterns across different levels of nonlinearity were robust to variations in both $M$ and $\tau$. The main hyperparameter of interest remained the number of nonlinear units $P$, which was systematically varied across experiments to assess its impact on task performance and dynamical properties. Hyperparameter scans for $P$ are reported directly in the results figures of the main text.

For optimization, we employed standard settings consistent with the original AL-RNN implementation (Brenner et al., 2024a), including initialization strategies from the official repository. We initialized the self-connection weights in $\boldsymbol{A}$ with small random values to prevent premature convergence to unstable dynamics, while the weights $\boldsymbol{W}$, the bias term $\boldsymbol{h}$, the input matrix $\boldsymbol{C}$ and readout matrix $\boldsymbol{D}$ (mapping to logit scores for classification) were sampled from a Gaussian distribution with zero mean and a standard deviation of 0.01. For optimization, we employed Adam (Kingma and Ba, 2017) with a learning rate of 0.001, which was further reduced during training by a cosine annealing learning rate scheduler. We used these settings for all datasets and did not perform fine-grained tuning.

During training, the model parameters were trained using standard backpropagation through time (BPTT). An independent validation set, comprising 10% of the training data, was used for model selection. For datasets with pre-existing train/test splits (e.g., sMNIST, Speech Commands, IMDb, SCAN), this was taken from the training set. For datasets without predefined splits, a validation set was artificially created. Other hyperparameters, such as batch size and learning rate scheduler, were set to conventional values that ensured stable convergence across all tasks.

**Hardware Usage**   All models presented in this study are relatively lightweight, using single-layer AL-RNNs with a maximum of 128 units. Due to this efficiency, all experiments could be conducted on a single CPU, with total training times for each model not exceeding 12 hours. Hence, the memory footprint remained well within the capacity of standard computing setups (less than 8GB RAM). The modest computational requirements ensure that our approach is easily replicable on conventional hardware.

#### A.1.2   Manifold Attractor Regularization

To promote the reconstruction of both fast and slow time scales - and associated memory - in the latent space, we regularize a subset $M_{\text{reg}} \leq M$ of the latent states, as suggested in Schmidt et al. (2021), according to:

$$\mathcal{L}_{\text{reg}} = \tau \left( \sum_{i=1}^{M_{\text{reg}}} (\tilde{A}_{ii} - 1)^2 + \sum_{i=1}^{M_{\text{reg}}} \sum_{j \neq i}^{M} (W_{ij})^2 + \sum_{i=1}^{M_{\text{reg}}} h_i^2 \right). \tag{3}$$

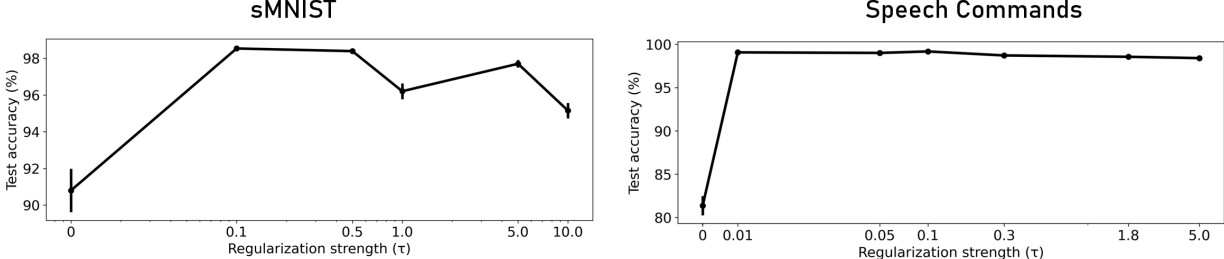

Figure 7: Manifold Attractor Regularization (MAR, Schmidt et al. (2021)) leads to a strong performance boost on sMNIST and Speech Commands Task, while not being overly sensitive to the exact value of the regluarization strength $\tau$.

Here, $\tilde{A}$ represents the matrix with the effective diagonal term for each unit: for linear units, the diagonal elements are derived directly from the $W$ matrix, as $A$ is zero in this case. For nonlinear units, the diagonal contribution is computed as the sum of their respective entries in $A$ and $W$. This ensures that the regularization correctly constrains self-connections to be near 1, irrespective of the unit's type. The regularization loss is added to the total loss with a scaling factor $\tau$ for each setting.

By constraining the self-connections of certain units to be near 1 and suppressing their cross-connections, these sub-units function as near-perfect integrators, capable of encoding arbitrarily slow time scales. Dynamically, this regularization encourages the formation of a continuous manifold of marginally stable fixed points, supporting long-term integration without requiring architectural modifications such as gating. It enables efficient memory retention even in simple RNN architectures, and has been shown to outperform LSTMs and other RNN architectures on long-range tasks (Schmidt et al., 2021). As recommended by Schmidt et al. (2021), we applied the regularization to half of the latent units $M$, hence primarily targeting the linear subspace.

### A.1.3 Encoders for Text, Audio, and Images

To incorporate domain-specific inductive biases, we used custom encoders for each modality, optimized for capturing the relevant temporal and structural patterns inherent in the data.

**IMDB** For the IMDB sentiment classification task, we used pre-trained $100D$ GloVe embeddings (Pennington et al., 2014) to initialize word vectors, followed by a multi-layer perceptron (MLP) as a nonlinear encoder. This additional nonlinear mapping allows the model to capture high-level semantic abstractions from raw word embeddings that aid with sentiment classification (Kim, 2014; Zhou et al., 2015) and enable the AL-RNN to focus on evidence integration, leading to a dynamically simple linear process.

**sMNIST** For the sMNIST task, we applied a series of stacked local 1D convolutional layers to the pixel sequences. These 1D convolutions respect the sequential nature of the input while extracting local stroke-based features that are beneficial for digit recognition.

**Speech Commands** For the Speech Commands dataset, we transformed the raw audio signals into Mel Frequency Cepstral Coefficients (MFCCs), a widely used feature representation for speech processing (Davis and Mermelstein, 1980). MFCCs provide a compact and noise-robust representation, emphasizing the perceptually relevant aspects of speech. This transformation also downsamples the sequence length based on the MFCC hyperparameters. In our setup, we select parameters that preserve moderate sequence lengths comparable to sMNIST (784 time steps). The transformed MFCC features are then passed through a series of stacked local 1D convolutional layers to extract temporal patterns. To ensure consistency across samples, the MFCC features are normalized using global mean and standard deviation, computed only over the training data.

### A.1.4   Training on Spike Data

To train on the neural spike data in Sect. A.4.5 and Appx. A.3, we embedded the AL-RNN within the Multimodal Teacher Forcing (MTF) framework (Brenner et al., 2024b), which enables training directly on non-Gaussian observations through likelihood-based optimization. Specifically, we employed a 1D temporal convolutional encoder to map observed spike trains to the latent space, followed by a Poisson decoder defined as

$$\log p(\boldsymbol{c}_t \mid \mathbf{z}_t) = \sum_i \left( c_{ti} \log \lambda_{ti} - \lambda_{ti} - \log(c_{ti}!) \right) \quad \text{with} \quad \boldsymbol{\lambda}_t = \exp(\boldsymbol{B}\boldsymbol{z}_t + \boldsymbol{b}),$$

where $\boldsymbol{z}_t$ is the latent state of the AL-RNN, $\lambda_{ti}$ the predicted spike rates, and $\boldsymbol{c}_t$ the observed counts. The MTF framework incorporates a latent consistency loss that penalizes the distance between inferred and predicted latent trajectories by

$$\mathcal{L}_{\text{consistency}} = \sum_t \| \boldsymbol{z}_t^{(\text{enc})} - \boldsymbol{z}_t^{(\text{gen})} \|^2,$$

where $\boldsymbol{z}_t^{(\text{enc})}$ is the latent trajectory inferred from the encoder and $\boldsymbol{z}_t^{(\text{gen})}$ the trajectory generated by the AL-RNN. Given the short duration of the spike train trials (30–40 time steps) and their high noise levels, we found that training without teacher forcing but relying solely on the consistency loss yielded the best results, likely because teacher forcing caused overfitting to individual spike events.

To account for drift in trial-specific spike statistics while preserving a shared task representation, we hierarchically parameterized the decoder following the approach in (Brenner et al., 2025), but applied this structure exclusively to the decoder parameters. Specifically, each trial was associated with a five-dimensional feature vector $\boldsymbol{l}^{(j)}$ (with its dimension determined by PCA, Fig. 23) that generated trial-specific decoder weights and biases via learned linear projections. The resulting hierarchical Poisson decoder was defined as

$$\log p(\boldsymbol{c}_t \mid \boldsymbol{z}_t, \boldsymbol{l}^{(j)}) = \sum_i \left( c_{ti} \log \lambda_{ti}^{(j)} - \lambda_{ti}^{(j)} - \log(c_{ti}!) \right), \quad \text{with} \quad \boldsymbol{\lambda}_t^{(j)} = \text{softplus}(\boldsymbol{B}^{(j)}\boldsymbol{z}_t + \boldsymbol{b}^{(j)}),$$

where $\boldsymbol{B}^{(j)}$ and $\boldsymbol{b}^{(j)}$ are decoder parameters generated as linear projections from $\boldsymbol{l}^{(j)}$.

This design choice ensures that the AL-RNN learns a consistent dynamic structure across trials. Hierarchizing the RNN itself could allow it to absorb task variation into the dynamics. By restricting flexibility to the output mapping, the AL-RNN remains responsible for solving the core disambiguation problem imposed by the context-dependent task, while the decoder accounts for slow drift or baseline shifts in firing statistics.

Across all trials, we observe a total of 783,840 time points. For the results presented, we use an AL-RNN with $M = 10$ latent dimensions, yielding 220 trainable parameters for the recurrent model. The hierarchical decoder includes $N_{\text{feat}} = 5$ learnable features per trial, which, combined with the shared linear projections, results in a total of 5,375 trainable parameters. Given the large number of observed data points relative to the model's capacity, this setup remains well within a regime where overfitting is unlikely.

### A.2   Analysis of Trained Models

**Bitcode Analysis**   To quantify the distribution of latent states across the linear subregions, we extract a subset of the PWL units from the full latent state vector. Let $\tilde{\boldsymbol{z}}_t \in \mathbb{R}^P$ denote the last $P$ components of the full latent state at time step $t$, corresponding to the PWL units. We represent their activation patterns as bitcodes, where each component is assigned a binary value based on its sign (1 if positive, 0 otherwise). Formally, the bitcode for $\tilde{\boldsymbol{z}}_t$ is defined as:

$$b_t = \sum_{i=1}^{P} \mathbb{I}[\tilde{z}_{t,i} > 0] \cdot 2^{P-i}, \tag{4}$$

where $\mathbb{I}[\cdot]$ is the indicator function. The empirical distribution of these bitcodes across a set of latent states is then given by:

$$p(b) = \frac{n(b)}{\sum_{b' \in B} n(b')}, \tag{5}$$

where $n(b)$ is the count of bitcode $b$ and $B$ is the set of all observed bitcodes.

**Stability Analysis**   In each linear subregion of the AL-RNN, its fixed points and their stability can be analytically computed. For a given bitcode, we first compute the masked recurrent weight matrix:

$$\boldsymbol{W}_{\text{masked}} = \boldsymbol{W} \odot \boldsymbol{M}(\text{bitcode}), \tag{6}$$

where $\boldsymbol{M}(\text{bitcode})$ applies masking to the connectivity matrix $W$, effectively zeroing out the outgoing connections of inactive nonlinear units.

Fixed points are then computed by solving the linear system:

$$(\boldsymbol{A} + \boldsymbol{W}_{\text{masked}} - \boldsymbol{I})\boldsymbol{z}^* = -\boldsymbol{h} \tag{7}$$

using a standard linear solver (implemented with `numpy.linalg.solve`). If the matrix is singular, no fixed point is considered valid. To assess the stability of each fixed point, we compute the eigenvalues of the Jacobian matrix:

$$\boldsymbol{J} = \boldsymbol{A} + \boldsymbol{W}_{\text{masked}}. \tag{8}$$

The fixed point is classified as stable if all eigenvalues $\lambda$ satisfy $|\lambda_i| < 1$, which were computed via the eigenvalue decomposition using `numpy.linalg.eigvals`.

**Maximum Lyapunov Exponent**   The maximum Lyapunov exponent, $\lambda_{\text{max}}$, characterizes the average rate of divergence of infinitesimally close trajectories in a DS. It is formally defined as:

$$\lambda_{\text{max}} = \lim_{T \to \infty} \frac{1}{T} \log \left\| \prod_{r=0}^{T-2} \boldsymbol{J}_{T-r} \right\| \tag{9}$$

where $\boldsymbol{J}_{T-r}$ represents the Jacobian of the system at time step $T - r$, $\|\|$ denotes the spectral norm, and the product of Jacobians accumulates the local expansion or contraction at each step. For the AL-RNN, these Jacobians are analytically tractable (Eq. 8) within each subregion. To approximate this exponent numerically, we evolved the trained model forward for $5,000$ time steps from a randomly sampled initial condition, discarding initial transients of $500$ time steps. Since the product of Jacobians grows exponentially in chaotic systems (Mikhaeil et al., 2022), we applied the algorithm outlined in (Vogt et al., 2022), which maintains numerical stability by re-orthogonalizing the Jacobian products at regular intervals via QR decomposition.

**Variance Analysis**   To quantify the spatial distribution of class manifolds in the AL-RNN latent space for the sMNIST task, we computed four metrics based on the final latent states of each digit class (Fig. 12). The final latent states were first normalized dimension-wise to account for scaling differences and then projected onto their PCs. The number of components was dynamically selected to capture at least $80\%$ of the total variance. We first evaluated the Coefficient of Variation (CV), which is defined as the ratio of the standard deviation to the mean of the class-specific variances $\boldsymbol{v}$: $\text{CV} = \frac{\sigma(\boldsymbol{v})}{\mu(\boldsymbol{v})}$. The CV provides a measure of relative dispersion, indicating how spread out the class-specific variance is in relation to its average. We further computed the Gini coefficient, which captures inequality in the distribution of variance across class manifolds, given by $G = \frac{\sum_{i=1}^{N} \sum_{j=1}^{N} |v_i - v_j|}{2N^2 \cdot \mu(\boldsymbol{v})}$. The Gini coefficient specifically quantifies how unevenly the variance is distributed among the different classes. Third, we calculated the Max-Min Ratio, defined as the ratio of the maximum to the minimum class variance, defined as $R_{\text{max / min}} = \frac{\max(\boldsymbol{v})}{\min(\boldsymbol{v})}$. This ratio highlights the degree of disparity between the most and least represented class manifolds, providing a direct measure of distribution extremes. Finally, we measured the Shannon Entropy of the variance distribution: $H(\boldsymbol{v}) = -\sum_{i=1}^{N} p_i \log(p_i)$, where $p_i$ represents the normalized variance for each class. Lower entropy values imply that variance is concentrated within a limited number of classes, while higher entropy suggests a more even spread across all class manifolds.

**Flow Field Approximation**   To approximate the continuous latent dynamics of the AL-RNN, we computed the flow field over a two-dimensional grid of points in the latent space. At each grid point, we applied the AL-RNN's step (Eq. 1) to estimate the local velocity vectors. This provides an approximation of the underlying continuous flow. For Fig. 20, we used a grid density of $30 \times 30$ points.

### A.3 Datasets

**IMDb**   The IMDb dataset (Maas et al., 2011) consists of 50,000 movie reviews, evenly split between training and test sets, with sentiment labels (positive or negative) for binary classification. It is available on Kaggle at https://www.kaggle.com/datasets/lakshmi25npathi/imdb-dataset-of-50k-movie-reviews. Each review was tokenized and converted to word vectors using pre-trained GloVe embeddings (Pennington et al., 2014). The sequences are truncated or padded to a fixed length of 128 tokens during training and testing. We did not apply any further text preprocessing beyond tokenization, as the GloVe representations handle most standard normalization. Training and evaluation follow the standard 25,000/25,000 train/test split.

**sMNIST**   The sequential MNIST (sMNIST) dataset transforms the standard MNIST (Lecun et al., 1998) digit images into sequences by flattening each $28 \times 28$ image into a 1D sequence of length 784. We used the dataset as provided in `torchvision.datasets` (maintainers and contributors, 2016). The pixel intensity values are normalized to the range $[0, 1]$ before being passed to the AL-RNN. The dataset is split into $60,000$ training sequences and $10,000$ test sequences, following the standard MNIST partitioning.

**Speech Commands**   The Speech Commands dataset (Warden, 2018) consists of 1-second audio recordings with a sampling rate of 16kHz of spoken words from a fixed vocabulary of commands: `["down", "go", "left", "no", "off", "on", "right", "stop", "up", "yes"]`. We used the version of the dataset provided by `TensorFlow Datasets` (TFD), which includes standardized preprocessing, splitting, and metadata.

**Copy Task**   The Copy Task is a synthetic memory benchmark where the model is required to store and reproduce a random sequence of discrete symbols after a delay period. Each input sequence consists of a random sequence of symbols represented in one-hot encoding, followed by a blank segment of the same length, and finally a cue signal indicating when to begin recall. The target output is an exact reproduction of the original symbol sequence during the recall phase. Sequence lengths are fixed during training. The training set comprised 1000 sequences, while the test set contained 200 sequences. Given that the task is defined over 4 symbols and 8 time steps, the total number of unique sequences is $4^8 = 65,536$. This represents a substantial combinatorial space, ensuring that the relatively small training set could not fully cover the distribution of possible sequences, thereby preventing the model from simply memorizing the training examples. Reported test accuracies reflect the percentage of correctly predicted symbols in the test sequences after the delay period.

**Addition Problem**   In the Addition Problem, popularized in (Hochreiter and Schmidhuber, 1997), the goal is to sum the values at the marked positions and output the result at the end of the sequence. This task tests the model's ability to integrate sparse, context-dependent information across long temporal spans. The two randomly chosen indices indicating the values to be summed are presented in the first half of the sequence. The AL-RNN is trained to produce the sum of the two digits at the final time step. The training set comprised 2000 examples, while the test set contained 200 examples. Reported test errors quantify the mean-squared error between the predicted and the correct sum.

**Contextual Multistability**   The Contextual Multistability task is a custom-designed integration task inspired by neuroscience paradigms of context-dependent decision-making (Mante et al., 2013). Each trial begins with a one-hot context cue, indicating which integration policy the model should adopt. A sequence of noisy evidence is presented across several time steps, and the model is required to integrate this evidence according to the initial context. At the final time step, a recall cue triggers the model to produce a decision based on the integrated evidence. If the initial context is inverted, the decision boundary is reversed. The AL-RNN is trained to predict the correct label based on its final latent state. We trained on 1000 example sequences, and tested on 200 sequences.

**Prefrontal Cortex Task Recordings**   The CRCNS PFC-1 dataset (Rodgers and DeWeese, 2014), includes single-unit recordings from medial prefrontal cortex and auditory cortex of rats performing a context-dependent stimulus selection task. We focused on the longest available session (Day 4) from the first rat (CR12B), comprising approximately 900 trials. For this rat, only auditory cortex (AC) neurons were recorded. Each

trial involves auditory stimulus comprising a pitch component (high or low warble, presented bilaterally) and a spatial cue (broadband noise from left or right). The correct response—"go" or "no-go"—depends on the current task rule: either localization (respond based on side) or pitch discrimination (respond based on pitch). The task is structured in repeating blocks. The task was structured into repeating blocks of four trial types that defined the task stage.

1. Block 1: spatial rule with spatial-only stimuli (left vs. right broadband noise),

2. Block 2: spatial rule with compound stimuli (both spatial and pitch cues)

3. Block 3: pitch rule with pitch-only stimuli (low vs. high warble, bilaterally presented)

4. Block 4: pitch rule with compound stimuli

We aligned each trial from one second before stimulus onset to the decision point (center-port withdrawal for go trials). No-go trials lacked an explicit decision timestamp, so we truncated them to match the duration distribution of go trials, avoiding length-based confounds between trial types. Spike times were binned at 50ms resolution, resulting in spike count vectors for each neuron over the trial window.

External inputs were encoded as one-hot vectors: two dimensions for the spatial cue (left/right), two for pitch (low/high), and four for the current block (task stage). To mimic the contextual cue structure used in memory tasks, task stage information was only provided during the first 100 ms of the trial, reflecting the fact that rats had learned the current rule through block structure and early trial cues. Only trials with correct responses were retained to ensure that task-trained and spike responses were consistent with each other.

**SCAN Task**   The SCAN dataset (Lake and Baroni, 2018) is a synthetic benchmark designed to evaluate systematic compositional generalization in sequence-to-sequence learning. In SCAN, natural language commands are translated into sequences of low-level actions in a navigation environment. For example, the command "jump twice" is mapped to the action sequence "JUMP JUMP", while a more complex command like "walk around right" is interpreted as "RTURN WALK RTURN WALK RTURN WALK RTURN WALK". The mapping is deterministic and unambiguous, with each command generated by a phrase-structure grammar that defines valid language-action mappings. The commands are constructed from a small set of primitives:

- **Primitive Actions**: walk, run, jump, look.

- **Directional Modifiers**: left, right, opposite.

- **Frequency Modifiers**: twice, thrice.

- **Compositional Operators**: and, after.

We specifically use the *Simple Split* of the SCAN dataset, where 80% of all command-action pairs are randomly assigned to the training set, while the remaining 20% are reserved for evaluation. The primary challenge is to generalize compositional patterns from seen instructions to novel ones. For instance, if the model learns "walk" and "jump", along with the modifier "twice", it should generalize to "jump twice" even if that specific combination was not observed during training. We follow the canonical implementation of SCAN for preprocessing and evaluation, where commands are tokenized into one-hot vectors. During training, input sequences are encoded with the encoder AL-RNN and predicted using the decoder AL-RNN, which runs freely from the initial state provided by the encoder AL-RNN. Reported test accuracies quantify the amount of correctly predicted test sequences.

**Multi-Task Training**   We jointly trained AL-RNNs on 11 cognitive benchmark tasks adapted from Driscoll et al. (2024) presented in randomly interleaved trials. The task suite includes pro- and anti-response variants of delayed response (remember a stimulus direction across a delay, then respond in the same or opposite direction), reaction time (immediately respond in the same or opposite direction as a presented stimulus), and category decision (classify stimulus angle as above or below threshold, then respond with the

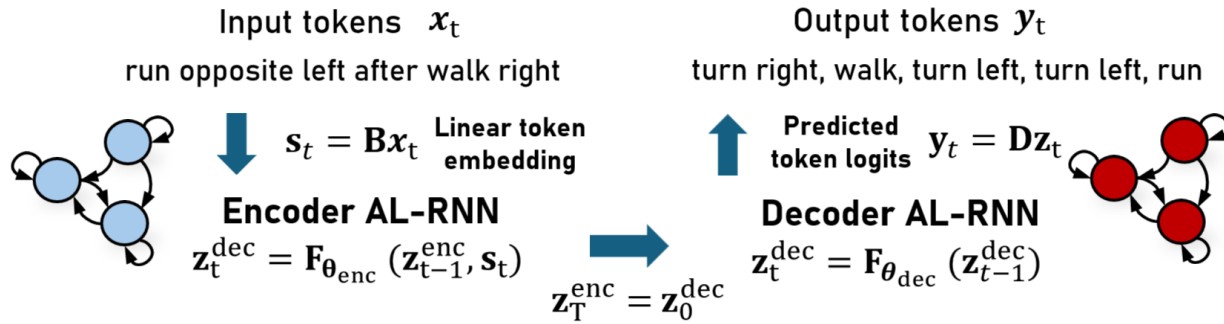

Figure 8: Outline of the the encoder-decoder architecture used for solving the SCAN task.

corresponding or reversed category mapping). We also included match and non-match versions of delayed match-to-sample, where the network must compare two stimuli separated by delays and respond based on whether they match within a 45° tolerance. Two context integration tasks require selective attention to one of two simultaneously presented sensory modalities while integrating noisy evidence across sequential presentations. Finally, a go/nogo task tests simple threshold detection, requiring a response when stimulus amplitude exceeds 0.5 and fixation maintenance otherwise. Together, these tasks probe working memory, response inhibition, selective attention, and context-dependent computation while sharing common temporal structure and input encodings that encourage discovery of reusable computational strategies.

### A.4 Further Results

Table 2: Comparison of different models and activations across tasks.

| Model / Activation | Addition MSE | Copy Acc. | sMNIST Acc. | SpeechCmd Acc. |
|---|---|---|---|---|
| AL-RNN (GELU) | $2 \pm 1 \cdot 10^{-4}$ | $96 \pm 3\%$ | $98.2 \pm 0.2\%$ | $99.2 \pm 0.2\%$ |
| AL-RNN (ReLU) | $3 \pm 1.5 \cdot 10^{-4}$ | $92 \pm 3\%$ | $98.2 \pm 0.2\%$ | $99.1 \pm 0.2\%$ |
| AL-RNN (tanh) | $9 \pm 3 \cdot 10^{-4}$ | $90 \pm 3\%$ | $98.0 \pm 0.2\%$ | $99.0 \pm 0.2\%$ |
| AL-RNN (hardtanh) | $4 \pm 5 \cdot 10^{-3}$ | $88 \pm 6\%$ | $97.9 \pm 0.3\%$ | $99.0 \pm 0.1\%$ |
| LSTM | $8.8 \pm 10 \cdot 10^{-4}$ | $54 \pm 12\%$ | $98.5 \pm 0.2\%$ | $99.1 \pm 0.1\%$ |
| GRU | $6.3 \pm 10 \cdot 10^{-4}$ | $69 \pm 7\%$ | $98.0 \pm 0.2\%$ | $99.0 \pm 0.1\%$ |

Table 3: Performance across tasks for different activation types and sparsity levels.

| Model | Copy Acc. | Addition MSE | Contextual Integration Acc. |
|---|---|---|---|
| Linear ($P = 0/50$) | $58 \pm 3\%$ | $0.15 \pm 0.01$ | $\approx 50\%$ |
| Sparse (GeLU, $P = 3/50$) | $96 \pm 3\%$ | $2 \pm 1 \cdot 10^{-4}$ | $\approx 95\%$ |
| Sparse (ReLU, $P = 3/50$) | $92 \pm 3\%$ | $3 \pm 1.5 \cdot 10^{-4}$ | $\approx 95\%$ |
| Sparse (tanh, $P = 3/50$) | $90 \pm 3\%$ | $9 \pm 3 \cdot 10^{-4}$ | $\approx 95\%$ |
| Sparse (hardtanh, $P = 3/50$) | $88 \pm 3\%$ | $4 \pm 5 \cdot 10^{-3}$ | $\approx 95\%$ |
| Full (GeLU, $P = 50/50$) | $38 \pm 6\%$ | $6 \pm 2 \cdot 10^{-4}$ | $\approx 95\%$ |
| Full (ReLU, $P = 50/50$) | $39 \pm 5\%$ | $7 \pm 3 \cdot 10^{-4}$ | $\approx 95\%$ |
| Full (tanh, $P = 50/50$) | $58 \pm 6\%$ | $3.6 \pm 2 \cdot 10^{-3}$ | $\approx 95\%$ |
| Full (hardtanh, $P = 50/50$) | $46 \pm 4\%$ | $6 \pm 5 \cdot 10^{-3}$ | $\approx 95\%$ |

#### A.4.1 Impossibility of Solving the Addition Problem with Linear RNNs

**Proposition 1.** *An AL-RNN without ReLU nonlinearity cannot solve the addition problem.*

*Proof.* Consider a linear RNN where $\phi(z) = z$, yielding the dynamics

$$z_t = \tilde{A}z_{t-1} + Cs_t + h, \tag{10}$$

where $\tilde{A} = A + W$ and $s_t = (s_{1,t}, s_{2,t})^\top$. The output at time $T$ is given by

$$y_T = Bz_T = B\left(\tilde{A}^T z_0 + \sum_{t=1}^{T} \tilde{A}^{T-t}(Cs_t + h)\right). \tag{11}$$

This can be rewritten as a linear functional of all inputs:

$$y_T = \sum_{t=1}^{T} (\alpha_t s_{1,t} + \beta_t s_{2,t}) + \gamma, \tag{12}$$

where $\alpha_t = B\tilde{A}^{T-t}C_{:,1}$, $\beta_t = B\tilde{A}^{T-t}C_{:,2}$, and $\gamma$ includes contributions from $z_0$ and $h$. Crucially, the coefficients $\alpha_t$ and $\beta_t$ are fixed functions of time, independent of the input data.

For the addition problem, the target output must satisfy $y_T = s_{1,t_1} + s_{1,t_2}$, where $t_1$ and $t_2$ are the (random) time points at which $s_{2,t_1} = s_{2,t_2} = 1$. This requires the effective coefficient of $s_{1,t}$ in the output to be

$$\alpha_t^{\text{eff}} = \begin{cases} 1 & \text{if } s_{2,t} = 1 \\ 0 & \text{if } s_{2,t} = 0 \end{cases}. \tag{13}$$

However, in the linear system, $\alpha_t$ is fixed and cannot depend on $s_{2,t}$. Since $t_1$ and $t_2$ are randomly positioned across trials, no fixed sequence $\{\alpha_t\}_{t=1}^{T}$ can satisfy $\alpha_{t_1} = \alpha_{t_2} = 1$ and $\alpha_t = 0$ for $t \notin \{t_1, t_2\}$ for all trials simultaneously. Therefore, a linear PLRNN cannot implement the conditional integration required by the addition problem, as it lacks the ability to gate the contribution of $s_{1,t}$ based on the value of $s_{2,t}$. □

### A.4.2 Classification Tasks

**sMNIST** In the sMNIST task, $28 \times 28$ digit images are flattened into a sequential time series of length 784, with one pixel input per time step. Classification is based solely on the final latent state. Bitcode analysis provides a clear mechanism by which nonlinearity increases performance: each class aligns with its own set of closely neighboring bitcodes. For instance, for $P = 50$, the average variation in the bitcode within each class was just $0.06 \pm 0.02$ (see Fig. 10, 0.5 indicates chance level), meaning that on average 48 out of 50 PWL units had the same sign at the final state. Between classes, PWL units use different bitcodes. We observed that switching between these codes during sequential processing is aligned with the onset of visually informative stroke patterns (see Fig. 11 for example illustrations). This allows the AL-RNN to implement class-specific linear integration dynamics: while all digit classes are processed through slow modes with maximum eigenvalues close to 1 (Fig. 9**c**), subtle class-dependent differences in these eigenvalues, arising from the distinct linear subregions occupied by each class, implement differentiated integration rates. Since the input projection itself is linear (via the C matrix), it is the autonomous dynamics within each subregion that reshape the latent manifold: local flows with slightly divergent eigenvalues drive trajectories for different digit classes apart in latent space, resulting in well-separated clusters that facilitate discrimination without requiring the input integration mechanism itself to be nonlinear (Figs. 2**f** & 12).

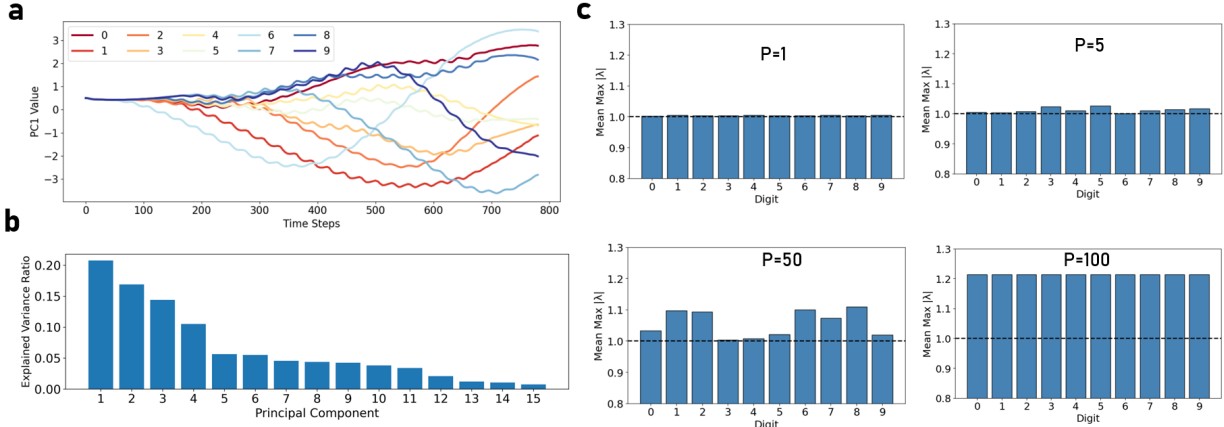

Figure 9: **a**: Mean latent trajectories of the first PC for each digit class in the *sMNIST task*. The trajectories reveal a slow accumulation of evidence along the primary axis, similar to sentiment classification (Fig. 2**c**). Interestingly, distinct branching patterns are visible: the digit '6' diverges first, reflecting its characteristic structure in the initial image segments. Digits '8' and '9' share overlapping paths until around the midway point, aligning with their visual similarity in the upper halves of the images. **b**: Explained variance ratio of the first 15 PCs indicates a more complex structure compared to the 2$D$ sentiment attractor in Fig. 2. **(c)** Maximum eigenvalue analysis across digit classes for models with varying nonlinearity. Linear and minimally nonlinear models ($P = 1, 5$) maintain maximum eigenvalues consistently near 1.0 across all digits, indicating that a uniform slow mode is shared across all classes. Models with moderate nonlinearity ($P = 50$) preserve eigenvalues close to unity but show class-specific variations, demonstrating that nonlinearity enables subtle, digit-specific modulation of the slow manifold while maintaining the overall slow integration character. Fully nonlinear models ($P = 100$) exhibit systematically larger eigenvalues across all classes, providing a mechanistic explanation for the overdispersion and reduced performance observed at high $P$.

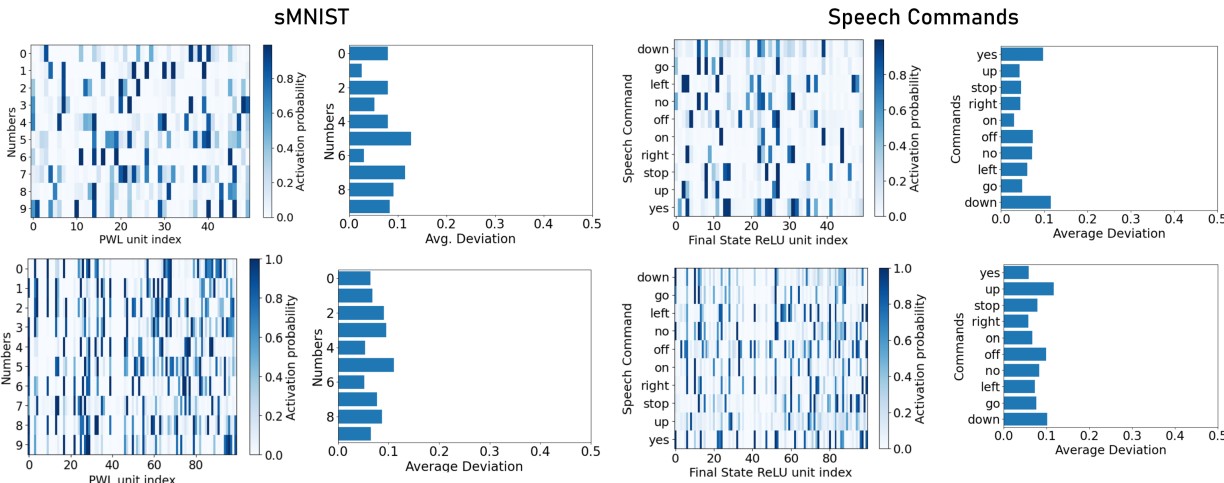

Figure 10: Analysis of class-specific bitcode activation for the *sMNIST task* (left) and Speech Commands (right). Left subpanel: Average bitcode activation probability per class for the final states $z_T$ for two representative models of $P = 50$ (top) and $P = 100$ (bottom). High and low probabilities indicate that bitcode activations are consistent within a class. Right subpanel: Average within-class deviation from these predominant activation patterns (rows in the left figure). Notably, comparing $P = 50$ (bottom) to the fully nonlinear model $P = 100$ (top), the within-class variability of bitcode activations is reduced. This reduction in variation implies more stable and distinct bitcode representations, enhancing the robustness and reliability of the model's classification outcomes.

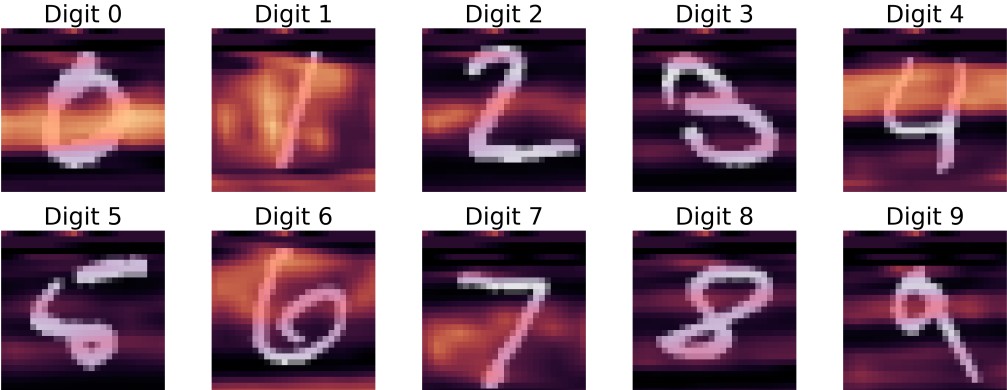

Figure 11: Analysis of bitcode activations for the *sMNIST task*. Each panel shows the average Hamming distance from the reference state $(0, 0, 0)$ for a model with $P = 3$ bitcode activations, aligned across time for all 10 digits. The PWL units switch primarily during visually informative stroke segments, effectively gating nonlinear processing. Digits with sharp vertical onsets (e.g., "1", "4" and "6") display early bitcode activations, whereas digits with round strokes (e.g., "0", "8") activate later. Digits with similar overall shapes (e.g., "8" vs. "9") diverge in their bitcode representation toward the end of the sequence. These qualitative differences illustrate how the different linear subregions of the AL-RNN support the dynamic discrimination of the different digits.

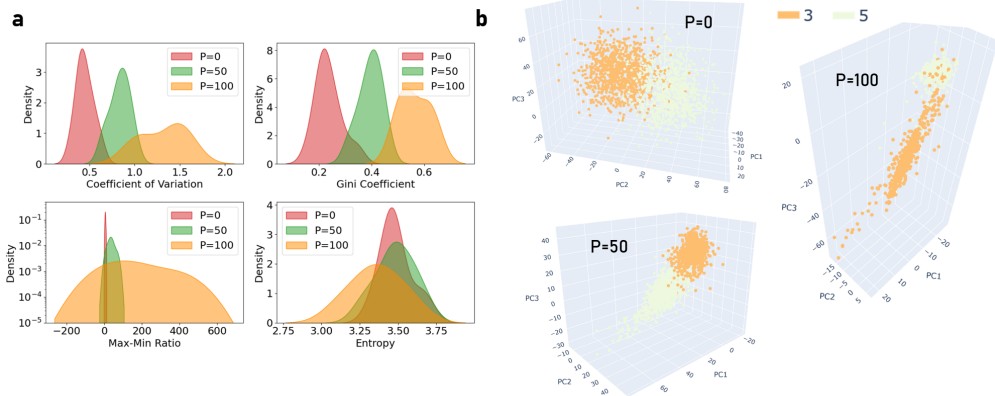

Figure 12: Analysis of class manifolds for the *sMNIST task*. (**a**) Kernel Density Estimates of key metrics across nonlinear configurations ($P = 0, 50, 100$). The fully linear model ($P = 0$) exhibits tightly packed, uniform class variance (low Gini coefficients and Max-Min ratios). Moderate nonlinearity ($P = 50$) achieves optimal balance with $CV \approx 1$, producing spatially distinct manifolds. The fully nonlinear model ($P = 100$) shows high Gini coefficients and wide Max-Min ratios, indicating uneven variance distribution where some manifolds are over-compressed while others are spread out. (**b**) Final latent states projected onto the first three PCs for frequently confused digits 3 (orange) and 5 (green). At $P = 50$, manifolds are spatially distinct and well-separated. At $P = 100$, they become elongated and entangled, with digit 5 particularly over-compressed, directly correlating with observed misclassification patterns.

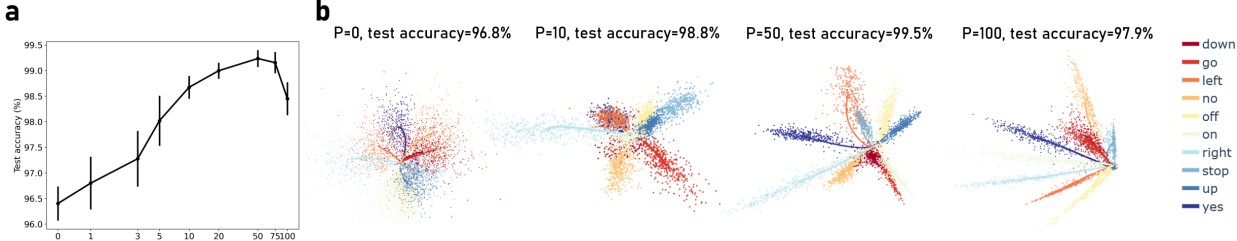

Figure 13: Same figure as Fig. 2 but for the *Speech Commands task*. **a**: Accuracy plotted against number of nonlinear units. Increasing nonlinearity leads to moderate increases, while performance deteriorates for fully nonlinear models. **b**: Final latent states projected onto the first 3 PCs highlight how nonlinearity partitions the latent space according to class labels.

**Speech Commands**   To verify that the trends observed in sMNIST generalize across modalities, we evaluated the AL-RNN on the Speech Commands dataset (Warden, 2018), a ten-class audio classification task using spoken commands sampled at 16 kHz. We used the same evaluation setup as for sMNIST, with classification based on the final latent state after processing the input sequence. As in the visual domain, we observed that increasing nonlinearity improved classification accuracy up to an intermediate number of nonlinear units, after which performance declined for fully nonlinear models (Fig. 13**a**). Latent states from models with moderate nonlinearity again exhibited clear class-specific clustering in PC space (Fig. 13**b**), and bitcode alignment within classes remained high (Fig. 10)—closely mirroring the gating behavior observed in sMNIST (Sect. A.4.2). These results confirm that the latent integration and partitioning mechanisms learned by the AL-RNN generalize robustly across visual and auditory modalities.

### A.4.3   Additional Results Addition and Copy Task

**Autonomous Dynamics in the Copy Task**   To understand how the flow field implements these computations, we further examined the autonomous dynamics of the system. These establish a global k-cycle with a precise period of 100 time steps (exactly half the delay phase duration), as confirmed by a neutral maximum Lyapunov exponent ($\lambda_{max} \approx 0$, Fig. 3**b**, bottom, see Appx. A.2 for methodological details), for which trajectories neither diverge exponentially nor converge to a fixed point. While the initial transient dynamics are constrained entirely to the lower subregion, the fully developed k-cycle spans both subregions, oscillating between them in a structured manner. This cycle is stabilized through the interplay of two different dominant eigenvalues of the local Jacobians: one with a contracting virtual fixed point ($\lambda_{max} \approx 0.992$) and the other around a diverging fixed point ($\lambda_{max} \approx 1.005$) which drives the initial transient dynamics. Intuitively, once the trajectory has settled onto the orbit that alternates between those two subregions (as visible in Fig. 3**b**, bottom, where the PWL unit repeatedly crosses the switching boundary), the net effect over a full cycle is neutral expansion $\lambda_{max} \approx 0$), which stabilizes the limit cycle.

**Copy Task With Variable Delay**   In the main text, we analyzed the copy task under fixed delays between encoding and recall. We additionally implemented a variable delay version of the task, in which the cue to initiate recall occurred after a randomly chosen interval. In this setting, solutions discovered under fixed delays did not transfer successfully, but required a different strategy for solving the task. However, consistent with the fixed delay task, we find that successful AL-RNNs still maintain latent activity within a single linear subregion throughout the delay period, indicating that storage continues to rely on stable linear dynamics. However, the decoding mechanism is substantially more complex: during the recall phase, trajectories now transition across multiple linear subregions, rather than remaining confined to one. Analysis of the resulting bitcodes shows that symbol identity is reflected in the sequence of subregion switches, with decoding accuracy from the bitcode alone reaching around 60% (compared to a 25% chance level) (Fig. 15**b**). These results indicate that variable timing shifts the computational burden from storage to decoding. Whereas fixed-delay tasks can be solved by maintaining activity in a single regime and reactivating stored content at a fixed time, the variable-delay setting requires a more flexible, state-dependent decoding process.

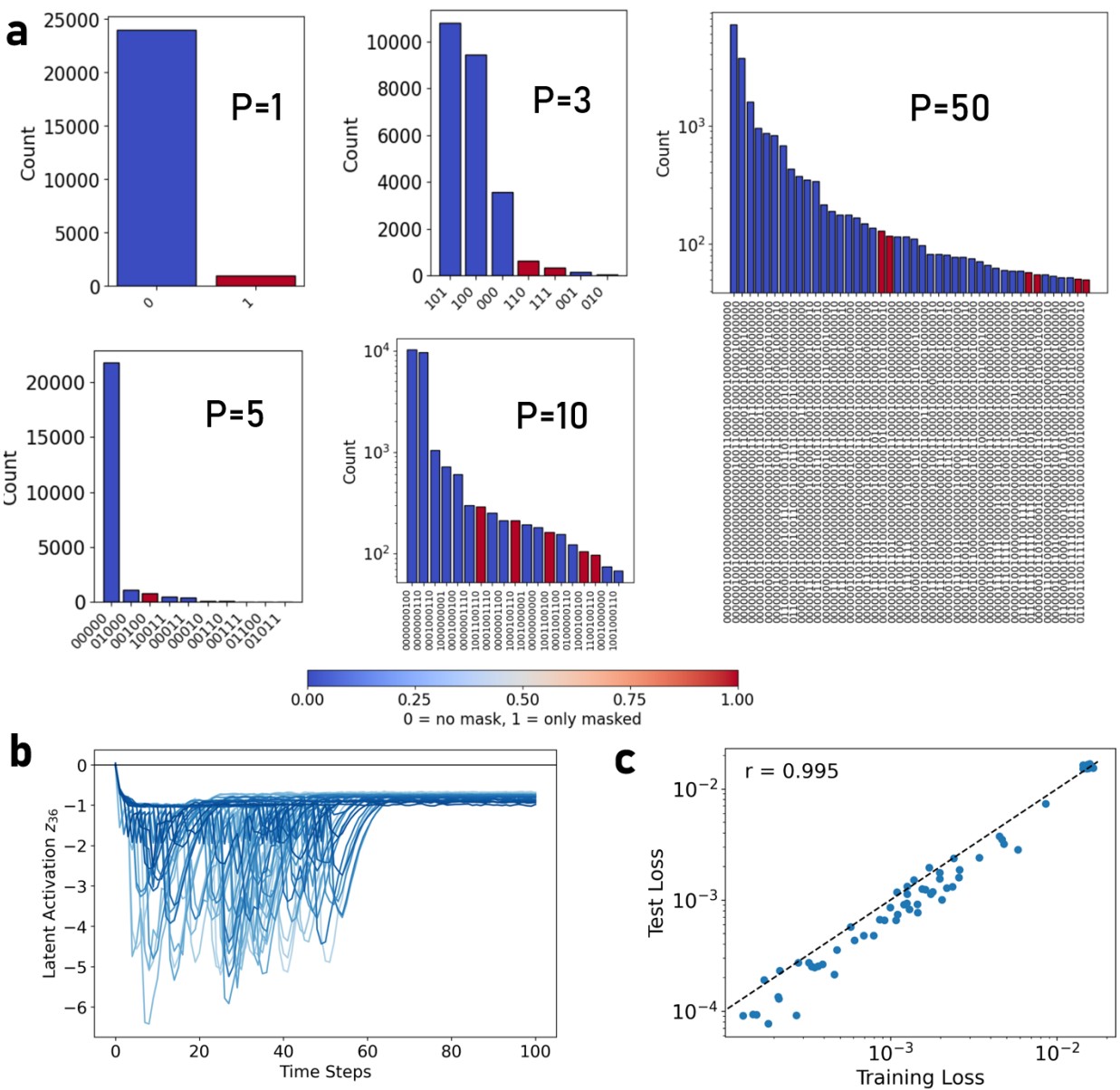

Figure 14: **a**: Distribution of bitcode subregions in the *addition task* for different levels of nonlinearity ($P$) for the best-performing models. A small subset of dominant subregions (blue) encode linear integration dynamics, while sparse gating is implemented by a few low-frequency subregions (red). The relative ratio of subregion activity for masked time points shows clear separation between integration and gating functions, with no region implementing both. While more nonlinear settings ($P = 10, 50$) distribute integration dynamics across multiple neighboring regions, this could introduce instability compared to the compact representations with sparse nonlinearity, potentially explaining performance decreases in fully nonlinear models. Note that for $P = 10 \& 50$, the y-axis is scaled logarithmically. **b**: Networks often "linearize" certain units, pushing individual activations far from the piecewise-linear boundary such that only a small subset effectively switches. This inductive bias simplifies the optimization landscape by operating in the linear regime where possible, which may explain why fully nonlinear models are less robust. **c**: Correlation between training and test losses across runs ($r = 0.995$). Each point represents a trained model; the dashed line indicates $y = x$. Near-perfect correlation with slightly higher test losses rules out classical overfitting.

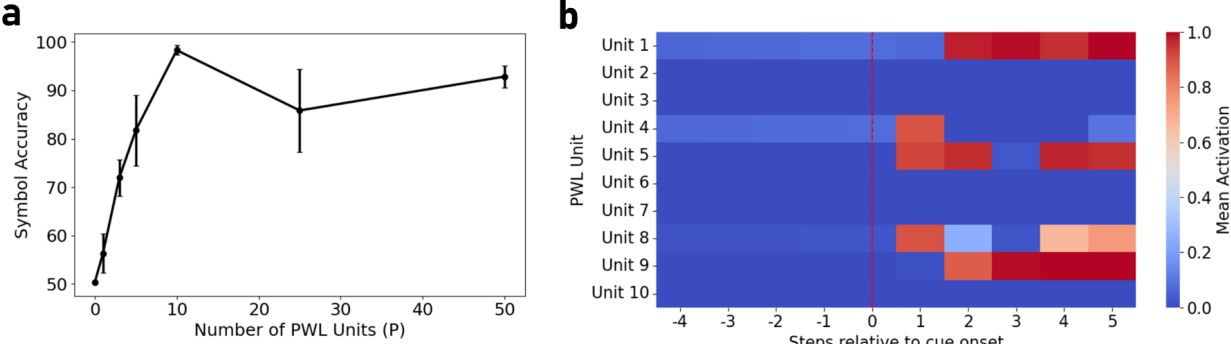

Figure 15: Results for the *variable delay copy task*. **a**: Symbol recall accuracy as a function of the number of PWL units P. Performance rises steeply with sparse nonlinearity and peaks at intermediate values of P. Error bars denote standard deviation across seeds. **b**: Mean activation of individual PWL units aligned to cue onset (dashed red line). During the delay period (steps $< 0$), activity remains confined to a largely linear regime. After cue onset, recall requires structured transitions across multiple subregions, reflected in distinct activation patterns across units and time steps.

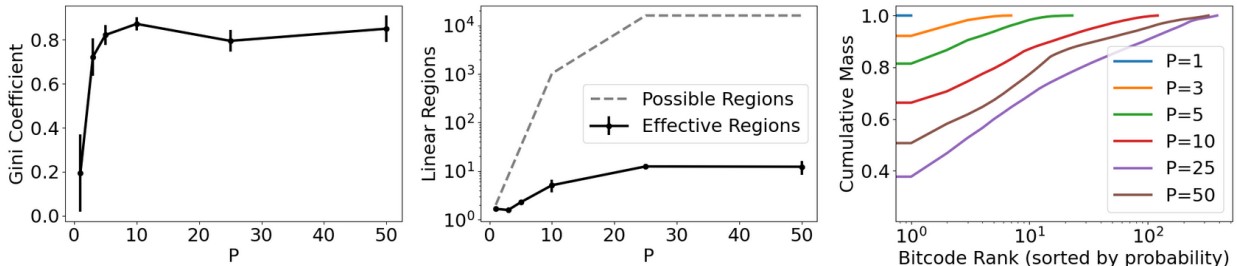

Figure 16: Distribution of linear subregions during decoding stage for the *copy task*. **(a)** The Gini coefficient of bitcode distributions as a function of $P$, reflecting the imbalance in subregion usage. As $P$ increases, the Gini coefficient rapidly rises, indicating that only a limited subset of regions is dominantly occupied. **(b)** The effective number of linear regions occupied by the AL-RNN, plotted alongside the theoretical maximum (dashed gray line) computed as $\min(2^P, \text{total samples})$. Despite the potential to explore exponentially many regions, the network only populates a small fraction. **(c)** Cumulative mass plot for the bitcode distributions sorted by probability, visualized across different values of $P$.

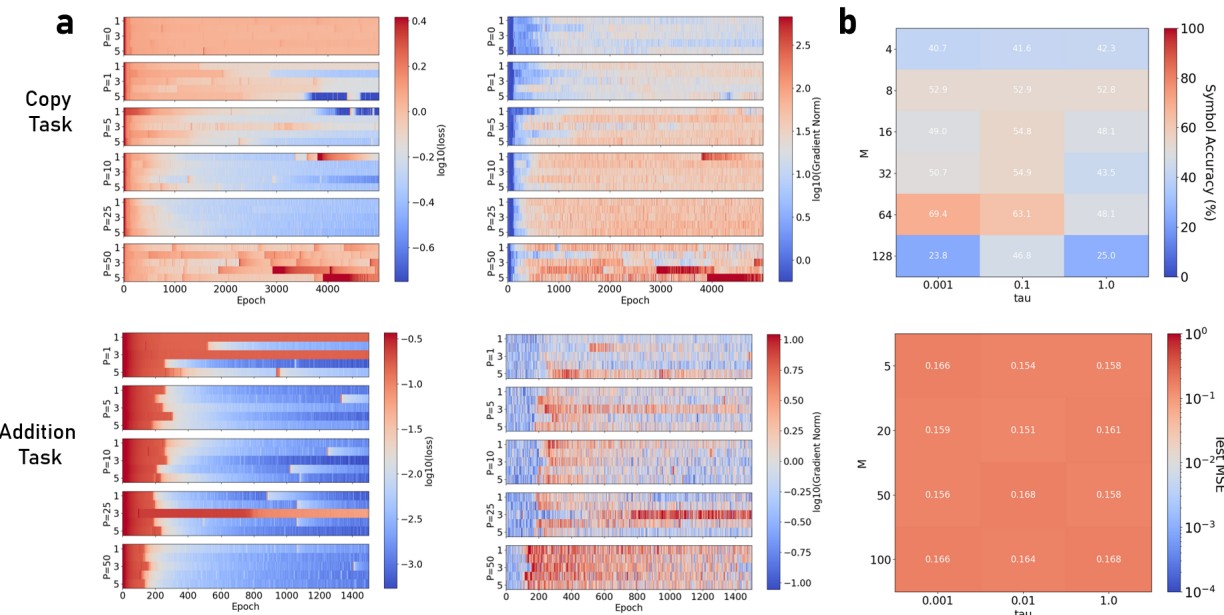

Figure 17: Training dynamics and performance of AL-RNNs on *copy task* and *addition task*. **(a)** Loss (left) and gradient norm (right) trajectories for models with varying numbers of $P$ PWL units out of 50 (see y-axis labels). *Copy task (top):* Linear models ($P = 0$) show smooth training but fail to reach low loss values. With sparse nonlinearity ($P = 1, 5$), some models achieve good performance with relatively smooth training, though bifurcations begin to appear. Fully nonlinear models ($P = 50$) exhibit highly irregular training with frequent bifurcations and consistently elevated gradient norms. *Addition task (bottom):* Loss drops faster for models with more nonlinearity, likely since higher model capacity means solutions are discovered more rapidly, followed by fine-tuning. Models with minimal nonlinearity ($P = 1$) often fail to find solutions, possibly because fewer nonlinear units reduce the probability of stumbling upon the correct mechanism. Intermediate configurations ($P = 10, 25$) still show training instabilities where losses suddenly jump up. Fully nonlinear models converge fastest but maintain higher gradient norms throughout. **(b)** Test accuracy for purely linear networks across latent dimensions $M$ and regularization strengths $\tau$. Linear networks completely fail on the addition task, performing at chance level across all settings. On the copy task, accuracy ranges from $40 - 70\%$ with apparent random fluctuation across hyperparameters, never approaching optimal performance. This demonstrates that the results from the main text are no overly sensitive to hyperparameter search.

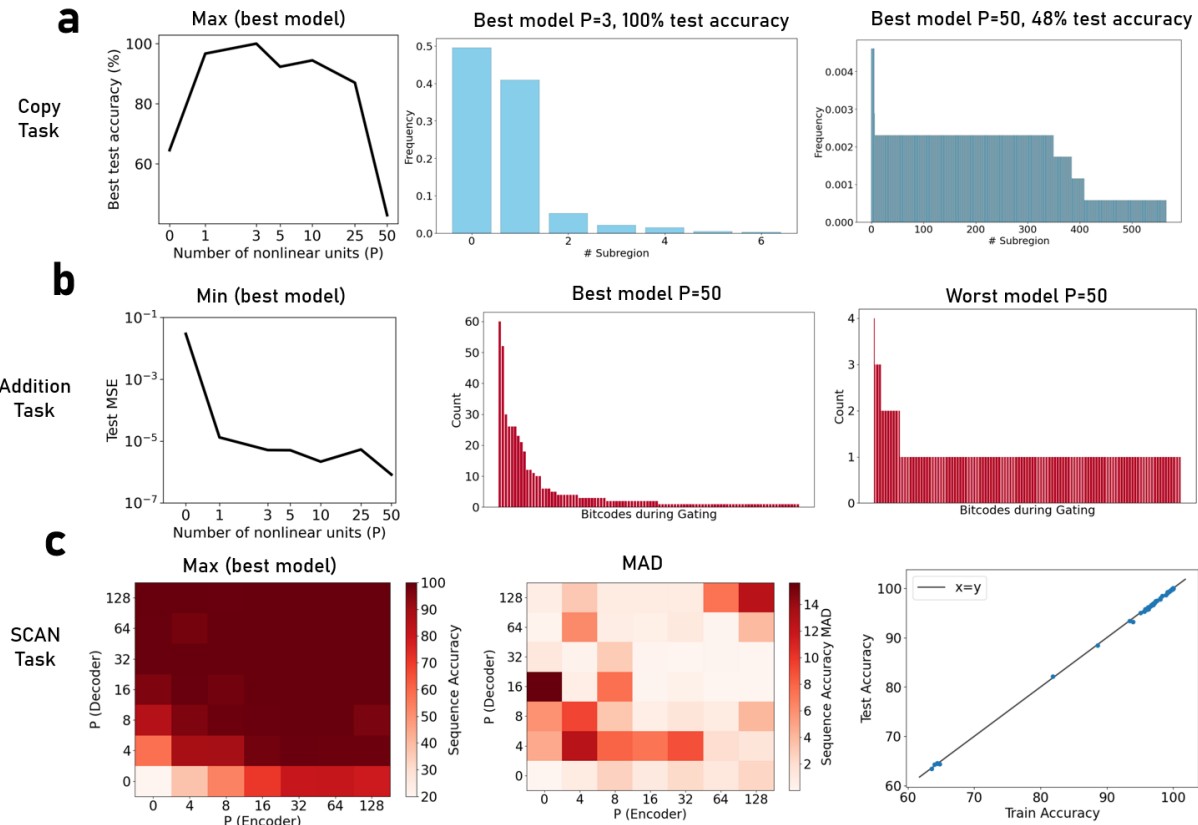

Figure 18: Performance and bitcode distributions on *copy task*, *addition task*, and *SCAN task*. **(a)** Copy task performance across nonlinearity levels $P$. Best model accuracy (left) peaks at intermediate $P = 3$, reaching 100% accuracy. Subregion usage for the best $P = 3$ model (middle, 100% accuracy) concentrates in a small number of regions, while the best $P = 50$ model (right, 48% accuracy) fragments activity across hundreds of subregions. **(b)** Addition task test MSE. The best individual model is fully nonlinear ($P = 50$, left), while median performance was slightly worse with full nonlinearity. Comparison of the the best $P = 50$ model (middle) vs. worst performing models (right) shows that the well-performing model concentrates gating in few subregions with a strongly skewed distributions, while the worst model spreads activity across many subregions. **(c)** SCAN task accuracy. Best models (left) achieve near-perfect performance across all $P$ values, including $P = 128$ (fully nonlinear). However, mean absolute deviation (MAD, middle) increases substantially for fully nonlinear models, indicating higher training variability. Train-test accuracy correlation (right) shows this variance is not due to overfitting but reflects training sensitivity. Fully nonlinear models can still find optimal solutions but do so less reliably.

### A.4.4 Nonlinearity Enables Internal Task Switching Through Context-Dependent Routing

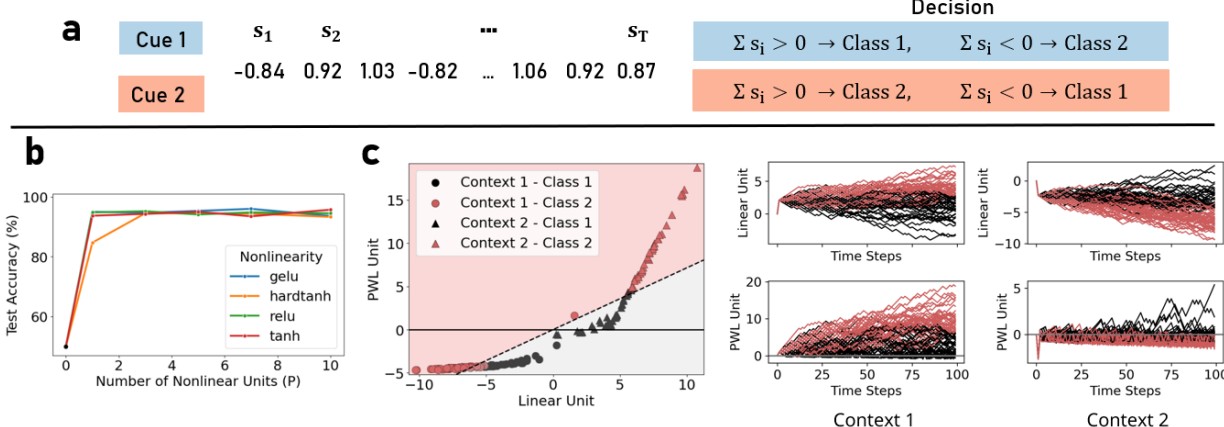

Figure 19: **a**: Task schematic: Depending on the initial context cue (blue or red), the network must classify the sign of the integrated input sequence with a reversed decision rule. **b**: Test accuracy as a function of nonlinear units, $P$. **c**: Final latent states for both contexts with a linear decision boundary (dotted line, red and grey regions indicate classes). Right: Latent trajectories over time (black for Class 1, red for Class 2) cumulatively integrate evidence. The initial context determines the activation of the PWL unit, selecting the linear subspace: the PWL unit is positive in Context 1 and mostly negative in Context 2, inverting the temporal integration process to enable linear separability.

We assess the network's ability to switch between tasks using a *context-dependent integration task* (Fig. 19**a**), a design inspired by behavioral paradigms in neuroscience for studying flexible decision-making (Mante et al., 2013). Each trial begins with a one-hot context cue, followed by a sequence of scalar inputs ($T_{seq} = 100$). This task requires the network to switch between two internal decision policies based on a contextual cue. Purely linear AL-RNNs cannot solve this task, and plateau at 50% accuracy (Fig. 19**b**), independent of how many linear units are provided. In contrast, the task is almost perfectly solvable with just one linear and one PWL unit, achieving up to 96% accuracy.

We illustrate the central mechanism with one successfully trained model with one linear and one PWL unit. Here, the PWL unit primarily operates in two distinct subregions, based on the initial cue (Fig. 19**c**). In Context 1, the nonlinear unit essentially mimics linear integration, residing within a single subregion for the entire sequence, while the linear unit accumulates the input symmetrically. In Context 2, the nonlinear unit transitions to a different subregion, effectively inverting the sign of evidence accumulation within the linear unit, achieving the required task switching, leading to linear separability of the final states (Fig. 19**c**, right). This mechanism exemplifies context-dependent routing: the nonlinear unit acts as a switch, directing identical inputs through distinct internal linear dynamics depending on the initial context. See Fig. 20 for flow field and example trajectories.

### A.4.5 Nonlinearity Enables Task Switching in a Stimulus Selection Task in Joint Task–Neural Modeling

We next evaluated a structurally similar setting, this time training the AL-RNN not only on the task but also to jointly reconstruct neural population activity from the CRCNS PFC-1 dataset (Rodgers and DeWeese, 2014). This dataset records single-unit activity in rodent auditory cortex during a flexible stimulus-selection task, in which animals alternate between two rules: choosing the correct response based on either the spatial origin or the pitch of the sound (Fig. 21**a**). The two stimulus dimensions are presented simultaneously but independently, with either a low or high warble (pitch cue) played on both sides, and a pure tone (side cue) played either on the left or right. Depending on the current rule, the animal must attend to either the pitch (low = "go", high = "no-go") or the side (left = "go", right = "no-go") of the auditory input. This setup creates contextual ambiguity: for example, an identical stimulus combination of a high-pitched warble and a tone on the left requires different responses ("go" under the spatial rule, "no-go" under the pitch rule). Hence

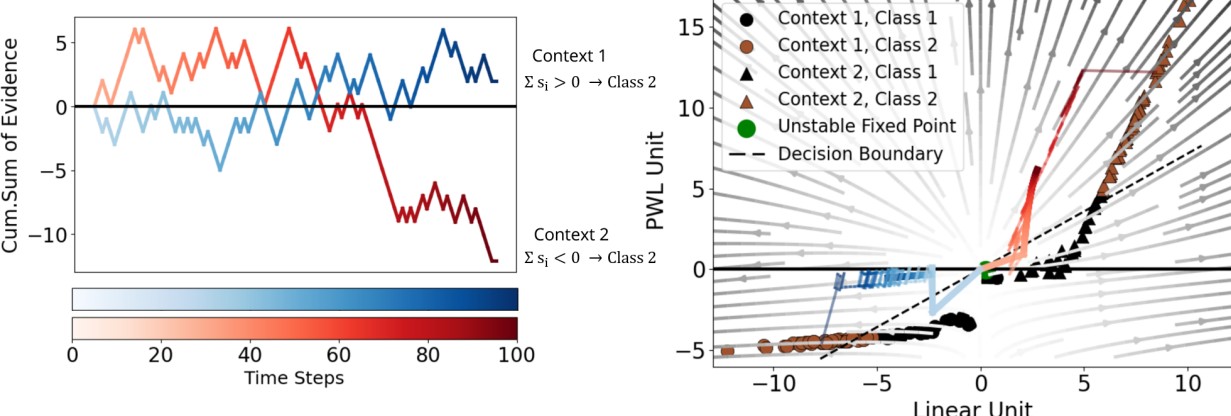

Figure 20: Flow field and latent trajectories in the linear and nonlinear subspaces for both contexts (Context 1: blue, Context 2: red) in the *contextual integration task*. The left panel depicts the cumulative sum of evidence for two example sequences, with context inverting the decision rule. The right panel shows the vector field, where thick lines represent paths taken when no input is provided except the initial context cue, gradually drifting towards the decision boundary in the respective context. Thinner lines illustrate the two example trajectories under continuous input (corresponding to the cumulative sum of evidence shown in the left panel). In these cases, evidence integration drives the trajectories along context-specific pathways—positive accumulation in Context 1 and inverted accumulation in Context 2. An unstable fixed point (green dot) near the origin demarcates the boundary between the two decision regions.

the task studies linear and nonlinear neural computations in context-dependent behavior. Preprocessing steps and selection criteria are detailed in the Appx. A.3. To ensure a consistent task representation, we restricted training and evaluation to the 723 trials in which the animal made the correct choice.

**Task-training the AL-RNN**  We first trained an AL-RNN to solve the task based purely on structured temporal input, independent of neural data. Each trial mimicked one of the 723 empirically observed trials, beginning with a brief context cue indicating the active rule (spatial or pitch), presented for 100 ms (two time steps), followed by a one-second delay. Then, an auditory stimulus was presented (one-hot encoded for pitch and side), and the model was required to produce a binary decision (go/no-go), which was read out linearly from the latent state at the empirically observed decision time $z_{t_{dec}}$ (see Appendix A.3). Despite the task's contextual nature, we find that a 2D model is already often sufficient to linearly separate identical stimuli based on their preceding rule cue, and linear 3D models often solved the task perfectly. Systematic evaluation confirmed that increasing capacity beyond this dimension does not yield substantial gains in task accuracy alone (Fig. 21**c**, left), and that errors were largely restricted to contextually ambiguous stimuli (e.g., Right–Low under the spatial rule; Fig. 21**c**, right).

**Task Information Improves Spike Reconstruction**  We next used the same AL-RNN to jointly model both the animal's behavior and the neural spike trains. That is, the network received identical inputs as in the task-only training—context cue, delay, and stimulus—and was required not only to produce the correct decision at the appropriate time, but also to reproduce the observed spiking activity across neurons *from the same latent states of the AL-RNN* (Eq. 1). To achieve this, we trained the AL-RNN within the Multimodal Teacher Forcing (MTF) framework (Brenner et al., 2024b), which allows direct training on non-Gaussian spike data via a Poisson decoder. The decoder is hierarchically conditioned on a 5-dimensional, trial-specific feature vector (Brenner et al., 2025), enabling it to capture slow drift in spike statistics while preserving a shared task representation in the AL-RNN (see Appx. A.1.4 for training details and Fig. 23 for an analysis of the learned features).

We found that including task information markedly improved spike reconstruction (Fig. 21**d**), suggesting a strong alignment between latent decision dynamics and observed neural variability. Quantitatively, spike rates

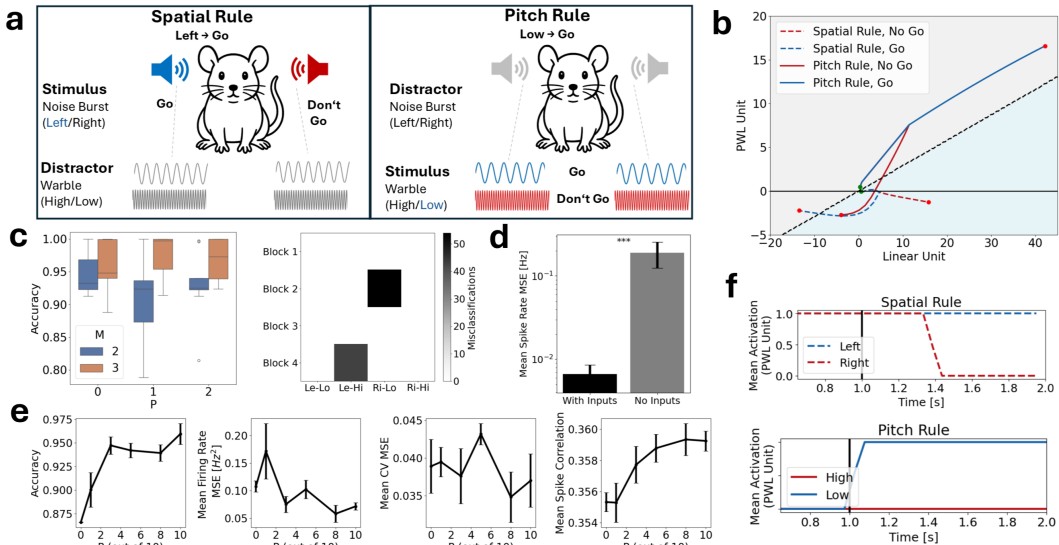

Figure 21: **a**: Schematic of the contextual stimulus selection task used in the PFC-1 dataset (Rodgers and DeWeese, 2014). In each trial, animals responded to auditory stimuli based on either the spatial origin (left/right) or the pitch (high/low), depending on the current task rule. **b**: Latent trajectories from a task-trained 2D model for two identical stimuli under different rules, illustrating context-dependent separation at decision time by the subregion boundary (black horizontal line). **c**: Task performance across models of increasing nonlinearity (P) and different latent dimensions (M) (left) and misclassification counts by stimulus and block (right). **d**: Mean firing rate MSE with and without inputs. **e**: Joint training performance when modeling both task and spike data: test accuracy (left), mean spike rate MSE (middle left), and coefficient of variation (CV) MSE (middle right) and mean spike correlation (right) as a function of the number of nonlinear units. **f**: Mean activation over time of a representative nonlinear unit that differentiates rules and modulates stimulus-driven activity conditionally.

and coefficients of variation were well captured (see Fig. 22**a**). To evaluate the model's temporal reconstruction, we correlated recorded and generated spike trains, yielding a mean correlation of $r = 0.36 \pm 0.002$. This approached the correlation between separate Poisson samples drawn from the same latent trajectory ($r = 0.371 \pm 0.057$), which serves as an approximate upper bound given the high intrinsic variability of sparse cortical spiking.

**Nonlinearity Supports Integration In Joint Modeling** In the joint setting, nonlinearity began to play a more significant role. Increasing the number of nonlinear units improves both task accuracy and spike reconstruction ($r \approx -0.37, p = 0.005$, low firing rate MSE/higher task accuracy are better, Fig. 21**e**). In contrast, linear models plateau around 87% accuracy and exhibit an almost identical error pattern: they consistently respond "go" to all trials with left or low cues, regardless of the active rule, thus failing to resolve the contextual ambiguity (see Fig. 21**c**, right). In many such nonlinear models, we observed the emergence of single units that effectively gated the task rule, becoming active in one rule context and inactive in the other, with stimulus integration occurring only within the active subregion (Fig. 21**f**). Quantitatively, we examined the linear subregion active at the time the input is first presented (after one second) and compared which subregions were active in blocks 2 and 4 (the two conditions where both spatial and pitch cues are present but different rules are active; see Appx. A.3). In successful models ($> 99\%$ accuracy), trials from these blocks were routed into almost entirely distinct subregions (on average, 97% of trials fell into block-specific linear subregions), while in unsuccessful models this separation was much less pronounced (only 32%). Task accuracy was strongly correlated with this separation score (ratio of trials falling into non-overlapping subregions for different rules, $r = 0.66$). This shows a clear link between successful models leveraging nonlinearity to assign different task rules to distinct linear regions, allowing them to react differently to identical stimuli depending on context. This mechanism is illustrated for a successful task-trained 2D model in Fig. 21**b**.

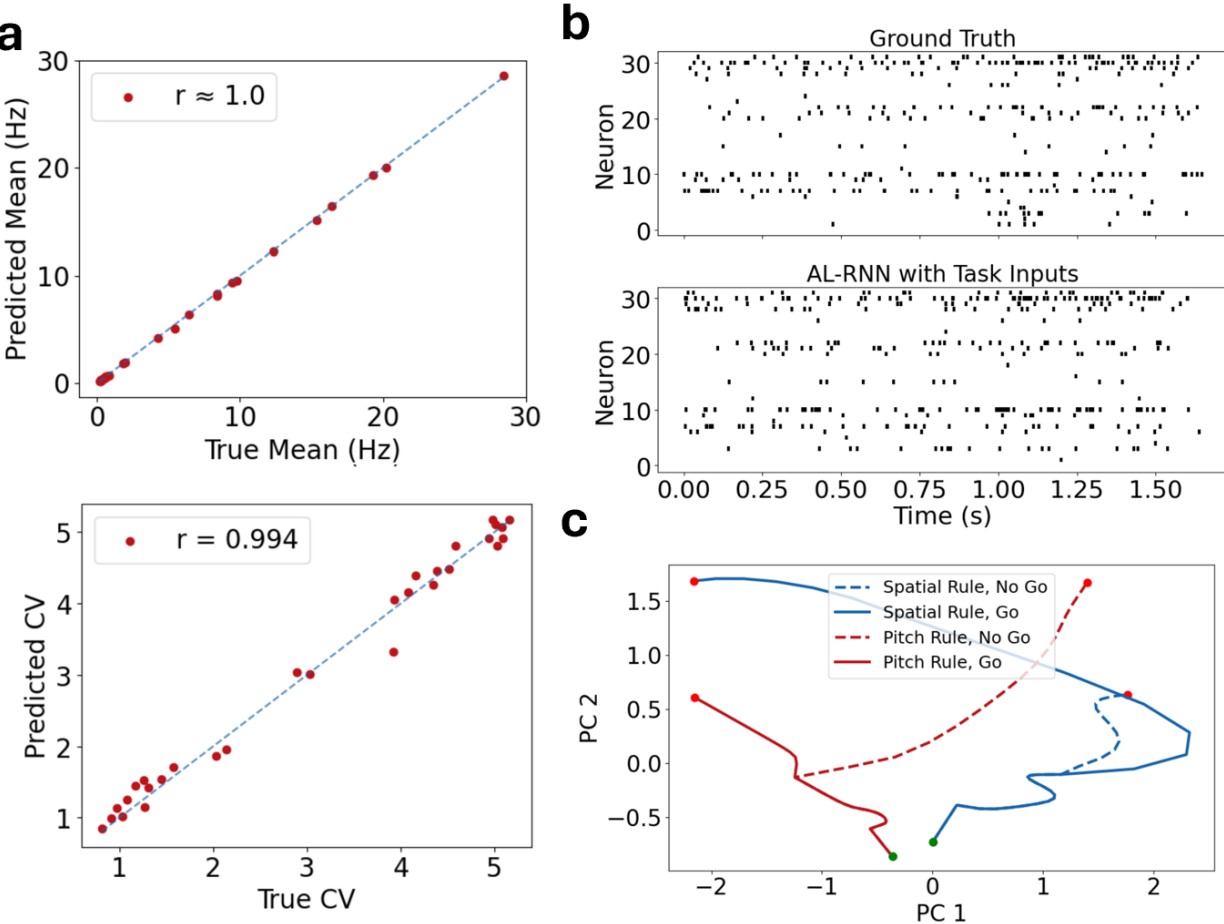

Figure 22: **a** Predicted vs. true mean firing rate (top) and coefficient of variation (CV; bottom) across all neurons for an example model trained with task inputs ($M = 10, P = 3$). The model accurately captures both first- and second-order spike statistics. **b** Example raster plot comparing true and reconstructed spike trains for a single trial for the same model ($M = 10, P = 3$). **c** Example PCA-projected latent trajectories for the four ambiguous input conditions from Fig. 21, showing clear separation based on context.

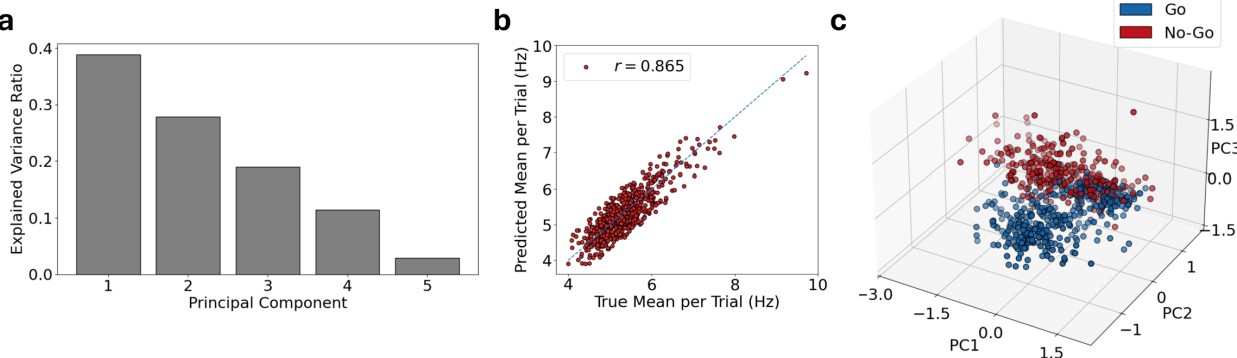

Figure 23: **a**: Explained variance of the five PCs of the learned 5D trial-specific feature vectors, used to generate trial-specific decoder parameters. **b**: Predicted vs. true trial-averaged firing rates for all neurons. **c**: PCA projection of the feature vectors into the first three PCs, colored by Go (blue) and No-Go (red) labels. Trial-level task information is partially encoded in the decoder feature space; a linear classifier trained on the first three components achieves 86% accuracy.

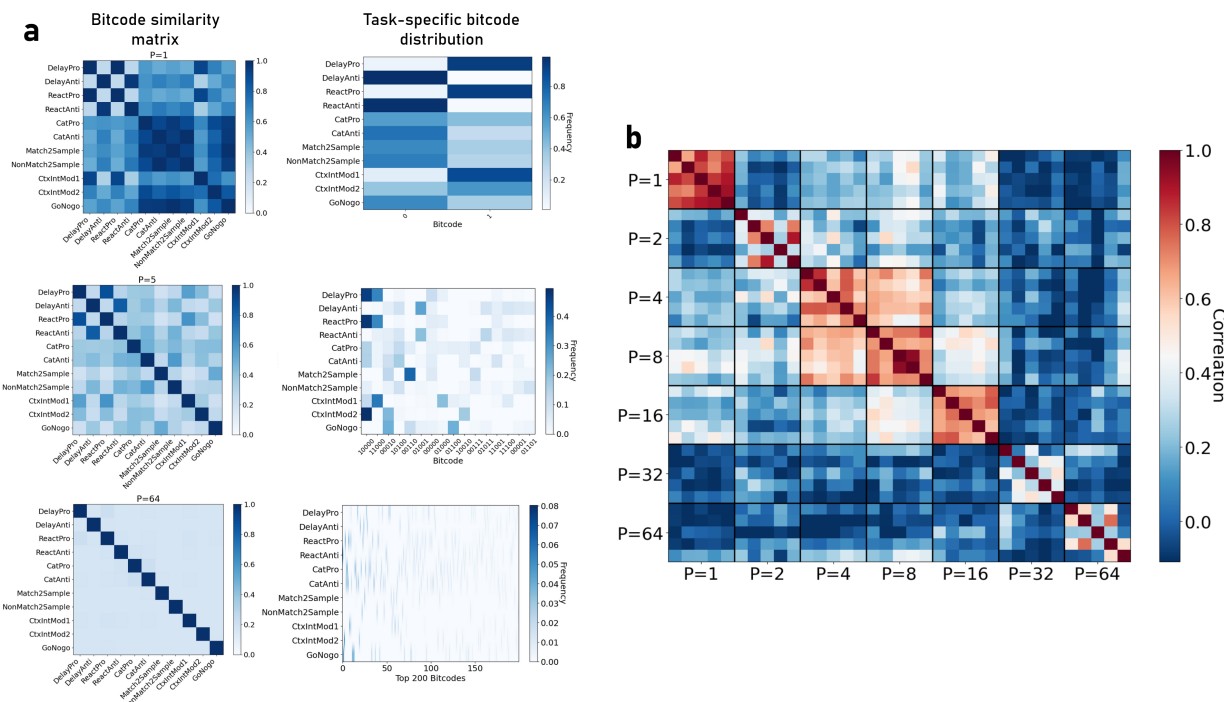

Figure 24: Bitcode usage from AL-RNN training on a *multi-task* paradigm. **a**: *Left:* Bitcode similarity matrices computed from Jensen-Shannon divergence between task-specific subregion distributions, as in Fig. 5. *Right:* Task-specific bitcode usage distributions corresponding to each $P$ level. For $P = 1$, only 2 possible bitcodes exist (representing the two sides of a single PWL boundary), and tasks cleanly partition between them, e.g., DelayPro and DelayAnti use opposite subregions, consistent with their opposite but individually linear task requirements (see also Fig. 25 for example latent trajectories). At $P = 5$, with $2^5 = 32$ possible bitcodes, distributions broaden but remain concentrated on small subsets, with some tasks showing distinct peaks (e.g., Match2Sample, CtxIntMod1/2) while others overlap. At $P = 64$, with $2^{64}$ possible bitcodes, distributions become extremely sparse and diffuse across the top 200 most-used regions, and tasks show almost no shared structure. **b**: Consistency of learned representations across runs with 20 samples per task via correlation of task similarity matrices (from **a**) across 5 independent training runs. Sparse nonlinearity ($P = 4 - 16$) produces highly robust subregion assignments. Representations also remain similar across adjacent sparsity levels (e.g., $P = 4$ to $P = 8$ shows correlations $\sim 0.5 - 0.7$), indicating that in the low data limit, the shared nonlinear structure is robustly captured.

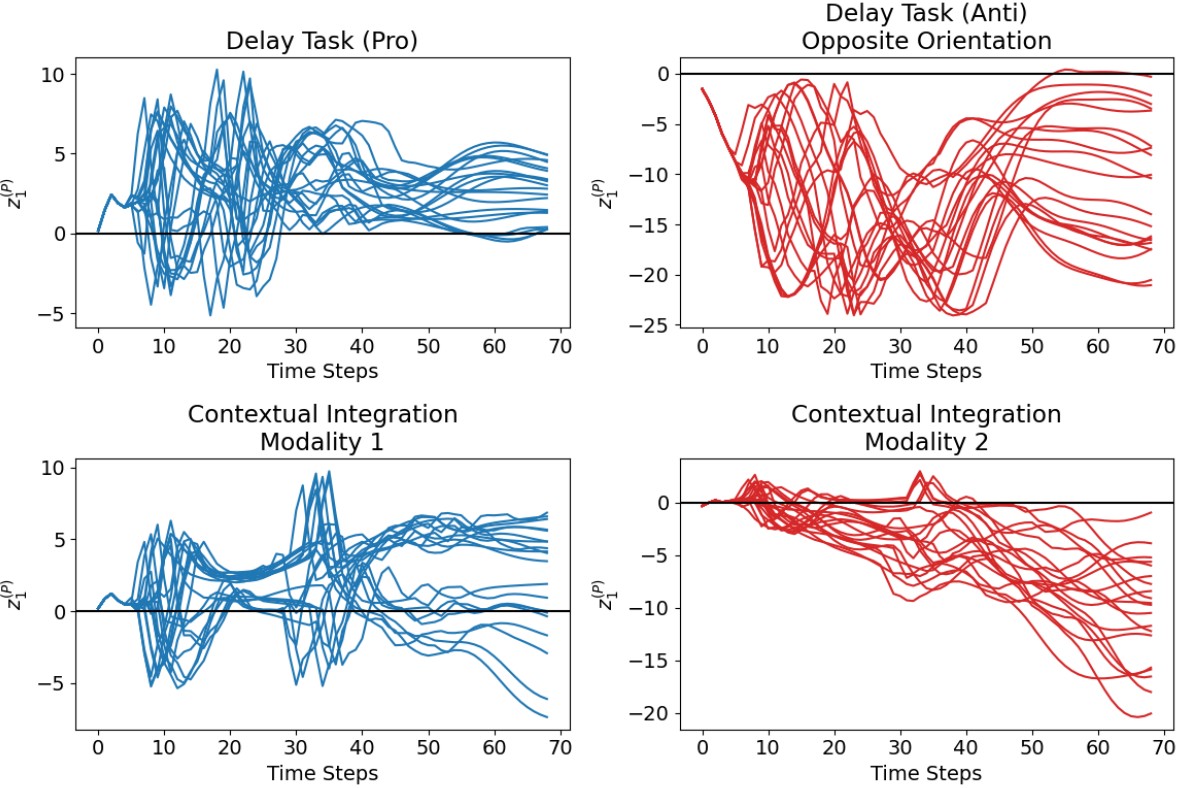

Figure 25: Latent trajectories of an AL-RNN with $P = 1$ trained on the multi-task suite for the pro and anti versions of the delay task, and the two modalities in the contextual integration task. The network leverages the single PWL boundary (black horizontal line) to separate the task mechanism for the two different task versions.

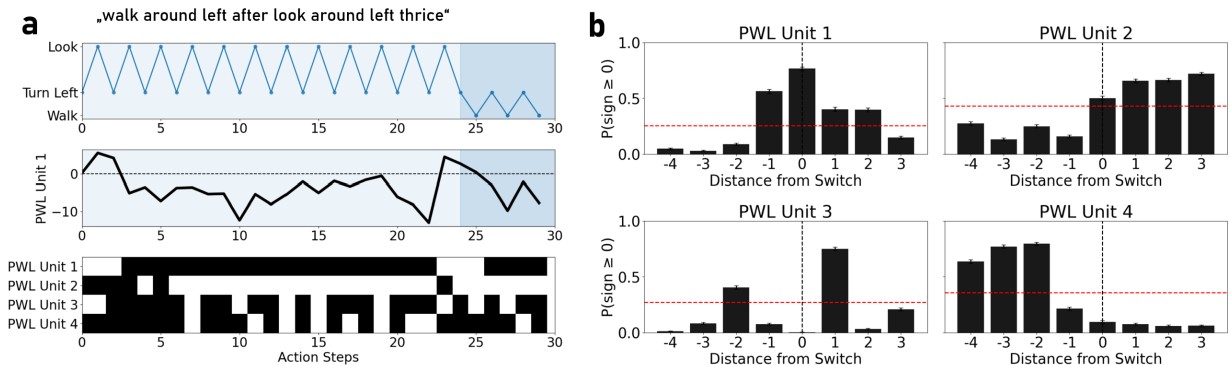

Figure 26: Latent Dynamics of PWL Units during composite actions in the *SCAN task.* **a**: Visualization of an example SCAN command *walk around right thrice after jump around left thrice.* Top: Executed actions over time, illustrating the switching behavior between distinct subtasks (look, turn left, walk). Middle: Activation of PWL Unit 1 throughout the sequence, highlighting a characteristic sign flip from negative to positive precisely before the transition point which triggers the switch to the next subtask (shaded background region). Bottom: Bitcode representation for the four PWL units, where black denotes negative activation and white represents positive activation. Notably, PWL Unit 1 consistently flips to positive at the switch, marking the boundary between subtasks. **b**: Statistical analysis of PWL unit activations around switch points between composite subtasks, averaged across all SCAN sequences. The x-axis represents time steps relative to the switch point ($t = 0$). Bars indicate the average probability of positive activation for each PWL unit (mean $\pm$ sem). The dashed red line is the global baseline activation for each unit. All PWL units showcase characteristic shifts in their sign around switch points: Unit 1 initiates switches with a brief peak to positive activity, Unit 2 transitions from predominantly negative to predominantly positive activation (and vice versa for Unit 4), and PWL Unit 3 synchronizes its oscillations tightly with subtask boundaries.

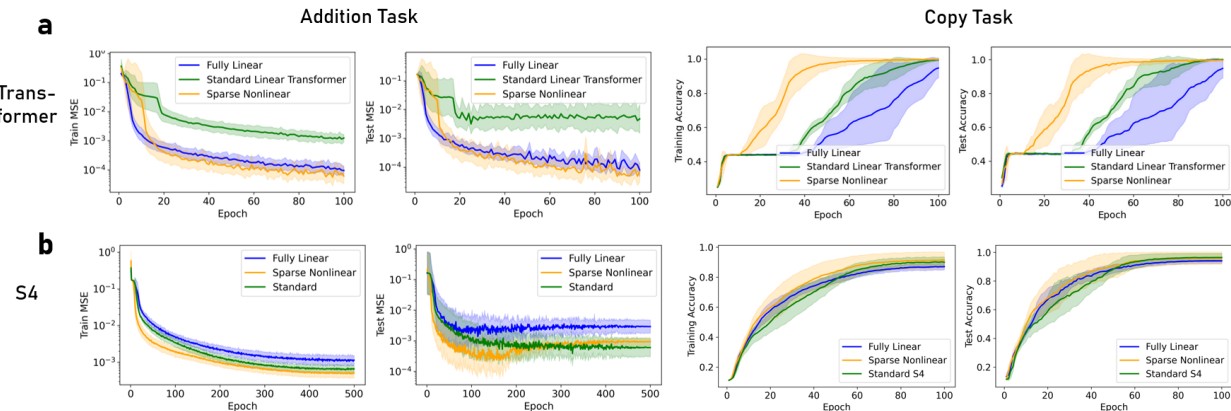

Figure 27: Comparison of three architectural variants for both Linear Transformer (Katharopoulos et al., 2020) and S4 (Gu et al., 2022) models: *Fully Linear* models contain no nonlinear activations anywhere in the architecture, *Standard* models include nonlinearity in every layer (ReLU in feedforward networks for Transformers, GELU in every S4D block for S4); and *Sparse Nonlinear* models only use nonlinearity at the final layer after all attention or state-space blocks, keeping intermediate computations linear. Note that the optimization is not directly comparable to the training of the AL-RNN. While we implemented causal versions of these architectures to maintain meaningful task structure, even with causal masking, these models fundamentally differ from first-order Markovian RNNs, with their multi-layer structure provides a view of the sequence that enables linear operations to solve tasks like addition (which our proof shows is impossible for purely linear Markovian models). **(a)** Linear Transformer performance. On the addition task (left), standard Transformers with distributed nonlinearity converge slower and converge at higher error than both fully linear and sparse nonlinear variants, which perform comparably. Test performance closely mirrors training, with standard Transformers reach $\sim 10^{-3}$ MSE while others achieving near $10^{-4}$. On the copy task (right), sparse nonlinear models substantially outperform both alternatives, reaching near-perfect accuracy ($>95\%$) on both train and test sets. Standard Transformers show intermediate performance, while fully linear models converge much slower and exhibit high variance. **(b)** S4 model performance. For the addition task (left), all variants converge to similar final performance ($\sim 10^{-3}$ MSE) with, with the nonlinear variants performing slightly better. On the copy task (right), all three variants achieve near-perfect performance, though sparse nonlinear models converge slightly faster than the alternatives. Overall, the architecture-dependent benefits of sparse nonlinearity observed in RNNs partially transfer to attention-based models but are less pronounced in state-space models for the two considered tasks.

