# OpenReview forum: "Uncovering the Computational Roles of Nonlinearity in Sequence Modeling Using Almost-Linear RNNs"
_TMLR — Accepted by TMLR_

### Review · Reviewer_V8Av · 2025-10-13

**Summary Of Contributions:**

**Summary:** This paper aims to understand the importance of non-linearity in recurrent models solving integration and memory-based tasks. The authors consider a recently proposed architecture, AL-RNN, to study this. AL-RNN allows controlling the number of non-linear units vs. linear units through the hyperparameter $P$, where $P=0$ indicates fully linear and $P=N$ for an RNN with $N$ units indicates fully non-linear. Furthermore, the AL-RNN formulation allows one to interpret the learned dynamics or solutions through bitcode analysis, i.e., looking at frequencies of activations over timesteps in specific piece-wise regions of the AL-RNN state space. The authors find in most tasks that at least a minimal amount of non-linearity improves performance, including some tasks where non-linearity is necessary in order to solve the task. They also show that fully non-linear models do not perform as well as those with less non-linearity, due to optimisation difficulties. Overall, the paper offers an experimental account on where and how much non-linearity is useful in integration and memory-related sequence tasks.

**Strengths:**
* The authors have carried out experiments on a diverse set of memory-related tasks, analysing in each the effect of non-linearity on task performance.
* Using an AL-RNN allows the authors to finely control the amount of non-linearity, enabling direct study of the main question.
* The AL-RNN formulation also allows in "symbolic" interpretability of the activations of the RNNs through bitcode analysis (i.e., 0 if the ReLU is inactive and 1 if active), allowing the authors to track the evolution of network activity over neighbouring regions/bitcodes to interpret the computations being performed.
* In general, the paper and experiments are presented well.

**Weaknesses/Suggestions/Questions:**
* The non-linearity here is restricted to ReLU or its variants due to the AL-RNN formulation. There is not much in terms of studying the roles of other non-linearities and gates. Could the authors comment on this? We do see in the addition task that you have gating enabled by non-linearity, however there could conceivably be tasks that LSTMs with forget gates or GRUs perform better than ReLU RNNs, e.g., some word- or character-level language modelling tasks with long-range dependencies. Could the authors comment on this?
* Is there an explanation for why, in the simpler fixed-delay copy task, any non-linearity more than $P=1$ consistently degrades performance?
* It isn't completely clear to me how to reconcile the worse performance of fully non-linear models in this paper on some tasks (except IMDB, see Fig. 2b) with the general observation in deep learning that non-linear transformers generally outperform linear models such as structured SSMs (ref. Sec. 5 on linear models' efficacy in long-range tasks). It is conceivable that there might be optimisation issues which could be alleviated by regularisation or sparsity constraints (Khona et al., 2023). Furthermore, it could just be a consequence of the tasks considered here (variants of integration and working memory recall) being much simpler than the computations in real-world tasks, causing more expressive models to overfit or over-linearise non-linear units without appropriate regularisation.
* Continuing from the previous points, while several tasks considered, each network is trained only on a single task or at best a multi-context task with a single underlying computational motif – and the tasks considered are variants of integration tasks or memory recall tasks, while networks in practice must implement more than just these computational motifs. Several previous works have studied recurrent neural networks trained to perform multiple cognitive tasks concurrently, where multiple dynamical motifs such as line attractors, ring attractors, fixed points, etc. may be implicated. Perhaps training individual models on multiple cognitive tasks (e.g., Yang et al., 2019; Khona et al., 2023; Driscoll et al., 2024) involving several computational motifs might further indicate the importance of non-linearity. It might also be interesting to see just how much non-linearity would be required in such a context, and how it relates to the amount of shared structure across these tasks (perhaps, e.g., more shared structure -> reuse of dynamics across contexts -> fewer partitions of state space required -> less non-linearity required?). The impact of regularisation and sparsity constraints is something to be accounted for as well.
* [Minor] The intro/related works could potentially also include references to hybrid attention + recurrence models, interleaving attention layers with linear SSMs while still achieving strong performance, e.g. Lieber et al., 2024. This relates to several points in the paper where a combination of both linear and non-linear computations results in best performance.
* [Minor] The authors have considered only AL-RNNs, and this is a limitation they have acknowledged, it might be interesting to extend these results to variants of linear attention (see Katharopoulos et al. 2020; could use $\phi(Q) = [elu(Q_i)\text{ if }i < n - p\text{ else }Q_i]_{i=0}^{n-1}$ and similarly for $K$).

**References:**
1. Khona, Mikail, et al. "Winning the lottery with neural connectivity constraints: Faster learning across cognitive tasks with spatially constrained sparse rnns." Neural Computation 35.11 (2023): 1850-1869.
2. Yang, Guangyu Robert, et al. "Task representations in neural networks trained to perform many cognitive tasks." Nature neuroscience 22.2 (2019): 297-306.
3. Driscoll, Laura N., Krishna Shenoy, and David Sussillo. "Flexible multitask computation in recurrent networks utilizes shared dynamical motifs." Nature Neuroscience 27.7 (2024): 1349-1363.
4. Lieber, Opher, et al. "Jamba: A hybrid transformer-mamba language model." arXiv preprint arXiv:2403.19887 (2024).
5. Katharopoulos, Angelos, et al. "Transformers are rnns: Fast autoregressive transformers with linear attention." International conference on machine learning. PMLR, 2020.

**Audience:**

Yes

**Audience Explanation:**

The submission is relevant to both sequence modelling in general and specifically to recurrent models used in computational neuroscience as well. In particular, the submission touches upon the following axes from TMLR's CfP:
* Experimental and/or theoretical studies yielding new insight into the design and behaviour of learning in intelligent systems – this is an experimental study on how non-linearity relates to task efficacy in sequence modelling/memory-related tasks.
* New approaches for analysis, visualization, and understanding of artificial or biological learning systems – the submission applies an existing model, AL-RNN, in a new context to shed light on when/how much non-linearity is useful in memory tasks.

**Broader Impact Concerns:**

None.

**Claims And Evidence:**

Yes

**Claims Explanation:**

I did not identify any major flaws in this paper, the experiments seemed mostly clear to me. The authors come up with 4 different contexts in which they study the central question of what non-linearity enables in sequences modelling and I believe the experiments and analyses provide evidence of having answered that question. A non-binding suggestion I have is to include a widely used set of multiple cognitive tasks in order to further study how much non-linearity matters in multi-task settings and also in tasks beyond integration. Otherwise, I just have questions and clarifications I believe the authors can address.

**Requested Changes:**

See the questions section of my initial review. In summary:
* Clarify the lack of analysis on other non-ReLU-like non-linearities.
* Clearer explanation for why increasing non-linearity worsens performance on the fixed-delay copy task.
* Comment on regularisation and sparsity in alleviating issues with fully non-linear models.
* Experiments on other cognitive tasks using different dynamical motifs, and multi-task RNNs trained on several cognitive tasks at once rather than separately.
* [Minor] Reference to/discussion on hybrid attention + SSM models such as Jamba and others.
* [Minor] Potentially experimenting with variants of linear attention to broaden the scope/impact and move beyond just recurrent models.

---

> ### Author Response · Authors · 2025-11-20
> **Rebuttal 1**
>
> First of all, many thanks for your fast, thorough and supportive review, particularly for the pointer to the multi-task setting, which we believe led us to some very interesting new results! We have uploaded an updated version of the paper, with important revisions highlighted in yellow. We respond to questions/comments in detail below.
>
> Role of other nonlinearity
>
> Our initial focus on ReLU-like nonlinearities is mainly driven by interpretability rather than a belief that other nonlinearities or gated architectures are unimportant. The AL-RNN is built around piecewise-linear (PWL) units because they allow us to partition state space into linear subregions, assign bitcodes to these regions, and then analytically characterize dynamics (fixed points, eigenvalue spectra) on a per-subregion basis, which enables our mechanistic analysis we perform. Doing the same type of analysis for smooth activations (like tanh) or gated architectures is more difficult, as their nonlinearities do not induce a simple linear tiling of state space. That said, we agree that it is important to understand whether our observations are specific to ReLU. We had therefore included experiments where we only changed the recurrent nonlinearity to GeLU, tanh, or hardtanh, see Tables 2 and (this might have been buried a bit in the Appendix). Across tasks, we observe the same qualitative pattern for all activations, where introducing a small number of nonlinear units yields performance gain, and as the number of nonlinear units approaches the full hidden size, performance deteriorates again. Thus the observation that sparse nonlinearity acts as a useful inductive bias, is not tied to ReLU specifically, and e.g. for the contextual integration task, a single nonlinearity, independent of what activation we use, unlocks the central task mechanism (see Fig. 10b). What does depend somewhat on the choice of nonlinearity is how cleanly interpretable mechanisms emerge. In the addition task, for instance, PWL activationsnaturally separate “integration” and “gating” into distinct linear subregions: one or few subregions carry out slow linear accumulation, while another subset of regions is selectively engaged at the two masked time points. With tanh, we see the network trying to achieve something similar by moving units between the near-linear regime around zero and saturated regimes, but the resulting partition is less clean, and performance is correspondingly worse by roughly an order of magnitude in MSE. We now describe these differences more explicitly in the revised manuscript.
>
> Other gated architectures
>
> We fully agree that gated architectures are effective on many sequence tasks, and there are good reasons they were introduced. Our goal in this paper is not to position AL-RNNs as a drop-in replacement or to claim superior performance in all regimes, but to provide a controlled, interpretable testbed for studying how much and what kind of nonlinearity is actually used in recurrent computation. Nevertheless, we included LSTM and GRU baselines on our main synthetic and classification tasks (Table 2). We also want to note that our competitive performance is helped by the manifold attractor regularization, which instantiates slow time scales in the AL-RNN dynamics (see also Schmidt et al., ICLR 2020). Without this regularization, AL-RNNs are indeed outperformed by LSTMs and GRUs. In this sense, the gating structure in LSTMs and GRUs can be viewed as an architectural inductive bias, which our approach achieves through explicit regularization. Both approaches address the same underlying computational need for maintaining information over time, but we hold that the AL-RNN framework, in combination with the simple regularization, makes this mechanism more transparent and amenable to mechanistic analysis. We now also explicitly acknowledge in the revision that there are certainly regimes, where more sophisticated gated or attention-based architectures are clearly preferable. We see our contribution as complementary: AL-RNNs provide a simple, analytically tractable framework in which mechanisms like gated integration, task switching, and context-dependent routing can be induced in a controlled way and then dissected in detail. We hope this makes the intended scope clearer.
>
> Further related work
>
> Thank you for this helpful pointer. We agree that hybrid architectures that interleave attention with linear SSM layers, such as Lieber et al. (2024), are very relevant to our framing, and we have included them in the related work section in the revised paper.

---

> > ### Author Response · Authors · 2025-11-20
> > **Rebuttal 2**
> >
> > Fixed-delay copy task
> >
> > In the fixed-delay copy task, the solutions that work well are simple: the network needs a mechanism to store the sequence over the delay, and then uses a clean nonlinear “switch” that separates encoding/decoding from storage. When (P) is small (e.g. (P=1–3), the AL-RNN reliably discovers exactly this kind of compact solution. The delay period is dominated by dynamics that stay almost entirely within a single linear subregion of the piecewise-linear system, and a single nonlinear unit is used as a switch to move the system into a different subregion at recall time. Bitcode analysis shows that only a tiny fraction of all theoretically available subregions are actually visited (see Fig. 19 in updated manuscript), and that most time steps lie in one or two dominant codes, which makes the resulting dynamics both robust and easy to optimize. As we increase (P), we are not adding anything the task fundamentally needs, but we do add many redundant nonlinear directions. In principle, a fully nonlinear model could ignore those extra degrees of freedom and implement the same compact mechanism as a low-(P) model, but in practice, gradient descent does not enforce this, and we instead observe that both storage and recall become “fragmented” across multiple neighbouring subregions (see also new Figure 26). We now further illustrate this influence of nonlinearity on optimization by including loss and gradient norm curves for the copy task (new Figure 25), which show that training for fully nonlinear models is substantially more unstable, with significantly higher gradient norms throughout and frequent bifurcations, while sparsely nonlinear models exhibit smoother training dynamics and occasionally discover the task mechanism suddenly, leading to rapid drops in loss that are not achieved by more nonlinear models.
> >
> > Generality of claims
> >
> > Thank you for this question, it is central to how we position our results. Our claim is much narrower than the general observation that strongly nonlinear Transformers outperform linear SSMs on large-scale real-world tasks. In response to this concern, we have now added preliminary experiments (Fig. 28, Appx. A.4) comparing fully linear, standard, and sparsely nonlinear variants of both Linear Transformers and S4 models on the addition and copy tasks. While these experiments do show task-dependent benefits of sparse nonlinearity, the nature of nonlinearity in these architectures is different from that in AL-RNNs, operating through layer-wise transformations rather than within the recurrence itself. Due to the complexity of these architectures, the computational role of this nonlinearity is not easily identified through DS analysis. We have updated the discussion (Sect. 5, Limitations) to acknowledge this distinction and note that extending our framework to modern architectures with different nonlinearity placement remains an important direction for future work.
> >
> > Regarding the results, within the family of AL-RNNs, we find that sparse nonlinearity in the recurrent layer outperforms both the fully linear and the fully nonlinear versions on four of the tasks we study. This is fully compatible with the fact that, at a different scale and with much deeper stacks, nonlinear Transformers are superior to purely linear models.  We have now clarified why fully nonlinear models are outperformed on these tasks (new Fig. 26), highlighting that the mechanisms are task-specific. For the copy task, more unstable training dynamics mean that good solutions are not discovered at all. In contrast, for the addition task and SCAN, fully nonlinear models can achieve very good or even optimal performance, but the variance across training runs is higher, reflecting less robust training rather than a fundamental inability to solve the task. For sMNIST, we find that full nonlinearity leads to systematically higher maximum eigenvalues of the slow manifold across digit classes, which causes overdispersion of the data manifold.
> >
> > Because the advantages of sparse nonlinearity are so task-dependent, we cannot make very general statements about where this principle holds, and we have now aimed to clarify this in the revised version of the paper. It is very likely that task-dependent appropriately chosen regularization (e.g., as in Khona et al., 2023) or initialization could mitigate these issues for fully nonlinear models on each task. Our approach can be seen as building in this sparsity structurally by fixing P, providing a simple and interpretable framework where nonlinearity itself is a tunable architectural parameter that provides one form of this regularization.

---

> > > ### Author Response · Authors · 2025-11-20
> > > **Rebuttal 3**
> > >
> > > Multi-task training
> > >
> > > Thank you for this suggestion and for pointing to the multi-task literature! In addition to the single-task results, we now also train a single AL-RNN jointly on a set of 11 tasks as discussed in Driscoll et al. (2024), which combines integration, memory, inhibition, and context-dependent decision tasks, some of which are similar to tasks we had already considered in the single-task setting. We vary the number of nonlinear units (P up to 64) while keeping the architecture and optimisation scheme fixed. In this multi-task regime, (P=0) can only solve two of the tasks, while a single nonlinearity dramatically improves performance, with as few as ($P \approx 4-8$) units already achieving good performance across tasks (in our minds, this is actually quite amazing). However, this performance advantage of nonlinearity is sample-size dependent. With limited training data (20 samples per task), sparse models substantially outperform fully nonlinear architectures, while if we train with sufficient data (500 samples per task), the model with all units nonlinear (P=M=64) indeed achieves the best aggregate performance across tasks. We therefore see our results not as evidence that "simple" tasks are the only place where sparse nonlinearity matters, but as showing that, within a fixed architecture, sparse nonlinearity provides a strong inductive bias toward compact, interpretable, and sample-efficient solutions, while full nonlinearity becomes increasingly useful as both task complexity and data availability increase. To address your point about shared structure, we analysed the bitcode distributions across tasks. For small (P), different tasks occupy overlapping sets of bitcodes, indicating that the same linear subregions (and thus dynamical motifs) are reused across tasks. As (P) grows, this overlap diminishes and tasks segregate into more distinct bitcode subsets, reflecting more specialised, task-specific partitions of state space. In other words, lower (P) enforces reuse of dynamics across contexts, while higher (P) allows specialised motifs at the cost of increased complexity. We quantified this via distributional overlap of the bitcode distribution, using the Jensen-Shannon entropy, based on which we defined a “similarity” score for the different tasks by bitcode usage. We found this overlap to be robust both across training runs, and even for different settings of P in the sparsely nonlinear regime. We believe this provides a useful new analysis of how different tasks share vs. specialize dynamics within a single recurrent model through the lens of their nonlinearity, and we thank the referee for prompting this very interesting extension.
> > >
> > > Extension to linear attention
> > >
> > > Thank you for this suggestion. We agree that extending the analysis to linear-attention variants would be very interesting and conceptually related to our approach. In response to your comment, we have now added preliminary experiments with Linear Transformers (Fig. 28, Appx. A.4), comparing fully linear, standard, and sparsely nonlinear variants on the addition and copy tasks. While these results show some task-dependent benefits of sparse nonlinearity, we could not thoroughly investigate the mechanisms involved, as our interpretability framework relies on dynamical systems properties of linear recurrence, and hence analyzing linear attention would require a completely different analytical toolkit. We now explicitly mention this limitation and extension in the discussion (Sect. 5).

---

### Review · Reviewer_8uNP · 2025-10-31

**Summary Of Contributions:**

This paper investigates the functional contribution of nonlinearity in modern sequence models, analyzing how nonlinear components influence representation power, stability, and expressivity. The authors decompose sequence architectures into their linear and nonlinear operators and systematically evaluate their roles in capturing temporal dependencies and hierarchical features. Through empirical and theoretical analysis, the paper provides deeper insight into when and why nonlinearities are essential and explores whether linear substitutes can approximate certain nonlinear effects.

**Additional Comments:**

**Weakness:**

1) While conceptually strong, the scope of datasets used for evaluation seems somewhat narrow.

2) Some theoretical sections are mathematically dense and may benefit from more intuitive explanations or diagrams.

3) The connection between the empirical findings and practical architecture design recommendations could be made more explicit.

**Questions:**

1) Have the authors explored whether nonlinearity can be adaptively modulated during training — for example, through learnable activation strengths — to balance efficiency and expressivity dynamically?

2) Could similar insights be extended to continuous-time or operator-based models (e.g., Neural Operators), where nonlinearity may play a different role in dynamics propagation?

3) How do nonlinearities affect stability and gradient flow in long sequence modeling, particularly in architectures without explicit recurrence?

4) Do the findings suggest an optimal trade-off between linear and nonlinear components that can guide model architecture design in future sequence models?

5) Can the decomposition approach proposed here be used to identify redundant nonlinear layers or to design interpretable linear surrogates for efficiency-critical applications?

6) Failure modes of full nonlinearity and training remedies: The author shows that fully nonlinear models often perform worse and sometimes “linearize” during training. Can you (a) characterize when and why this occurs (e.g., optimization landscape, bifurcations), and (b) test whether targeted initialization, regularization, or curriculum (e.g., start linear then gradually increase P) mitigates those failures?

7) The central claim is that sparse nonlinearity is a useful inductive bias. Do these findings transfer to structured linear SSMs (S4, Mamba) or Transformer-like architectures with nonlinear mixing?

**Audience:**

Yes

**Audience Explanation:**

Yes, the analysis of nonlinearity in modern sequence models, which examines how nonlinear components influence representation power, stability, and expressivity, motivates future sequence modeling, especially recurrent networks.

**Broader Impact Concerns:**

See Above

**Claims And Evidence:**

Yes

**Claims Explanation:**

For the most part, yes, for the claims made about AL-RNNs on the selected tasks, but not yet fully convincing as a general prescription for sequence modeling. The submission presents clear, targeted evidence that sparse nonlinearity in AL-RNNs yields interpretable, robust, and often optimal behavior on the studied benchmarks, but several gaps limit the broad applicability of these conclusions.

**Requested Changes:**

1) Run a small set of experiments that replicate the “sparse nonlinearity” vs “full nonlinearity” comparison on at least one structured SSM (e.g., S4/Mamba) and one Transformer-style model or token-mixer. Report whether the same inductive bias holds and discuss required modifications.

2) For cases where fully nonlinear models underperform, include training/validation curves, gradient norms, and any bifurcation/instability indicators. Try and report results for simple mitigations (e.g., progressive increase of nonlinearity, targeted initialization, or regularization) to clarify whether failures are representational or optimization-related.

3) Add experiments on a larger, practically relevant dataset (e.g., long-range language/audio/time-series) to probe whether the sparse-nonlinearity benefit scales to more realistic inputs and longer contexts.

4) Quantify bitcode/subregion usage and its variability across seeds and model sizes (plots or heatmaps). Add targeted ablations suggested by results (e.g., encoder vs decoder nonlinearity for SCAN, isolated removal of nonlinear units in critical layers) and visualize representative latent trajectories to support mechanistic claims.

---

> ### Author Response · Authors · 2025-11-20
> **Rebuttal 1**
>
> We thank the referee for the thorough review, which included many relevant pointers that led to further investigations that we hope clarified both the content and the scope of our findings. We have uploaded an updated version of the paper, with important revisions highlighted in yellow. We respond to questions/comments in detail below.
>
> Diagnostics/ablations
>
> Thank you for this suggestion. We have added several new diagnostics and ablations, and tried to clarify terminology. By a representational limit we mean that no parameter setting in the model class can implement the target mapping (the function lies outside the hypothesis class). By an optimization limit we mean that the mapping is representable, but standard training fails to reliably find the required parameters (e.g., due to a rugged loss landscape leading to unstable training dynamics). In our setting, the fully nonlinear AL-RNN strictly contains the sparse-(P) model as a special case, so when sparse models solve a task but fully nonlinear models trained from scratch do not, this is evidence against a representational limit and for an optimization limit.
> Empirically, training–validation curves show that degradation at high (P) is not classical overfitting, with training and test errors being very close. We have now aimed to clarify this by including loss and gradient norm curves during training on the addition and copy task (new Fig. 25), and adding clarification on the task-specific failure modes of fully nonlinear models, summarized in new Fig. 26. For the copy task, the substantially more unstable training dynamics mean that good solutions are never discovered at all. In contrast, for the addition task and SCAN, fully nonlinear models can actually achieve very good or even optimal performance, but the variance across training runs is higher, reflecting less robust training rather than a fundamental inability to solve the task. For sMNIST, we find that full nonlinearity leads to systematically higher maximum eigenvalues of the slow manifold across digit classes, which can cause overdispersion of the data manifold and degrade classification performance.Hence sparse nonlinearity provides advantages through different mechanisms, i.e. through enabling the discovery of simple solutions (copy task), ensuring more consistent convergence (addition and SCAN), and maintaining appropriate slow dynamics (sMNIST). We believe that these failure modes can be remedied by targeted regularization/training tweaks. However, the challenge is that failure modes differ across tasks, so it is unclear a priori what a "one-size-fits-all" regularization strategy would be. This is precisely where sparse nonlinearity offers a practical advantage, acting as  general structural constraint that implicitly provides regularization across multiple tasks simultaneously.
>
> Mitigation strategies
>
> Following your suggestion for training curves, on the addition task, we observed that fully nonlinear models initially decrease training loss fastest (Fig. 25), likely because their high nonlinearity allows them to stumble task mechanisms most rapidly (somewhat akin to the lottery ticket hypothesis), though they exhibit higher variance in final performance. This motivated us to test a simple iterative training scheme, where we progressively decreased P during training by pruning units that behaved most linearly (remaining predominantly on or off throughout solving the task). Initially, linearizing units often does not significantly worsen performance, and the resulting models exhibit similar structure to those trained with sparse nonlinearity from the start. However, at later stages of pruning, models sometimes exhibit catastrophic forgetting, where linearizing seemingly redundant nonlinear units dramatically worsens performance. In total, the iterative pruning scheme still yields worse average performance compared to initializing with sparse nonlinearity from the beginning. So while this progressive approach is very interesting, it requires further care to reliably provide benefits, e.g. by incorporating more sophisticated pruning criteria.

---

> ### Author Response · Authors · 2025-11-20
> **Rebuttal 2**
>
> Relevance for large-scale datasets
>
> Thank you for raising this important point. We had already included the speech command for audio classification (A.4.3. in Appx.), but left it in the Appendix, since the observed mechanisms closely mirror those found in sMNIST. We agree however, that larger, more heterogeneous settings are a natural stress test for whether the sparse-nonlinearity benefit persists. Motivated by the suggestion of Ref. V8Av and the multi-task literature, we therefore trained a single AL-RNN jointly on 11 tasks spanning integration, memory, and context-dependent decision tasks, keeping the architecture and optimization scheme fixed while varying the number of nonlinear units, see new Results 4.5.
>
> In this multi-task regime, ($P=0$) can only solve two of the tasks, while a single nonlinearity dramatically improves performance, with as few as ($P \approx 4-8$) units already achieving good performance across tasks. However, this performance advantage of sparse nonlinearity is critically sample-size dependent. With limited training data (20 samples per task), sparse models substantially outperform fully nonlinear architectures, while if we train with sufficient data (500 samples per task), the model with all units nonlinear ($P=M=64$) indeed achieves the best aggregate performance across tasks, which is consistent with the idea that more complex, heterogeneous domains benefit from higher expressive capacity.  We therefore see our results not as evidence that "simple" tasks are the only place where sparse nonlinearity matters, but as showing that, within a fixed architecture, sparse nonlinearity provides a strong inductive bias toward compact, interpretable, and sample-efficient solutions, while full nonlinearity becomes increasingly useful as both task complexity and data availability increase.
>
> To investigate the observed sample efficiency, we analysed the bitcode distributions across tasks. For small $P$, different tasks occupy overlapping sets of bitcodes, indicating that the same linear subregions (and thus dynamical motifs) are reused across tasks. As $P$ grows, this overlap diminishes and tasks segregate into more distinct bitcode subsets, reflecting more specialised, task-specific partitions of state space. In other words, lower $P$ enforces reuse of dynamics across contexts, while higher $P$ allows specialised motifs at the cost of increased complexity. We quantified this via distributional overlap of the bitcode distribution, using the Jensen-Shannon entropy, based on which we defined a “similarity” score for the different tasks by bitcode usage. Crucially, we found this overlap to be robust both across training runs, and even for different settings of $P$ in the sparsely nonlinear regime. We believe this provides a useful new analysis of how different tasks share vs. specialize dynamics within a single recurrent model through the lens of their nonlinearity.
>
> Clarification of scope
>
> We want to emphasize that “sparse nonlinearity beats full nonlinearity” is not our central message. It emerged repeatedly in single-task settings, but our aim is to provide a simple, interpretable framework in which nonlinearity is a tunable parameter and mechanisms (e.g., slow modes, gates, routing) can be read out and compared. The multi-task results above align with the reviewer’s intuition: in more realistic, composite settings, some additional nonlinearity is useful, while even here, a handful of nonlinear units capture much of the benefit, and can enforce a transfer learning effect that leads to better performance in low data contexts.
>
> Bitcode quantification
>
> Thank you for this helpful suggestion. Some of this analysis was already in the supplement (e.g., Fig. 16, 19 and 20 for seed/model-size effects; Fig. 24 for SCAN bitcodes and latent trajectories), and we have now added specific examples of bitcode distributions for well vs. bad performing models for the addition and copy task (new Figure 26), and included training and gradient norm curves w.r.t. P (new Fig. 25).
>
> On ablations: our architecture has no “critical layers,” since nonlinearity is only in the single layer recurrence. Regarding SCAN, we are not 100% sure we understood the suggestion, since in Fig. 8, we had already explicitly investigated separated where nonlinearity lives (encoder vs. recurrence), finding that encoder-only nonlinearity yields effectively linear recurrence with context encoded in embeddings.
>
> Finally, we include a new bitcode analysis for the multi-task suite (Results 4.5): using Jensen-Shannon similarity between task-wise bitcode distributions, we find that low-P models reuse overlapping subregions across tasks (shared motifs), while higher P produces more specialized, task-specific partitions. This extends our mechanistic account from single-task to multi-task settings and visualizes how sharing vs. specialization emerges as nonlinearity increases.

---

> ### Author Response · Authors · 2025-11-20
> **Rebuttal 3**
>
> Weaknesses (scope, diagrams, recommendations)
>
> Thank you for this feedback. We agree the original scope was narrow; in the revision we now include multi-task setting as discussed above, where a single AL-RNN is trained on 11 tasks.
>
> Regarding intuitive explanations: we have now updated Figure 1 and its caption to provide additional intuition on the overall structure of the AL-RNN and its role in splitting activity into composites of linear dynamics. We hope this clarifies the overall methodological backbone, but would be grateful for further pointers if things remain unclear.
>
> Questions: adaptive training
>
> While our main results modulate nonlinearity discretely via P, we did explore adaptive variants. First, we tested the adaptive P scheme as discussed above, linearizing units that are either predominantly "on" or "off". Second, we also tested the adaptive nonlinearity construction proposed in Brenner et al. (2024), where ReLU units are replaced by leaky ReLUs with learnable slopes. The slope parameter is passed through a steep sigmoid function to push it toward either 0 or 1, combined with a regularization term that encourages slopes near 1 (i.e., linearity). While this approach allows the model to dynamically determine which units remain linear versus nonlinear, we found that the outcome depends heavily on the regularization strength, leading to widely varying numbers of resulting linear subregions across different hyperparameter settings. For simplicity and interpretability, we therefore opted to directly control the number of PWL units through the fixed parameter P.
>
> Q: Continuous time/operators
>
> Thank you for this interesting suggestion. The specific PWL bitcode analysis we employ is tailored to discrete-time recurrent models with PWL activations and does not directly translate to continuous-time or operator-based architectures. For Neural Operators (e.g., FNOs), the propagation operates on entire function spaces rather than state transitions, and typically involves compositions of spatial transforms (e.g., FFTs) with forward mappings across layers. In our minds this functional form is difficult to represent as a PWL map. The most direct extension of our approach we could envision would be to continuous-time models like Neural ODEs, and indeed, RNNs with ReLU activations can be transformed into continuous-time representations (see e.g. Monfared & Durstewitz, Transformation of ReLU-based recurrent neural networks, ICML 2020). However, the discrete switching captured by bitcodes becomes less clear in continuous time. More broadly, we believe the underlying principle of analyzing solutions via composites of PWL activation functions, could potentially extend to these settings through architecture-appropriate analysis tools (e.g., time-indexed activation masks for Neural ODEs or FNOs operators). However, developing such frameworks rigorously is beyond the scope of the present work, but represents an interesting direction for future research.
>
> Nonlinearities and gradient flow
>
> We agree that an in depth analysis of architectures without explicit recurrence (e.g. Transformers) would indeed be very interesting. We have now included preliminary experiments of the role of nonlinearity in Transformers and SSMs (see new Fig. 28). However, given the time and paper constraints, we currently see this as falling outside the scope of our analysis, which focuses on recurrent dynamics where stability can be directly examined via the spectra of the linear subregions, combined with the switching behavior. Architectures without explicit recurrence (e.g., pure attention or operator/convolutional stacks) require a different toolkit and may warrant a dedicated paper in their own right.
>
> Recommendations for model design
>
> We believe the key takeaway is not a single optimal configuration, but rather that nonlinearity should be treated as a tunable model component that can be systematically included in hyperparameter scans. Our results across multiple tasks demonstrate that we cannot universally prescribe that recurrence should always be linear or always be fully nonlinear. Instead, the appropriate amount and placement of nonlinearity is task-dependent, and viewing it as a controllable architectural parameter provides a useful design principle for sequence modeling. While other sequence models have different structures, and e.g. fix recurrence to be linear for optimization reasons, nonlinearity still enters the architecture in other components (e.g., feedforward layers, activations between blocks). Our preliminary results for Linear Transformers and S4 models (Fig. 28) indicate that limiting nonlinearity can also be beneficial in these settings, though the underlying mechanisms may be harder to isolate given the different architectural constraints. We will make these recommendations more explicit in the revised manuscript.

---

> ### Author Response · Authors · 2025-11-20
> **Rebuttal 4**
>
> Method for finding linear surrogates
>
> While our AL-RNN architecture does not use stacked layers, the bitcode analysis provides a natural framework for identifying units that behave predominantly linearly. Specifically, units that remain consistently "on" or "off" across most inputs effectively contribute only linear transformations and can be detected through their bitcode usage patterns. As mentioned above, we have now included curriculum training experiments where we iteratively prune or linearize units that exhibit predominantly constant activation states. We find that removing or linearizing such units often does not significantly change the dynamics or loss values, allowing us to derive even sparser almost-linear models that may be more robust and interpretable for safety-critical applications. This approach provides a principled method for post-hoc simplification of trained models while preserving performance. However, the viability of linear surrogates remains fundamentally task-dependent. For tasks with inherently nonlinear mechanisms, purely linear approximations cannot fully capture the required computations.

---

### Review · Reviewer_Htgm · 2025-11-12

**Summary Of Contributions:**

The paper discusses the contributions of nonlinearity to computation.
AL-RNNs are trained on different tasks and their dynamics and computational strategy is analyzed and in particular linear and nonlinear networks are compared.


The paper investigates many different tasks, some of which are quite complicated.
The analysis of the trained networks is fairly extensive.


On the other hand, the claims in the paper are about RNNs in general, even though only one specific architecture was tested with specific hyperparameters.

**Audience:**

Yes

**Audience Explanation:**

How tasks are solved and how choices to the network architecture (e.g. nonlinearity) are a very important topics in successfully training RNNs and making them interpretable through post-training analysis.

**Claims And Evidence:**

No

**Claims Explanation:**

1. "Nonlinearity Reshapes Slow Modes in Temporal Integration"


"Similar slow-mode or line attractor based integration mechanisms", while the dynamics in these networks looks more like an approximate line attractor (i.e. a slow (invariant) manifold), see also
Ságodi, Á., Martín-Sánchez, G., Sokol, P. and Park, M., 2024. Back to the continuous attractor. Advances in Neural Information Processing Systems, 37, pp.66856-66906.

What is the contribution from the nonlinear part to slow dynamics? Slow invariants can develop in nonlinear networks (Ságodi, 2024).

Finally, it is unclear how tokenization influences the learned computation.

"Bitcode analysis provides a clear mechanism by which nonlinearity increases performance: each class aligns with its own set of closely neighboring bitcodes". It is unclear how the first statement follows from the second.

"These results support the view that nonlinearity in this setting acts not as a core computational mechanism, but as a tool for improving expressivity"
What are the set of core computational mechanisms? How do you measure expressivity?



2. "Minimal Nonlinearity Stabilizes Transients For Structured Memory Recall"
"This cycle is stabilized through the interplay of two distinct spectral properties: one with a contracting virtual fixed point ($\lambda_{\text{max}}\approx$ 0.992) and the other around a diverging fixed point ($\lambda_{\text{max}}\approx$ 1.005) which drives the initial transient dynamics."
Is this the eigenspectrum of the cycle? Or of two different fixed points?
But how does that then "stabilize the cycle"?
And where is this information about the spectrum coming from, i.e., how was the Lyapunov spectrum calculated?
Finally, it is not readable from Fig. 4b that the Lyapunov exponent is approximately 0.

It remains unclear how the networks trained on the Copy Task With Variable Delay solve the task.
"with decoding accuracy from the bitcode alone reaching around 60\%" seems to indicate that the subregions are not the correct level at which decoding takes place.

3. Nonlinearity Enables Gating in the Addition Task
It is clear that nonlinear networks can implement gating mechanisms more easily.
However, it is possible to let purely linear model implement one, especially if the task length is nonvariable.


4. "Nonlinearity Enables Task Switching in a Stimulus Selection Task in Joint Task–Neural Modeling"
To show task switching in trained models and the contribution from nonlinearity to this, it would be better to use a more simple task.
This task, involving real-world data, although very interesting and important to show the usefulness of AL-RNN, is too complex to clearly test the effects of the nonlinearity in the trained networks.
For example, this task shows that sparsity is a bad inductive bias, see Fig. 6e (against the claims of the authors).


5. "Nonlinear Decoding Enables Compositional Generalization
This is again an interesting setting to test."

6. "This may partially explain the efficiency of linear SSMs on long-range tasks".
It is surprising that this is the authors takeaway from the paper, while they showed that purely linear networks were insufficient on some of the tasks under consideration.


7. "highlights a useful inductive bias of sparsely nonlinear model"
There is some evidence coming from the networks trained on the different tasks that combining linear units with nonlinear ones is beneficial.
However, whether it is a certain size of (non)linear is optimal or whether it is sparsity is unclear.

8.
Overall, it is unclear whether the contributions are specific to AL-RNN or whether they are something general.


For example:
"The PWL structure of the AL-RNN makes these mechanisms transparent,
directly linking switches in model dynamics to computation."
I am not sure where it is show that there is a direct link.





Claims about used methods:
It is claimed that AL-RNN allows fine-grained control.
Fine-grained control of the nonlinearity would suggest that along a certain parametrization of the nonlinearity you slightly vary it and describe how the representations change (possibly smoothly).


Claims about/with proposed metrics:
1. It is unclear what the bitcode is showing about the dynamics and about the contribution of the nonlinearity (beyond the subregion type analysis which is specific to PWLs, but  it cannot be applied to the other nonlinearities)


Claims that need ablation experiments:
1. "Purely linear AL-RNNs cannot solve this task" and "Our pipeline accurately identifies that, without nonlinearity, the network fails on this task". This is only established for the current hyperparameter setting.
The following hyperparameters should be explored to make these statements stronger.:
i. Total hidden size.
ii. tau
iii. Mreg
iv. Task related: length, variable timing etc.

2. "tanh cannot enforce a clean partition" this is probably also total size dependent.


3. Contributions to slow modes from nonlinearity.
The paper lacks investigation of slow modes in the nonlinear part of the network. PWL networks can also create line attractors, see for example
Ságodi, Á., Martín-Sánchez, G., Sokol, P. and Park, M., 2024. Back to the continuous attractor. Advances in Neural Information Processing Systems, 37, pp.66856-66906.





Questions:
1. Task implementation:
a. Integration relies on reliable end of task. This leads to the summation being decomposed into an immediate and temporally unrolled component.

ii.

2. "such solutions are highly fragile to noise and perturbations"
But the networks are trained without noise, so why is robustness important?
"Minimal nonlinearity therefore greatly improves performance
and robustness." There are no experiments to test robustness properties of either the linear or nonlinear networks.

3. "These results indicate that variable timing shifts the computational burden from storage to decoding." The computational burden of what? Storage is still relevant in this task, right?

4. Why were different Mregs chosen for the different tasks?


Small claims:
1. "At $P=50$, class manifolds are spatially distinct and well-separated"
Needs: density or alpha or some quantification of average overlap rather than a single example with a visualization based evaluation with sub-optimal visualization.
2. "we analyzed a representative AL-RNN with P = 1 that perfectly solves the task". In terms of MSE? Cross-entropy?




### Clarity

1. Figure 3:  "first 3 PCs" but the axes are missing so the projection looks 2D.
1. The description of the bitcode is confusing: "To quantify the distribution of latent states across the linear subregions, we extract the last $P$ components of each latent state trajectory"
seems different from Eq. 3.

**Requested Changes:**

## Main requested changes

1. Other architectures should be tested to support the general claims or the claims should be about the AL-RNN architecture specifically.

2. Ablation experiments should be performed (see above).

---

> ### Author Response · Authors · 2025-11-20
> **Rebuttal 1**
>
> We thank the referee for their detailed and thorough review, which we hope enabled us to significantly clarify the content and scope of our paper. We have uploaded an updated version of the paper, with important revisions highlighted in yellow. We respond to questions/comments in detail below.
>
> Slow dynamics
>
> Thank you for this insightful comment. We agree that our original phrasing about "line attractor based integration mechanisms" could have been more precise. As correctly pointed out by the reviewer, slow invariant manifolds can indeed develop in nonlinear networks, and what we observe is more accurately described as an approximate line attractor or slow manifold rather than a strict line attractor. We wish to clarify our intended contribution: we are not arguing that only linear components can generate slow modes. Rather, our key finding is that nonlinearity enables class-specific reshaping of slow dynamics within what remains fundamentally a slow integration process. To address this, we have now included a stability analysis along the classification trajectories (Fig. 13c in the updated paper). Since the AL-RNN latent dynamics are composites of different linear systems, we can perform stability analysis within each subregion along the integration path. Since the input projection itself is linear (via the $C$ matrix), it is the autonomous dynamics within each subregion that reshape the latent manifold. We tracked the maximum eigenvalue (max EV) along classification trajectories for different digit classes across models with varying degrees of nonlinearity. First, we find that linear models and those with minimal nonlinearity maintain maximum eigenvalues very close to 1 across all digit classes, indicating a uniform slow mode shared across classes. Second, models with moderate nonlinearity continue to maintain eigenvalues close to 1 but show class-specific variations, demonstrating that nonlinearity subtly reshapes the slow dynamics for each digit class while preserving the overall slow integration character. Third, fully nonlinear models exhibit systematically larger maximum eigenvalues consistently across all trained models, which provides a mechanistic explanation for the high dispersion we observe at high $P$. We hope this analysis helps provide another perspective on how the nonlinear components contribute to slow dynamics by enabling fine-tuned, class-specific modulation of the slow manifold structure. Each digit class effectively operates within its own slow mode, facilitating better class separation while maintaining the overall computational mechanism of slow integration.
>
> Tokenization
>
> We assume the reviewer means the GloVe embeddings for the IMDb sentiment classification task? These provide semantically meaningful vector sequences where valenced words (e.g., "good" vs. "bad") are already separable in the embedding space. The role of the AL-RNN is then to selectively integrate these valenced signals over time and classify the overall sentiment (Fig. 2). This clarifies why the task does not require recurrent nonlinearity: the complexity of word representation is handled by the embedding, while the AL-RNN performs a computationally simple memory and integration process, essentially implementing the classic 2D line attractor mechanism we cite in the main text. We acknowledge that if the entire task were performed end-to-end by an AL-RNN (e.g., processing raw character sequences), the requirements would be different. However, our goal here is to showcase that sentiment classification, when formulated as integration of pre-valenced inputs, is a fundamentally linear memory process that does not require recurrent nonlinearity. We have clarified this further in the main text.
>
> Bitcode analysis
>
> In the revision, we aimed to  the mechanistic link explicit by connecting more directly to the analyses in Figs. 15 and 16. There, we align stroke patterns in the input image with switches between linear subregions. This indicates that nonlinearity is used to create input-dependent transitions between locally linear regimes in a task-aligned way. Second, we quantify how different levels of nonlinearity shape the geometry of class manifolds in latent space: medium levels of nonlinearity lead to class manifolds that relatively evenly distributed, while fully nonlinear models show more imbalanced, fragmented structures and worse accuracy. Beyond the classification example discussed in the main text, bitcode analysis systematically reveals distinct functional roles across tasks, summarized in Table 1: in the copy task, it separates delay-period storage from cue-triggered recall. In the addition task, it separates linear storage from the selective integration of inputs. In contextual tasks (both synthetic and the PFC-1 dataset), it routes different rules into distinct subregions. In SCAN, it organizes latent space by syntactical rules.

---

> ### Author Response · Authors · 2025-11-20
> **Rebuttal 2**
>
> Clarification copy task
>
> Thank you for these detailed questions. For the fixed-delay copy task, the Lyapunov spectrum and the eigenvalues we report are computed per linear subregion of the AL-RNN. Within each subregion, the dynamics are linear and governed by the effective Jacobian $J = A + W_{\text{masked}}$. For each such $J$, we can directly compute the eigenvalue spectrum and, in particular, the dominant eigenvalue that determines local contraction or expansion (see Eq. 5 and the analysis procedure described in Appx. A.2, especially Eq. 8). The numbers 0.992 and 1.005 refer to the dominant eigenvalues in two different linear subregions that are visited on the limit cycle, not to a single global spectrum of the cycle. In the “expanding” subregion, the leading eigenvalue is slightly larger than one $≈ 1.005$, giving weakly divergent dynamics; in the “contracting” subregion, the leading eigenvalue is slightly smaller than one $≈ 0.992$.
>
> To characterize the stability of the full trajectory, we estimate the maximum Lyapunov exponent along the generated sequence by multiplying the Jacobians along the trajectory and periodically re-orthogonalizing, following the standard algorithm summarized in Appx. A.2, Eq. 8. For the solution shown in Fig. 4b, this maximum Lyapunov exponent is numerically very close to zero $≈ 1e−4$, which is consistent with a stable limit cycle: trajectories neither diverge exponentially nor converge to a fixed point, but remain on a marginally stable closed orbit. We agree that this quantitative fact is not directly visible in Fig. 4b itself; the figure is meant to illustrate the alternation between subregions, not to display the exponent. In the revision, we will make this explicit in the caption and directly report the numerical value of the exponent in the main text when we first discuss the cycle.
>
> The phrase “the cycle is stabilized through the interplay of two distinct spectral properties” was meant to summarize how this alternation between subregions yields a robust periodic orbit without having to finely tune a single linear system to have an eigenvalue exactly at one. Intuitively, the slightly divergent dynamics in one subregion push the trajectory outward, while the slightly contracting dynamics in the other subregion pull it back in. Once the trajectory has settled onto the orbit that alternates between those two regimes (as visible in Fig. 4b, bottom, where the PWL unit repeatedly crosses the switching boundary), the net effect over a full cycle is neutral expansion (Lyapunov exponent ≈ 0), which stabilizes the limit cycle. We will clarify in the text that “two distinct spectral properties” refers to the dominant eigenvalues of the local Jacobians in the two subregions visited on the cycle and explain more explicitly how this alternation provides a piecewise-linear route to stable oscillations that would be harder to obtain with a single global linear recurrence.
>
> Copy task with variable delay
>
> We have clarified our description in the updated manuscript. Our intention was not to claim that subregions alone fully “solve” decoding, but to contrast the mechanism with the fixed-delay case. For fixed delays, successful models typically store the sequence by evolving within a single linear subregion during the delay and then decode entirely within a different subregion once the cue arrives. For variable delays, the delay-period dynamics still remain in a largely linear regime, but during recall the trajectory traverses a pattern of subregions, and symbol identity is reflected in the sequence of bitcodes rather than in a single, cue-triggered region (Fig. 17 in updated manuscript). The statement that “decoding accuracy from the bitcode alone reaches around 60%” is meant to show that the bitcode sequence contains substantial information about the recalled symbols (well above the 25% chance level) but not that it is the only relevant representation. The actual decoder in the trained AL-RNN still operates on the full latent state, hence the bitcodes are a coarse, symbolic summary that allow us to see that symbols are encoded in structured transitions across subregions in the variable-delay case, whereas they are confined to a single decoding region in the fixed-delay case. We will revise the relevant section to make this distinction clearer.

---

> ### Author Response · Authors · 2025-11-20
> **Rebuttal 3**
>
> “Expressivity” and “core computational mechanisms”:
>
> We agree that our wording was too informal. In this paper we intended to distinguish between tasks that are impossible without nonlinearity (such as the addition problem or our contextual integration task), and tasks like IMDb or sMNIST where a linear model already achieves reasonably high accuracy. In the latter case, the essential computation of slow evidence accumulation along a low-dimensional direction is already realizable in the fully linear AL-RNN: we show that such models exhibit a dominant eigenvalue close to one and that the associated eigenvector is almost perfectly aligned with the first PC of the latent trajectories, which separates the classes. In that sense, the “core” mechanism is the line-attractor-like slow mode, which Table 1 lists as the dominant motif for these tasks and which is present regardless of whether P>0. When we wrote that nonlinearity acts as a tool “for improving expressivity,” we meant that increasing P allows the model to flexibly shape how different classes are embedded, for example, by assigning them to different linear subregion with slightly different stability properties. We do not introduce a formal expressivity measure, but used the term in a task-specific, empirical sense: at fixed latent dimension M, models with sparse nonlinearity achieve higher accuracy and more structured class separation in latent space than purely linear ones, even though the underlying slow-mode mechanism is shared. In the revised manuscript, we will replace “expressivity” with more concrete phrasing such as “flexibility in shaping class-dependent dynamics”.
>
> Gating in addition task, linear models
>
> In our paper, we use “gating” in a specific sense: the same input stream is processed by different internal update rules depending on a control signal, so that some inputs are selectively integrated and others are effectively ignored. In the addition task, this means “integrate the value at time $t$ only if the mask at time $t$ is on.” In the AL-RNN with PWL units this is visible in the dynamics: for most of the sequence the trajectory stays in one “integration” subregion, and only at the two masked time points does the mask push the system into a distinct subregion. This conditional switching, illustrated in Fig. 5 and the bitcode analysis Fig. 20, is what we mean by gating. Empirically, when we train fully linear AL-RNNs with the same architecture, latent sizes, and regularizations, loss does not decrease and stays constant for all models (see Fig. 25b). We have now further supported this empirical point with a formal proof of why a purely linear AL-RNN fails to solve the addition task.
>
> Clarification task switching
>
> We agree that task switching is best understood first in a simple, controlled setting, which is why we started with a synthetic benchmark in Appx. A.4.1. There we use a simple contextual integration task, show that purely linear models remain at chance level, and that introducing even a single nonlinear unit is sufficient to implement rule-dependent routing. For the rodent stimulus selection task, we proceeded in two stages. First, we trained the AL-RNN only on the behavioral task (Fig. 6c) and found that low-dimensional linear models already solved it, consistent with the reviewer’s intuition that the task itself does not intrinsically require strong nonlinearity. Second, we then added the real spike data and trained the same latent model jointly on choices and neural activity. In this joint setting (Fig. 6e), increased nonlinearity improves both task accuracy and spike reconstruction, indicating that the additional complexity and variability in the neural data benefit from a more expressive recurrent model, even though the underlying task rule is linearly solvable. We do not intend to claim that “sparse nonlinearity is always the best inductive bias” for every task, but our main empirical claim is that in four quite different benchmark settings (addition, copy, sMNIST, Speech Commands) sparse nonlinearity repeatedly yields simple mechanisms that lead to more robust training. The PFC-1 experiment is, in fact, a useful counterpoint: it shows that our framework also reveals when additional nonlinearity is helpful (see also new multi-task results), here, to accommodate noisy, heterogeneous neural responses, without contradicting the simpler mechanisms observed on controlled benchmarks. At the same time, even in this dataset where more nonlinearity is more helpful, we still find a very clear and simple to understand mechanistic link between successful models and the distribution of rules to linear subregions. We have clarified this in the manuscript by tempering the generality of the “sparse nonlinearity” statement and by emphasizing that the primary goal is to provide an interpretable framework for probing how nonlinearity is used across different regimes.

---

> ### Author Response · Authors · 2025-11-20
> **Rebuttal 4**
>
> SCAN, Multitask
>
> Thank you for this positive remark! In the revision Section 4.5.), we now also include a multi-task analysis in which different tasks are forced to share the same AL-RNN and hence, at low P, must reuse overlapping sets of bitcodes across tasks. This reuse effectively enforces a compositional structure in the latent code: the same subregions (bitcodes) are recruited in different task contexts and combined to implement multiple behaviors. We see this as complementary evidence that sparse nonlinearity and the PWL structure naturally support compositional reuse of dynamical motifs.
>
> Linear SSMs
>
> Thank you for pointing this out; we see how the original phrasing could be misleading. We did not intend to suggest that purely linear models are sufficient for all long-range sequence tasks. Indeed, several of our own results show that they are not. Our intended takeaway was that across many of the tasks we study, the core memory mechanism itself is often implemented by a slow linear mode, with nonlinearity used sparingly for operations such as gating, routing, or decoding (see summary Table 1). This structure is broadly consistent with the design of modern structured SSMs, which typically combine stacks of linear state-space layers with nonlinear mixing layers, rather than being “purely linear” end to end. However, here the precise role of nonlinearity is more difficult to understand, since it does not explicitly occur in the recurrence. Nevertheless, we have now added experiments analyzing task performance for S4 and Linear Transformer for different levels of nonlinearity between layers. These results illustrate that at least for the setting of our benchmark tasks, sparsely linear models (here denoting a single nonlinear readout layer), can perform as well or better as fully nonlinear stacks of layers.
>
> Clarification sparse nonlinearity
>
> The referee is correct that the key question is whether it is the *sparsity* of nonlinearity (the ratio of linear to nonlinear units) or simply the overall capacity that drives performance. To disentangle these two factors, we deliberately chose overparameterized regimes in our experiments (e.g. $M=50$ units for the addition task and$M=100$ for sMNIST), which allows us to vary the sparsity ratio (via $P$) while keeping total capacity constant. Our results indicate that the *mixing* of linear and nonlinear units is crucial, not simply the total number of units. We observe performance degradation in fully nonlinear models across different total capacities, suggesting this is not merely a capacity issue. Mechanistically, we find that in fully nonlinear networks, many PWL units become effectively linearized during training (e.g., Fig. 19b), with their activations pushed far from the switching boundary so that only a subset actually contributes nonlinear switches. In that sense, a fully nonlinear model ends up recovering something close to a sparsely nonlinear solution, but in a more indirect and harder-to-optimize way. The fact that higher $P$ models often show higher training error, rather than overfitting, supports this interpretation.
>
> Computation and bitcode switches
>
> We agree that most of the computational motifs are not unique to AL-RNNs. Our claim is rather that the PWL structure makes them easy to analyze. By a “direct link” between switches and computation we mean that, in this architecture, we can explicitly align changes in bitcodes (i.e. transitions between linear subregions) with events and then verify that these transitions are functionally meaningful. In sMNIST we observe that switches in the PWL units coincide with informative stroke segments of the digit image, and that different digits end up in distinct, class-specific subregions (Fig. 15&16). In the copy task, the PWL unit remains in one subregion during the delay, and then switches into a different subregion for decoding (see also Fig. 19). In the addition task, bitcode transitions are sharply concentrated at the masked time points, separating integration subregions from gating (Fig. 20). In the rodent stimulus-selection task, high-performing models route trials from different rules into almost non-overlapping subregions at stimulus onset, and the degree of this rule-based separation is strongly correlated with task accuracy. In SCAN, initial states for compositional commands are mapped to different subregions than simple commands. Taken together, these examples show that their PWL structure provides a concrete handle for mapping specific switches in dynamics onto specific computational roles. To check whether these observations relate only to ReLU, we also repeated the main experiments with other nonlinearities (Table 2). Here we observe the same qualitative pattern (e.g., improved performance for smaller P). However, smooth activations such as tanh cannot be analyzed via bitcodes in the same way, but still show similar functional decompositions (e.g. selective integration in the addition task).

---

> ### Author Response · Authors · 2025-11-20
> **Rebuttal 5**
>
> "Fine-grained control"
>
> We understand that “fine-grained control of the nonlinearity” could be understood as continuously tuning a parameter of the nonlinearity. In this work, we use “fine-grained” in a architectural sense: the AL-RNN lets us systematically and incrementally vary the number of nonlinear (PWL) units using a fixed latent dimension M, and thus control how much of the recurrent state space is subject to nonlinear switching. This gives us a clean, discrete knob with which we can probe how adding or removing nonlinear units changes both performance and the structure of the learned dynamics (e.g. similarity matrices and bitcode usage as a function of P).  Second, we also tested the adaptive nonlinearity construction proposed in Brenner et al. (2024), where ReLU units are replaced by leaky ReLUs with learnable slopes. The slope parameter is passed through a steep sigmoid function to push it toward either 0 or 1, combined with a regularization term that encourages slopes near 1 (i.e., linearity). While this approach allows the model to dynamically determine both the number of linear subregions and which units remain linear versus nonlinear, we found that the outcome depends heavily on the regularization strength, leading to widely varying numbers of resulting linear subregions across different hyperparameter settings. For simplicity and interpretability, we therefore opted to directly control the number of PWL units through P.
>
> Bitcodes
>
> The bitcode shows which subregions are used at what stage of the task. We think it provides a number of important insights with respect to single task computations, see detailed responses above. As we now also show in a novel analysis, the bitcode distribution for the multi-task training additionally structures similarity across different tasks.
>
> Ablation linear models
>
> Thank you for this suggestions. We have now added a formal argument of why the addition task is not solvable with a linear model, which we further highlighted with empirical studies with linear AL-RNNs on the addition and copy task how they fail to solve the tasks for different regularizations and model sizes.
>
> Tanh
>
> When we wrote that “tanh cannot enforce a clean partition,” we were referring primarily to a property of the activation function itself, not to an absolute limitation independent of model size. Piecewise-linear units like ReLU introduce hard switching boundaries at zero, which naturally induce a partition of state space into linear subregions. In contrast, tanh is smooth and saturating, so there is no intrinsic, exact boundary where dynamics change discontinuously. In our experiments we indeed see that tanh models “attempt” to create an effective partition: units involved in gating are often pushed far into saturation (e.g. to activations far beyond 0), so that, functionally, they behave almost like binary switches. However, this separation is softer and more entangled with the rest of the dynamics, and we consistently found that the resulting solutions achieved slightly worse performance than their PWL counterparts at comparable sizes.
>
> Robustness
>
> There is no explicit noise in the models, but we use random initialization and stochastic optimization with Adam, so there is naturally variance across training runs. We have now illustrated that the failures of fully nonlinear models are often a question of training robustness rather than representational limitations. For tasks like addition and SCAN, fully nonlinear models can in principle find good solutions, as evidenced by the best-performing runs, but they do so less reliably, resulting in higher variance across runs compared to sparsely nonlinear models (Fig. 26). When we refer to "robustness," we specifically mean consistency across different initializations and training trajectories. The higher training variance and occasional failure modes of fully nonlinear models (e.g., catastrophic bifurcations in the copy task) demonstrate reduced optimization robustness rather than sensitivity to external perturbations. We will clarify this distinction in the revised manuscript.
>
> Copy task
>
> Indeed, storage remains important in the variable timing task, and the mechanism for storage is similar to the fixed delay case, remaining focused on a small set of subregions. What changes is the complexity of the decoding process (see also Fig. 17 in revised manuscript). In the fixed delay task, decoding is often implemented in a single subregion. In contrast, for variable timing, decoding is distributed across multiple subregions, and the relationship between these decoding subregions and the stored symbols is more complex and less transparent (as reflected in the 60% decoding accuracy from individual subregions). We have clarified this statement to better convey that both storage and decoding mechanisms are present in both tasks, but their relative complexity shifts with variable timing.

---

> ### Author Response · Authors · 2025-11-20
> **Rebuttal 6**
>
> Question: Regularization
>
> We did not choose different $M_{reg}$​ values for different tasks. Following the recommendation in Schmidt et al. (2021), we consistently set $M_{reg}=M/2$ (half of the total latent dimension) across all experiments.
>
> Small claims: class separability
>
> We had indeed quantified class manifold separability in Fig. 15 through several metrics beyond visual inspection. We computed the CV, Gini coefficients, and Max-Min ratios of variance across all digit classes, confirming that the spatial entanglement visible in the visualization reflects systematic imbalance in how fully nonlinear models partition latent space across classes.
>
> Copy talk performance
>
> For the copy task, we evaluate performance using symbol accuracy (i.e., the proportion of symbols correctly reproduced). For this model with $P=1$, all output symbols match the encoded symbols perfectly across all test sequences, achieving $100\%$ accuracy and thus constituting a "perfect" solution. We have clarified this metric in the text.
>
> Figure 3
>
> We apologize for the lack of clarity, and have now included the axes in the plots.
>
> Clarification bitcode assignment
>
> The AL-RNN latent state is partitioned into linear units (the first $M-P$ dimensions) and PWL units (the last $P$ dimensions), as illustrated in Fig. 1. To compute the bitcode distribution, we extract only the last $P$ components of each latent state trajectory, corresponding to the PWL units, and then apply the binary encoding defined in Eq. 3 based on whether each PWL unit is above or below zero. This gives us a $P$-bit binary representation (bitcode) for each time step, which we can then analyze to understand which linear subregions are being used.
>
> Requested changes: other architectures and clarification of scope
>
> We have significantly reformulated several sections based on this input. The breadth of nonlinear sequence models means that it is indeed impossible to make general claims about all sequence modeling based on our results, and we have accordingly rephrased our results to make them more specifically about AL-RNNs. However, to expand the range of applicability to other architectures, we had tested multiple types of nonlinear activations and two other standard sequence models (LSTMs and GRUs). We have now also included preliminary experiments examining the influence of sparse nonlinearity on task performance for Linear Transformers and S4 models (Fig. 28). While the different architectural structures make the bitcode analysis toolkit developed here not directly applicable, we still find that the idea of nonlinearity as a general structural tunable hyperparameter can be meaningfully extended to these architectures as well. These preliminary results suggest that strategic placement and sparsity of nonlinearity may offer benefits across different sequence model families, though the specific mechanisms and optimal configurations are likely architecture-dependent.

---

### Decision · Action_Editor_88W3 · 2025-12-14

**Recommendation:** Accept with minor revision

**Additional Comments:**

One of the reviewers noted that they believed the "main message of the text [should be changed] to reflect that the empirical results only support the effect of using nonlinearities for the used architectures, especially PWL AL-RNN." I would therefore request the authors add a little more discussion of this in the Introduction and Discussion, just to be clear and up-front about the results/contributions of the paper.

**Audience:**

Yes

**Audience Explanation:**

All three reviewers agree that there would be a broad part of the TMLR audience that would be interested in these results.

**Claims And Evidence:**

Yes

**Claims Explanation:**

Two of the three reviewers agreed that the claims were supported by accurate, convincing, and clear evidence. The reviewer that did not believe the claims were supported pointed out that theoretical support was lacking. While I agree that the authors can (and should) make the language surrounding their contributions clearer (see request for minor revisions), I believe that this is an addressable issue. Additionally, that the other 2 reviewers did not feel this same way, suggests that the reviewer who did not think the claims were supported may have a specific expectation of the paper (possibly due to the title of the paper).